# Generalization Analysis of Message Passing Neural Networks on Large Random Graphs

**Sohir Maskey**[*]
Ludwig-Maximilian University of Munich[†]
`maskey@math.lmu.de`

**Ron Levie**[*]
Technion - Israel Institute of Technology [‡]
`levieron@technion.ac.il`

**Yunseok Lee**
Ludwig-Maximilian University of Munich[†]
`ylee@math.lmu.de`

**Gitta Kutyniok**
Ludwig-Maximilian University of Munich[†]
University of Tromsø[§]
`kutyniok@math.lmu.de`

## Abstract

Message passing neural networks (MPNN) have seen a steep rise in popularity since their introduction as generalizations of convolutional neural networks to graph structured data, and are now considered state-of-the-art tools for solving a large variety of graph-focused problems. We study the generalization error of MPNNs in graph classification and regression. We assume that graphs of different classes are sampled from different random graph models. We show that, when training a MPNN on a dataset sampled from such a distribution, the generalization gap increases in the complexity of the MPNN, and decreases, not only with respect to the number of training samples, but also with the average number of nodes in the graphs. This shows how a MPNN with high complexity can generalize from a small dataset of graphs, as long as the graphs are large. The generalization bound is derived from a uniform convergence result, that shows that any MPNN, applied on a graph, approximates the MPNN applied on the geometric model that the graph discretizes.

## 1 Introduction

A graph is an abstract structure that represents a set of objects along with the connections that exist between those objects. In many important fields, such as chemistry, biology, social networks, or computer graphics, data can be described by graphs. This has led to a tremendous interest in the development of machine learning models for graph-structured data in recent years. A ubiquitous tool for processing such data are graph convolutional neural networks (GCNNs), which extend standard Euclidean convolutional neural networks (CNNs) to graph-structured data.

Most GCNNs used in practice can be described using the general architecture of *Message Passing Neural Networks (MPNNs)*. MPNNs generalize the convolution operator to graph domains by a neighborhood aggregation or message passing scheme. By $\mathbf{f}_i^{(t-1)}$ denoting the feature of node $i$ in layer $t-1$ and $\mathbf{e}_{j,i}$ denoting edge features from node $j$ to $i$, one layer in a message passing graph

---

[*]The first two authors contributed equally to this work

[†]Department of Mathematics, LMU Munich, 80333 Munich, Germany

[‡]Faculty of Mathematics, Technion - Israel Institute of Technology, 32000 Haifa, Israel

[§]Department of Physics and Technology, University of Tromsø, 9019 Tromsø, Norway

36th Conference on Neural Information Processing Systems (NeurIPS 2022).

neural network is given by

$$\mathbf{f}_i^{(t)} = \Psi^{(t)}\Big(\mathbf{f}_i^{(t-1)}, \mathbf{AGG}\big\{\Phi^{(t)}(\mathbf{f}_i^{(t-1)}, \mathbf{f}_j^{(t-1)}, \mathbf{e}_{j,i})\big\}_{j \in \mathcal{N}(i)}\Big), \tag{1}$$

where $\mathcal{N}(i)$ is the set of nodes connected to node $i$, $\mathbf{AGG}$ denotes a differentiable and permutation invariant function, e.g., sum, mean, or max, and $\Psi^{(t)}$ and $\Phi^{(t)}$ denote differentiable functions such as MLPs (Multi-Layer Perceptrons) [Fey and Lenssen, 2019].

MPNNs have shown state-of-the-art performance in many graph machine learning tasks such as node or graph classification. As such, MPNNs had a tremendous impact to the applied sciences, with promising achievements such as discovering a new class of antibiotics [Stokes et al., 2020], and has impacted the industry with applications in social media, recommendation systems, and 3D reconstruction, among others (see, e.g., [Ying et al., 2018, Wang et al., 2018a,b, Monti et al., 2019, Fan et al., 2019]). The practical success of MPNNs led to a significant boost in research aimed at understanding the theoretical properties of MPNNs. See, e.g., the variational inference point of view of MPNNs [Dai et al., 2016], and algorithmic alignment of MPNNs with combinatorial algorithms [Xu et al., 2019, Morris et al., 2019].

In this paper we study the generalization capabilities of MPNNs with mean aggregation in a graph classification task. Previous works developed generalization bounds that do not depend on any model of the data, namely, graphs in these works can be generated and labeled in any arbitrary way [Scarselli et al., 2018, Garg et al., 2020, Liao et al., 2021]. In this work, we consider a generative model for the graphs which is theoretically powerful and general on the one hand, and allows much tighter generalization bounds on the other hand.

Formally, we are given pairs of graphs and graph signals $\mathbf{x} = (G, \mathbf{f})$ and a target output $\mathbf{y}$, where $(\mathbf{x}, \mathbf{y})$ are jointly drawn from a distribution $\mu_{\mathcal{G}}(\mathbf{x}, \mathbf{y})$. The goal is to learn a MPNN $\Theta$ that approximates $\mathbf{y}$ by $\Theta(\mathbf{x})$. For this, one uses a loss function $\mathcal{L}$, which measures the discrepancy between the true label $\mathbf{y}$ and the output of the MPNN $\Theta(\mathbf{x})$. The aim of a machine learning algorithm is to minimize the statistical loss (also called expected loss)

$$R_{exp}(\Theta) = \mathbb{E}_{(\mathbf{x}, \mathbf{y}) \sim \mu_{\mathcal{G}}}\Big[\mathcal{L}(\Theta(\mathbf{x}), \mathbf{y})\Big].$$

In (data-driven) machine learning one has only access to a training set instead of knowing the distribution $\mu_{\mathcal{G}}$. Namely, we consider a multi-graph setting, where the training set $\mathcal{T} = (\mathbf{x}^i = (G^i, \mathbf{f}^i), \mathbf{y}^i)_{i=1}^m$ is a collection of $m$ samples drawn i.i.d. from the distribution $\mu_{\mathcal{G}}(\mathbf{x}, \mathbf{y})$. Then, instead of minimizing the statistical loss, one minimizes the empirical loss, given by

$$R_{\text{emp}}(\Theta) = \frac{1}{m} \sum_{i=1}^m \mathcal{L}(\Theta(\mathbf{x}^i), \mathbf{y}^i).$$

The optimized MPNN then depends on the dataset, and is hence denoted by $\Theta_{\mathcal{T}}$. The *generalization error* is defined to be

$$GE(\Theta_{\mathcal{T}}) = |R_{\text{exp}}(\Theta_{\mathcal{T}}) - R_{\text{emp}}(\Theta_{\mathcal{T}})|. \tag{2}$$

One then usually bounds (2) by the *uniform generalization error*

$$GE = \sup_{\Theta} |R_{\text{exp}}(\Theta) - R_{\text{emp}}(\Theta)|, \tag{3}$$

where the supremum is taken over some space of MPNNs. Bounds of $GE$ typically take the form $GE^2 \leq \frac{C}{m}q(N)$, where $C$ is a constant that describes the complexity of the model class (e.g., number of parameters), $m$ is the size of the training set, and $q(N)$ is a constant that depends on the (average) size of the graphs. For such bounds, see, e.g., VC-dimension based bounds [Scarselli et al., 2018], Rademacher complexity based bounds [Garg et al., 2020], and PAC-Bayesian based bounds [Liao et al., 2021].

While in previous bounds from the literature $q(N)$ either increases in $N$ or in the average degree, in this paper we develop a generalization bound that decays in the average number of nodes $N$. The idea is to treat the nodes of each graph as randomly sampled from some random graph model. In this point of view, not only the different graphs $\mathbf{x}^i$ are seen as random samples, but the union of all nodes of all graphs comprise together the random samples of the empirical loss. In the spirit of Monte Carlo theory, such a point of view should lead to a decay of the error between the empirical

and statistical losses as $N$ increases. As opposed to graphs, nodes cannot be seen as independent, due to the correlations entailed by the graph structure. Hence, our analysis focuses on developing Monte Carlo error bounds in a correlated nodes regime.

Since in our approach we model graphs as randomly sampled from underlying continuous models, we define the application of message passing neural networks, not only on graphs, but also on the underlying space from which graphs are sampled. We then formulate and prove the following convergence result, that we write here informally. Let $\mathbf{x} = (G, \mathbf{f})$ be drawn from the model $\chi$, then with high probability, we have for all MPNNs $\Theta$

$$\|\Theta(\mathbf{x}) - \Theta(\chi)\| = O(N^{-\alpha}),$$

where $N$ is the number of nodes in $\mathbf{x}$ and $\alpha > 0$. Based on this convergence result, we are able prove a generalization bound that decays in $N$.

## 1.1 Validity of the Proposed Model

The random graph models in our work are graphons [Lovász, 2012] with associated graphon signals (see Definition 2.3). The main assumption in our analysis is that graphs that are sampled from the same graphon belong to the same class. While this may seem like a limitation, it is actually a very mild and reasonable assumption. It is well known that equivalence classes of isomorphic graphs can be characterized by homomorphism densities [Lovász, 1967]. Namely, given two graphs $G_1, G_2$, if (and only if) for every simple graph $F$ the number of homomorphisms from $F$ to $G_1$ is equal to the number of homomorphisms from $F$ to $G_2$, then $G_1$ is isomorphic to $G_2$. Graphon analysis relaxes this observation to a continuous similarity measure. A sequence of graphs $\{G_j\}_{j \in \mathbb{N}}$ is said to converge in the graphon sense, if for every simple graph $F$ the homomorphism densities of $F$ in the graphs $\{G_j\}_{j \in \mathbb{N}}$ converge to some value. Graphs from such a sequence can be thought of as being similar in some sense which relaxes the combinatorial notion of graph isomorphism. Moreover, for each such converging sequence, there is a unique (up to some symmetry) limit object, called a graphon. This graphon is also seen as a generative model for graphs in the respective sequence, where graphs are generated by randomly sampling the graphon (see Definition 2.3). Now, since it is well known that MPNNs cannot distinguish between isomorphic graphs, it is also unreasonable to expect them to separate two graphs that are sampled from the same graphon. We hence assume that two graphs that are sampled from the same graphon belong to the same class (but not necessarily vice versa). This assumption allows us to derive a generalization bound that is much tighter than previously proposed bounds (see Figure 1 for comparison).

From a practical stance, our graphon assumption is reasonable since many graph models are special cases of graphons, like Erdős–Rényi, stochastic block model, and random geometric graphs [Penrose, 2003]. Moreover, the decoder of a graph variational autoencoder [Kipf and Welling, 2016] can be seen as a graphon.

## 1.2 Related Work

In this subsection we briefly survey different approaches for studying the convergence and generalization capabilities of GCNNs that were introduced in previous contributions. We give a comparison with our results in Section 3.

Levie et al. [2021] introduce the notion of GCNN transferability – the ability to transfer a GCNN between different graphs, which is closely related to generalization. For example, Levie et al. [2019], Gama et al. [2020], Kenlay et al. [2021] show that the output of spectral-based GCNNs is linearly stable with respect to perturbations of the input graphs. Levie et al. [2021] prove that spectral-based methods are transferable under graphs and graph signals that are sampled from the same latent space. Keriven et al. [2020], Ruiz et al. [2021a,b], Maskey et al. [2021] show that spectral-based GCNNs are transferable under graphs that approximate the same limit object – the so called graphon. Cervino et al. [2021] show that gradients of spectral-based GCNNs are transferable under graphs that approximate the same graphon.

Scarselli et al. [2018] provide generalization bounds that are comparable to VC-dimension bounds known for CNNs. These bounds are improved by Garg et al. [2020], who provide the first data dependent generalization bounds for MPNNs with sum aggregation that are comparable to Rademacher bounds for recurrent neural networks. Liao et al. [2021] derive a generalization bound via a PAC-Bayesian approach that is governed by the maximum node degree and spectral norms of the weights.

Verma and Zhang [2019a] consider generalization abilities of single-layer spectral GCNNs for node-classification task and provide a generalization bound that is directly proportional to the largest eigenvalue of the graph Laplacian. Another work along these lines comes from Yehudai et al. [2021], who show that certain MPNNs (with sum aggregation) do not generalize from small to large graphs.

## 1.3 Main Contributions

We follow the route of Keriven et al. [2020] and consider graphs as discretizations of continuous spaces in our analysis, called random graph models (RGM, see Definition 2.3). We introduce a continuous version of message passing neural networks – the realization of MPNNs on random graph models, which we call cMPNNs. Such cMPNNs are seen as limit objects of graph MPNNs, when the number of graph nodes goes to infinity. We prove, up to our knowledge, the first convergence result of the graph MPNN to the corresponding cMPNN as the number of nodes increases, which is uniform in the choice of the MPNN.

For the generalization analysis, we assume that the data distribution $\mu_{\mathcal{G}}$ represents graphs which are randomly sampled from a collection of template RGMs, with a random number of nodes. Using our convergence results, we can then prove that the generalization error between the training set and the true distribution is small. Here, we give the following informal version of Theorem 3.3.

**Theorem 1.1** (Informal version of Theorem 3.3). *Consider a graph classification task with* $m$ *training samples* $\mathcal{T} = (\mathbf{x}^i = (G^i, \mathbf{f}^i), \mathbf{y}^i)_{i=1}^m$ *drawn i.i.d. from the data distribution* $\mu_{\mathcal{G}}(\mathbf{x}, \mathbf{y})$ *on a metric-measure space* $\chi$ *of dimension* $D_\chi$*. Suppose that the size* $N$ *of each graph in* $\mathcal{T}$ *is drawn from a distribution* $\nu$*. Then*

$$\mathbb{E}_{\mathcal{T} \sim \mu_{\mathcal{G}}^m} \left[ \sup_\Theta \left( R_{emp}(\Theta) - R_{exp}(\Theta) \right)^2 \right] \leq \frac{C}{m} \mathbb{E}_{N \sim \nu} \left[ N^{-\frac{1}{D_\chi + 1}} \right].$$

The constant $C$ represents the complexity of the hypothesis space of the network, via the Lipschitz constants of the message and update functions and the depth of the MPNNs.

Theorem 3.3 shows how we can use fewer graphs $m$ than model complexity $C$ when training MPNNs if the graphs are sufficiently large.

## 2 Preliminaries

A weighted graph $G = (V, \mathbf{W}, E)$ with $N$ nodes is a tuple, where $V = \{1, \ldots, N\}$ is the node set. The edge set is given by $E \subset V \times V$, where $(i, j) \in E$ if node $i$ and $j$ are connected by an edge. $\mathbf{W} = (w_{k,l})_{k,l}$ is the weight matrix, assigning the weight $w_{i,j}$ to the edge $(i, j) \in E$, and assigning zero if $(i, j)$ is not an edge. The degree $\mathrm{d}_i$ of a node $i$ is defined as $\mathrm{d}_i = \sum_{j=1}^N w_{i,j}$. If $G$ is a simple graph, i.e., a weighted graph with $\mathbf{W} \in \{0, 1\}^{N \times N}$, the degree $\mathrm{d}_i$ is the number of nodes connected to node $i$ by an edge. We define a *graph signal* $\mathbf{f} : V \to \mathbb{R}^F$ as a function that maps nodes to their features in $\mathbb{R}^F$, where $F \in \mathbb{N}$ is the feature dimension. The signal $\mathbf{f}$ can be represented by a matrix $\mathbf{f} = (\mathbf{f}_1, \ldots, \mathbf{f}_N) \in \mathbb{R}^{N \times F}$, where $\mathbf{f}_i \in \mathbb{R}^F$ is the feature at node $i$. We also call $\mathbf{f}$ a *(graph) feature map*.

For a random variable $Y$ distributed according to $\kappa$, and a function $F$ of $Y$, we denote by $\mathbb{E}_{Y \sim \kappa}[F(Y)]$ the expected value of $F(Y)$. Similarly, we denote by $\mathrm{Var}_{Y \sim \kappa}[F(Y)]$ the variance of $F(Y)$.

### 2.1 Message Passing Graph Neural Networks

*Message passing graph neural networks (gMPNNs)* are defined by realizing an architecture of a *message passing neural network (MPNN)* on a graph. MPNNs are defined independently of a particular graph.

**Definition 2.1.** *Let* $T \in \mathbb{N}$ *denote the number of layers. For* $t = 1, \ldots, T$*, let* $\Phi^{(t)} : \mathbb{R}^{2F_{t-1}} \to \mathbb{R}^{H_{t-1}}$ *and* $\Psi^{(t)} : \mathbb{R}^{F_{t-1} + H_{t-1}} \to \mathbb{R}^{F_t}$ *be functions that we call the* message *and* update *functions, where* $F_t \in \mathbb{N}$ *is called the feature dimension of layer* $t$*. The corresponding* message passing neural network (MPNN) $\Theta$ *is defined to be the sequence*

$$\Theta = ((\Phi^{(t)})_{t=1}^T, (\Psi^{(t)})_{t=1}^T).$$

The message and the update function in Definition 2.1 are often defined as multi-layer-perceptrons (MLPs). In a MPNNs, messages are sent between nodes and aggregated. An *aggregation scheme* is a permutation invariant function that takes the collection of features in the edges of each node and computes a new nodes feature. In this paper, we consider MPNNs with *mean aggregation*. Then, a gMPNN processes graph signals by realizing a MPNN on the graph as follows.

**Definition 2.2.** *Let $G = (V, \mathbf{W})$ be a weighted graph and $\Theta$ be a MPNN, as defined in Definition 2.1. For each $t \in \{1, \ldots, T\}$, we define the* gMPNN $\Theta_G^{(t)}$ *as the mapping that maps input graph signals $\mathbf{f} = \mathbf{f}^{(0)} \in \mathbb{R}^{N \times F_0}$ to the features in the $t$-th layer by*

$$\Theta_G^{(t)} : \mathbb{R}^{N \times F_0} \to \mathbb{R}^{N \times F_t}, \quad \mathbf{f} \mapsto \mathbf{f}^{(t)} = (\mathbf{f}_i^{(t)})_{i=1}^N,$$

*where $\mathbf{f}^{(t)} \in \mathbb{R}^{N \times F_t}$ are defined sequentially by*

$$\mathbf{m}_i^{(t)} := \frac{1}{\mathrm{d}_i} \sum_{j=1}^N w_{i,j} \Phi^{(t)}(\mathbf{f}_i^{(t-1)}, \mathbf{f}_j^{(t-1)})$$

$$\mathbf{f}_i^{(t)} := \Psi^{(t)}(\mathbf{f}_i^{(t-1)}, \mathbf{m}_i^{(t)}),$$

*for every $i \in V$. We call $\Theta_G := \Theta_G^{(T)}$ a message passing graph neural network (gMPNN).*

Given a MPNN $\Theta$ as defined in Definition 2.1, the output $\Theta_G(\mathbf{f}) \in \mathbb{R}^{N \times F_T}$ is a graph signal. In graph classification or regression, the network should output a single feature for the whole graph. Hence, the output of a gMPNN after *global pooling* is a single vector $\Theta_G^P(\mathbf{f}) \in \mathbb{R}^{F_T}$, defined by

$$\Theta_G^P(\mathbf{f}) = \frac{1}{N} \sum_{i=1}^N \Theta_G(\mathbf{f})_i.$$

For brevity, in this paper we typically do not distinguish between a MPNN and its realization on a graph.

## 2.2 Random Graph Models

Let $(\chi, d, \mu)$ be a metric-measure space, where $\chi$ is a set, $d$ is a metric and $\mu$ is a probability Borel measure.

A *kernel* (also called a *graphon*), is a measurable mapping $W : \chi \times \chi \to \mathbb{R}$. The points $x \in \chi$ of the metric space are seen as the nodes of a continuous model, and the kernel is seen as a continuous version of a weight matrix. Kernels are treated as generative graph models using the following definition.

**Definition 2.3.** *A* random graph model (RGM) *on $(\chi, d, \mu)$ is defined as a pair $(W, f)$ of a kernel $W : \chi \times \chi \to \mathbb{R}$ and a measurable function $f : \chi \to \mathbb{R}$ called a* metric-space signal. *We define a* random graph *with corresponding node features $(G, \mathbf{f})$ by sampling $N$ i.i.d. random points $X_1, \ldots, X_N$ from $\chi$, with probability density $\mu$, as the nodes of $G$. The weight matrix $\mathbf{W} = (w_{i,j})_{i,j}$ of $G$ is defined by $w_{i,j} = W(X_i, X_j)$ for $i, j = 1, \ldots, N$. The graph signal $\mathbf{f}$ is defined by $\mathbf{f}_i = f(X_i)$. We say that $(G, \mathbf{f})$ is* drawn *from $W$, and denote $(G, \mathbf{f}) \sim (W, f)$.*

## 2.3 Continuous Message Passing Neural Networks

Given a MPNN, we define *continuous message passing neural networks (cMPNNs)* that act on kernels and metric-space signals $f : \chi \to \mathbb{R}^F$, by replacing the graph node features and the aggregation scheme in (2.2) by continuous counterparts. Let $W$ be a kernel. We define the *kernel degree* of $W$ at $x \in \chi$ by

$$\mathrm{d}_W(x) = \int_\chi W(x, y) d\mu(y). \tag{4}$$

Consider a message signal $U : \chi \times \chi \to \mathbb{R}^H$, where $U(x, y)$ is interpreted as a message sent from the point $y$ to the point $x$ in $\chi$. We define the continuous mean aggregation of $U$ by

$$M_W(U)(x) = \int_\chi \frac{W(x, y)}{\mathrm{d}_W(x)} U(x, y) d\mu(y).$$

Given the messages $U(x, y) = \Phi(f(x), f(y))$, where $\Phi : \mathbb{R}^{2F} \to \mathbb{R}^H$, we have

$$M_W(U)(x) = M_W\Big(\Phi\big(f(\cdot), f(\cdot\cdot)\big)\Big)(x) = \int_\chi \frac{W(x, y)}{\mathrm{d}_W(x)} \Phi\big(f(x), f(y)\big) d\mu(y).$$

By abuse of notation, we often denote in short $\Phi(f, f) := \Phi\big(f(\cdot), f(\cdot\cdot)\big)$.

By replacing mean aggregation by continuous mean aggregation in Definition 2.2, the same message and update functions that define a graph MPNN can also process metric-space signals.

**Definition 2.4.** *Let $W$ be a kernel and $\Theta$ be a MPNN, as defined in Definition 2.1. For each $t \in \{1, \ldots, T\}$, we define $\Theta_W^{(t)}$ as the mapping that maps the input signal to the signal in the $t$-th layer by*

$$\Theta_W^{(t)} : L^2(\chi) \to L^2(\chi), \quad f \mapsto f^{(t)}, \tag{5}$$

*where $f^{(t)}$ are defined sequentially by*

$$
\begin{aligned}
g^{(t)}(x) &= M_W\Big(\Phi^{(t)}\big(f^{(t-1)}, f^{(t-1)}\big)\Big)(x) \\
f^{(t)}(x) &= \Psi^{(t)}\Big(f^{(t-1)}(x), g^{(t)}(x)\Big)
\end{aligned}
\tag{6}
$$

*and $f^{(0)} = f : \chi \to \mathbb{R}^{F_0}$ is the input metric-space signal. We call $\Theta_W := \Theta_W^{(T)}$ a continuous message passing neural network (cMPNN).*

As with graphs, the output of a cMPNN $\Theta_W$ on a metric-space signal $f : \chi \to \mathbb{R}^{F_0}$ is another metric-space signal $\Theta_W(f) : \chi \to \mathbb{R}^{F_T}$. The output of a cMPNN after *global pooling* is a single vector $\Theta_W^P(f) \in \mathbb{R}^{F_T}$, defined by $\Theta_W^P(\mathbf{f}) = \int_\chi \Theta_W(f)(x) d\mu(x)$.

## 2.4 Data Distribution for Graph Classification Tasks

In the following, we consider a training data $\mathcal{T} = \big(\mathbf{x}^i = (G^i, \mathbf{f}^i), \mathbf{y}^i\big)_{i=1}^m$ of graphs $G^i$, graph signals $\mathbf{f}^i$, and corresponding values $\mathbf{y}^i$ that can represent the classes of the graph-signal pairs. The training data is assumed to be drawn i.i.d. from a distribution $\mu_\mathcal{G}(\mathbf{x}, \mathbf{y})$ that we describe next.

In this paper, we focus on classification tasks. More precisely we have classes $j = 1, \ldots, \Gamma$, each represented by a RGM $(W^j, f^j)$ on a metric-measure space $(\chi^j, d^j, \mu^j)$. In fact, we suppose that each class corresponds to a set of metric spaces. For example, a graph representing a chair can be sampled from a template of either an office chair, a garden chair, a bar stool, etc., and each of these is represented by a metric space. For simplicity of the exposition, we however treat every template metric space as its own class. This does not affect our analysis.

The distribution $\mu_\mathcal{G}(\mathbf{x}, \mathbf{y})$ is defined via the following procedure of data sampling. For sampling one graph, first, choose a class with probability $\gamma_j$, i.e., for $(\mathbf{x}, \mathbf{y}) \sim \mu_\mathcal{G}$ and $j = 1, \ldots, \Gamma$, $\gamma_j = \mathbb{P}(\mathbf{y} = j)$. Independently of the choice of the class, choose the number of nodes $N \sim \nu$, where $\nu$ is a discrete distribution on $N \in \mathbb{N}$. After choosing a class $\mathbf{y} \in \{1, \ldots, \Gamma\}$ and the graph size $N$, a random graph $(G, \mathbf{f}) \sim (W^{\mathbf{y}}, f^{\mathbf{y}})$ with $N$ nodes is drawn from the space $\chi^{\mathbf{y}}$ with probability density of the nodes $(\mu^{\mathbf{y}})^N$.

The notation $\mathcal{T} \sim \mu_\mathcal{G}^m$ describes a dataset $\mathcal{T}$ consisting of $m$ samples $(\mathbf{x}^1, \mathbf{y}^1), \ldots, (\mathbf{x}^m, \mathbf{y}^m)$ drawn i.i.d. from $\mu_\mathcal{G}$. We refer to Subsection C.1 in the appendix for a detailed definition of the distribution $\mu_\mathcal{G}$.

# 3 Convergence and Generalization of MPNNs

In this section, we provide our main results on convergence (Subsection 3.1) and generalization (Subsection 3.2) of MPNNs. For $z \in \mathbb{R}^F$, we define $\|z\|_\infty = \max_{j=1,\ldots,F} |z_j|$. Given a metric space $(\mathcal{Y}, d_\mathcal{Y})$, we define the infinity norm of a vector valued function $g : \mathcal{Y} \to \mathbb{R}^F$ by $\|g\|_\infty = \max_{j=1,\ldots,F} \operatorname{ess\,sup}_{y \in \mathcal{Y}} |(g(y))_j|$. The function $g$ is called *Lipschitz continuous* if there exists a constant $L_g \in \mathbb{R}$ such that for all $y, y' \in \mathcal{Y}$,

$$\|g(y) - g(y')\|_\infty \le L_g d_\mathcal{Y}(y, y').$$

If the domain $\mathcal{Y}$ is Euclidean, we always endow it with the $L^\infty$-metric.

We measure the error between the output of a continuous MPNN and a gMPNN after pooling as follows. Given a graph signal $\mathbf{f} \in \mathbb{R}^{N \times F}$ and a metric-space signal $f : \chi \to \mathbb{R}^F$, both the graph and the continuous MPNN map to the same output space, i.e, $\Theta_W^P(f), \Theta_G^P(\mathbf{f}) \in \mathbb{R}^{F_T}$. Namely, the output dimension of $\Theta^P$ is independent of the random graph model it is realized on and also independent of the graph. Hence, we define the error to be the supremum norm $\|\Theta_W^P(f) - \Theta_G^P(\mathbf{f})\|_\infty$. We define the $\varepsilon$-covering numbers of the metric space $\chi$, denoted by $\mathcal{C}(\chi, \varepsilon, d)$, as the minimal number of balls of radius $\varepsilon$ required to cover $\chi$.

For every $j = 1, \ldots, \Gamma$, we make the following assumptions, which hold for the remainder of the paper. We assume that there exist constants $C_{\chi^j}, D_{\chi^j} > 0$ such that

$$\mathcal{C}(\chi^j, \varepsilon, d) \leq C_{\chi^j}\, \varepsilon^{-D_{\chi^j}} \tag{7}$$

for every $\varepsilon > 0$. Denote $D_\chi = \max_j D_{\chi^j}$ and $C_\chi = \max_j C_{\chi^j}$. Such constants exist for every metric space with finite Minkowski dimension (see Appendix A). We assume that $\mathrm{diam}(\chi^j) := \sup_{x,y \in \chi^j}\{d(x,y)\} \leq 1$. Further, we only consider kernels $W^j$ such that there exists a constant $\mathrm{d}_{\min} > 0$ satisfying

$$\mathrm{d}_{W^j}(x) \geq \mathrm{d}_{\min}, \tag{8}$$

where the kernel degree $\mathrm{d}_{W^j}$ is defined in (4). We moreover assume that $W^j(x, \cdot)$ and $W^j(\cdot, x)$ are Lipschitz continuous (with respect to its second and first variable, respectively) with Lipschitz constant $L_{W^j}$ for every $x \in \chi$. We also assume that the metric-space signal $f^j : \chi \to \mathbb{R}^F$ is Lipschitz continuous. Since the diameter of $\chi^j$ is finite, this means that $f^j \in L^\infty(\chi)$. We consider the following class of MPNNs

$$\mathrm{Lip}_{L,B} :=$$
$$\left\{ \Theta = \big( (\Phi^{(l)})_{l=1}^T, (\Psi^{(l)})_{l=1}^T \big) \,\Big|\, \forall l = 1, \ldots, T, \ \ \Phi^{(l)} : \mathbb{R}^{F_l} \to \mathbb{R}^{H_l} \text{ and } \Psi^{(l)} : \mathbb{R}^{F_l + H_l} \to \mathbb{R}^{F_{l+1}} \right.$$
$$\left. \text{satisfy } L_{\Phi^{(l)}}, L_{\Psi^{(l)}} \leq L \text{ and } \|\Phi^{(l)}(0,0)\|_\infty, \|\Psi^{(l)}(0,0)\|_\infty \leq B \right\}.$$

### 3.1 Convergence

In this subsection we show that the error between the cMPNN and the according gMPNN decays when the number of nodes increases.

**Theorem 3.1.** *Let $W : \chi^2 \to \mathbb{R}$ be a Lipschitz continuous kernel with Lipschitz constant $L_W$, where the metric space $\chi$ satisfies (7) with respect to the constants $C_\chi, D_\chi > 0$, and $W$ satisfies (8). Consider a graph $(G, \mathbf{f}) \sim (W, f)$ with $N$ nodes $X_1, \ldots, X_N$ drawn i.i.d. from $\chi$ with probability density $\mu$. Then, for every Lipschitz continuous $f : \chi \to \mathbb{R}^F$,*

$$\mathbb{E}_{X_1,\ldots,X_N \sim \mu^N} \left[ \sup_{\Theta \in \mathrm{Lip}_{L,B}} \big\| \Theta_G^P(\mathbf{f}) - \Theta_W^P(f) \big\|_\infty^2 \right] \leq C' \big( 1 + \|f\|_\infty^2 + L_f^2 \big) \frac{\log(N)}{N^{1/(D_\chi+1)}} + \mathcal{O}(N^{-1}),$$

*where $C'$ is defined in Subsection B.2 of the appendix.*

**Remark 3.2.** *The constant $C'$ in Theorem 3.1 depends polynomially on the Lipschitz constants $L_{\Phi^{(l)}}$ and $L_{\Psi^{(l)}}$ of the message and update functions $\Phi^{(l)}$ and $\Psi^{(l)}$, on the so called formal biases $\|\Phi^{(l)}(0,0)\|_\infty$ and $\|\Psi^{(l)}(0,0)\|_\infty$, on $\|W\|_\infty$, on the Lipschitz constant $L_W$ of $W$, on $\sqrt{\log(C_\chi)} + \sqrt{D_\chi}$, and on $\frac{1}{\mathrm{d}_{\min}}$, where the degree of the polynomial is $T$. A regularization of these constants can alleviate the exponential dependency of the bound on $T$.*

The proof of Theorem 3.1 is given in Subsection B.2 of the appendix.

**Discussion and Comparison to other Convergence Results** The work closest related to our convergence results is by Keriven et al. [2020], where the authors show convergence of a fixed spectral GCNN to its continuous counterpart with comparable regularity assumptions as in Theorem 3.1. Our result holds for MPNNs, which are more general than spectral GCNNs. Moreover, our bound is uniform in the choice of the MPNN $\Theta$. This last property is essential for leveraging the convergence result to derive a generalization bound. Indeed, using the bound from Keriven et al.

[2020], for each MPNN $\Theta$ there is a different high probability event $\mathcal{E}_\Theta$ where the convergence error is small. However, the trained MPNN $\Theta = \Theta_\mathcal{T}$ depends on the dataset $\mathcal{T}$ and cannot be fixed in the analysis. Hence, we would need to intersect all events $\bigcap_\Theta \mathcal{E}_\Theta$ to guarantee a small convergence error of the trained network $\Theta_\mathcal{T}$, which would not result in an event of high probability.

## 3.2 Generalization

In this subsection, we state the main result of our paper, which provides a non-asymptotic bound on the generalization error of MPNNs, as defined in (3). We consider a graph classification task with a training set $\mathcal{T} = (\mathbf{x}^i = (G^i, \mathbf{f}^i), \mathbf{y}^i)_{i=1}^m$ and $\Gamma$ classes. The graphs and graph features in $\mathcal{T}$ are drawn i.i.d. from a probability distribution $\mu_\mathcal{G}(\mathbf{x}, \mathbf{y})$ as described in Subsection 2.4. We recall that the distribution that samples the size of the graph is denote by $\nu$.

Given a MPNN with pooling, $\Theta^P$, and its output dimension $\mathbb{R}^{F_T}$, we consider a non-negative loss function $\mathcal{L} : \mathbb{R}^{F_T} \times \{1, \dots, \Gamma\} \to [0, \infty)$. Additionally, we assume that $\mathcal{L}$ is Lipschitz continuous with Lipschitz constant $L_\mathcal{L}$. Note that although the cross-entropy loss, a popular choice for loss function in classification tasks, is not Lipschitz-continuous, cross-entropy composed on softmax is.

**Theorem 3.3.** *There exists a constant $C > 0$ such that*

$$\mathbb{E}_{\mathcal{T} \sim p^m} \left[ \sup_{\Theta \in \mathrm{Lip}_{L,B}} \left( R_{emp}(\Theta^P) - R_{exp}(\Theta^P) \right)^2 \right] \leq \frac{2^\Gamma 8 \|\mathcal{L}\|_\infty^2 \pi}{m}$$
$$+ \frac{2^\Gamma L_\mathcal{L}^2 C}{m} \sum_j \gamma_j \left( 1 + \|f^j\|_\infty^2 + L_{f^j}^2 \right) \cdot \left( \mathbb{E}_{N \sim \nu} \left[ \frac{1}{N} + \frac{1 + \log(N)}{N^{1/(D_{\chi^j}+1)}} + \mathcal{O}\left( \exp(-N) N^{3\frac{T-1}{2}} \right) \right] \right),$$

*where $C$ is specified in Subsection C.2 of the appendix.*

The proof of Theorem 3.3 is given in Subsection C.2 of the appendix.

**Remark 3.4.** *The constant $C$ in Theorem 3.3 represents the complexity of the class $\mathrm{Lip}_{L,B}$ and can be bounded similarly to the constant $C'$ from Theorem 3.1, as described in Remark 3.2. We summarize its dependencies on the parameters of the MPNN and the RGM by $\sqrt{C} \lesssim BL^{2T} \frac{1}{d_{\min}^{T+1}} \max_{j=1,\dots,\Gamma} \left( \sqrt{\log(C_{\chi^j})} + \sqrt{D_{\chi^j}} \right) L_{W^j} \|W^j\|_\infty^T$ and refer to Subsection C.3 of the appendix for more details. Similarly to Remark 3.2 the exponential dependency of the constant $C$ in Theorem 3.3 on the depth $T$ and the polynomial dependency on the uniform Lipschitz bound $L$ can be alleviated by regularizing the latter. We also note that the exponential dependency on the number of classes $\Gamma$ in Theorem 3.3 can be eliminated by assuming that the data is representative, i.e., if the number of training samples that fall into class $j = 1, \dots, \Gamma$ is deterministically $\gamma_j m$.*

The term $\frac{2^\Gamma 8 \|\mathcal{L}\|_\infty^2 \pi}{m}$ in Theorem 3.3 does not depend on the model complexity and is typically much smaller than the second term. Hence, it does not affect bias–variance tradeoff considerations, and can be ignored in the situation where $m \gg C \mathbb{E}_{N \sim \nu}[\log(N) N^{-\frac{1}{D_\chi + 1}}] \gg 1$. Theorem 3.3 allows us to think not just about graphs as samples, but also about individual nodes as samples. However, nodes are correlated with their neighbors, and the higher the dimension $D_\chi$ is, the larger the neighborhoods are. This is why the dependency on the number of nodes is $N^{-\frac{1}{2(D_\chi + 1)}}$ and not $N^{-1/2}$. Still, this dependency of the bound on $N$ explains one way in which we train on less graphs than model complexity and still generalize well. Another insight is that the generalization bound becomes smaller the smaller the Lipschitz constants of the message and update functions (see Remark 3.4). This indicates that regularization methods like weight decay promote generalization.

**Comparison to other generalization bounds in graph classification**  We compare our generalization bound with other generalization bounds derived by bounding the VC-dimension [Scarselli et al., 2018], the Rademacher complexity [Garg et al., 2020], and using a PAC-Bayesian approach [Liao et al., 2021]. We do not compare with Verma and Zhang [2019b] since they derive generalization bounds for single-layered MPNNs in node-classification tasks. Hence, the role of depth is unexplored. Furthermore, their bound scales as $\mathcal{O}(\lambda_{\max}^{2T}/m)$, where $T$ is the number of SGD steps and $\lambda_{\max}$ is the largest eigenvalue of the graph Laplacian. Hence, the generalization bound can increase monotonically for increasing $T$ (see [Liao et al., 2021] for more details). We summarize the comparison in Table 1 and provide more details, specially on the comparability, in Subsection C.4 of the appendix.

Table 1: Comparison of generalization bounds for GNNs. We consider the following formula for a generic generalization bound: $GE \leq m^{-1/2}A(d,N)B(h)C(L,T) + Em^{-1/2}$, where $m$ is the samples size, $T$ is the depth, $L$ is the bound of the Lipschitz constants of the message and update functions, $h$ is the maximum hidden dimension, $d$ is the average node degree and $N$ is the graphs size and $E$ is a term that does not depend on the model complexity.

| | $A(d,N)$ | $B(h)$ | $C(L)$ |
|---|---|---|---|
| VC-Dimension [Scarselli et al., 2018] | $\mathcal{O}(\log(N)N)$ | $\mathcal{O}(h^4)$ | - |
| Rademacher Complexity [Garg et al., 2020] | $\mathcal{O}(d^{T-1}\sqrt{\log(d^{2T-3})})$ | $\mathcal{O}(h\sqrt{\log(h)})$ | $\mathcal{O}(L^{2T})$ |
| PAC-Bayesian [Liao et al., 2021] | $\mathcal{O}(d^{T-1})$ | $\mathcal{O}(\sqrt{h\log(h)})$ | $\mathcal{O}(L^{2T})$ |
| Ours | $\mathcal{O}(\mathbb{E}_{N\sim\nu}[\log(N)N^{-\frac{1}{2(D_\chi+1)}}])$ | $\mathcal{O}(1)$ | $\mathcal{O}(L^{2T})$ |

Our analysis derives a generalization bound on MPNNs that has essentially the same dependency on the sample size $m$ (up to a logarithmic factor), but does not directly depend on the number of hidden units. We emphasize that our bound depends on negative moments of the expected node size $N$. In contrast, the VC-dimension based bound [Scarselli et al., 2018] scales as $\mathcal{O}(\log(N)N)$, the Rademacher complexity based bound [Garg et al., 2020] scales as $\mathcal{O}(d^{T-1}\sqrt{\log(d^{2T-3})})$, and the PAC-Bayesian approach based bound [Liao et al., 2021] scales as $\mathcal{O}(d^{T-1})$, where $d$ denotes the maximum node degree.

## 4    Numerical Experiments

We give empirical evaluations of our generalization bound in comparison to the PAC-Bayesian based bound [Liao et al., 2021] and the Rademacher complexity based bound [Garg et al., 2020]. We note that the VC dimension bound by Scarselli et al. [2018] is written in O notations and hence cannot be quantitatively evaluated. We experiment on a synthetic dataset of 100K random graphs of 50 nodes, sampled from three different RGMs: the Erdös-Rényi model (ERM) with edge probability $0.4$, a smooth version of a stochastic block model (SBM), based on the kernel $K(x,y) = \sin(2\pi x)\sin(2\pi y)/2\pi + 0.25$ on $[0,1]^2$, and a geometric graph with kernel $K(x,y) = \exp(-|x-y|^2)$. The corresponding signals are given in Appendix D.2.1. Each RGM represents one class in three binary classification problems, comparing all pairs of RGMs. For the MPNN we consider GraphSAGE [Hamilton et al., 2017] with mean aggregation, and number of layers $T = 1, 2$ or 3, implemented using Pytorch Geometric [Fey and Lenssen, 2019]. We consider a maximal hidden dimension of 128. In Appendix D we give more details and also consider synthetic data sampled from additional RGMs.

Our generalization bound becomes smaller the smaller the Lipschitz constants of the message and update functions are. To control the Lipschitz constants, we consider two learning settings. First, we train with weight decay regularization, which decreases the Lipschitz bounds, and second, we train with no regularization. For each setting (each choice of the number of layers and regularization) we train the MPNN, and read the resulting Lipschitz constants of the network. We then plug all constants into our generalization bound formula (see Theorem C.7 in the appendix for the full formula), and into the generalization bound formulas of the PAC-Bayes and Rademacher bounds (see Appendix C.4 for the formulas). The results are reported in Figure 1. We observe that our generalization bounds are orders of magnitude smaller than the other works. In fact, theoretical generalization bounds typically teach us about the asymptotic behavior of generalization, and about the hyperparameters that affect generalization, but rarely give realistic numerical bounds (less than 1) that guarantee generalization. Nevertheless, in one of the scenarios (one layer MPNN) our theory gives the bounds 0.08837 and 0.1325 (respectively in the two datasets of Figure 1), which guarantees generalization in practice.

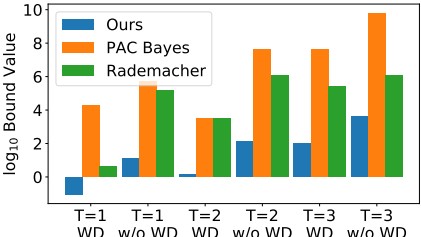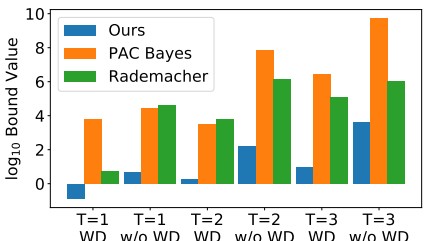

Figure 1: Generalization bounds given by our theory, PAC-Bayes [Liao et al., 2021] and Rademacher complexity [Garg et al., 2020] on a binary classification problem over Erdös-Rényi and SBM (left) and Erdös-Rényi and a geometric graph (right). Training is done with weight decay (WD) and without weight decay (w/o WD), and on three models with $T = 1, 2$ and 3 layers.

## 5  Conclusion

In this paper we proved that MPNNs with mean aggregation generalize from training to test data in classification tasks, if the graphs are sampled from RGMs that represent the different classes. This follows from the fact that the MPNN on sampled graphs converges to the MPNN on the RGM when the number of nodes goes to infinity. Our generalization bounds become smaller the larger the graphs, which gives one explanation to how MPNNs with high complexity can generalize well from a relatively small dataset of large graphs. We observe two main limitations of our current model. First, the dependency of the generalization bound on the size of the graph $N$ is $\mathcal{O}(N^{-\frac{1}{2(D_{\mathcal{X}}+1)}})$, which is typically slower than the observed decay in experiments (See Appendix D.1). One potential future direction is to improve this dependency using a more sophisticated models of the trained network and of the message and update functions. Secondly, our model of the data is somewhat limited. One future direction is to allow deformations of the RGMs, to consider a continuum of RGMs instead of a finite set, and to consider sparse graphs.

## Acknowledgments

S.M acknowledges partial support by the NSF–Simons Research Collaboration on the Mathematical and Scientific Foundations of Deep Learning (MoDL) (NSF DMS 2031985) and DFG SPP 1798, KU 1446/27-2.

Y.L. acknowledges support by DFG-SPP-2298, KU 1446/32-1.

G.K. acknowledges support from the ONE Munich Strategy Forum (LMU Munich, TU Munich, and the Bavarian Ministery for Science and Art), the Konrad Zuse School of Excellence in Reliable AI (DAAD), the Munich Center for Machine Learning (BMBF) as well as the German Research Foundation under Grants DFG-SPP-2298, KU 1446/31-1 and KU 1446/32-1 and under Grant DFG-SFB/TR 109, Project C09 and the Federal Ministry of Education and Research under Grant MaGriDo.

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
