## Appendix

In Appendix A, we introduce notations that we use throughout the rest of the appendix. In Appendix B, we study the convergence of MPNNs and give the proof of Theorem 3.1. In Appendix C, we analyze generalization properties of MPNNs and prove our main contribution, Theorem 3.3 from Section 3. We give some details on the numerical experiments from Section 4 in Appendix D. For completeness, we recall in Appendix E well-known results that we frequently use.

## A   Definitions and Notation

We denote metric spaces by $(\chi, d)$, where $d : \chi \times \chi \to [0, \infty)$ denotes the metric in the space $\chi$. The ball around $x \in \chi$ of radius $\epsilon > 0$ is defined to be $B_\epsilon(x) = \{y \in \chi \mid d(x, y) < \epsilon\}$. Since, in our analysis, the nodes of the graph are taken as the sample points $X = (X_1, \ldots, X_N)$ in $\chi$, we identify node $i$ of the graph $G$ with the point $X_i$, for every $i = 1, \ldots, N$. Moreover, since graph signals $\mathbf{f} = (\mathbf{f}_1, \ldots, \mathbf{f}_N)$ represent mappings from nodes in $V$ to feature values, we denote, by abuse of notation, $\mathbf{f}(X_i) := \mathbf{f}_i$ for $i = 1, \ldots, N$.

**Definition A.1** ([Vershynin, 2018]). *Let $(\chi, d)$ be a compact metric space.*

1. *The $\varepsilon$-covering numbers of $\chi$, denoted by $\mathcal{C}(\chi, \varepsilon, d)$, is the minimal number of balls of radius $\varepsilon$ required to cover $\chi$.*

2. *The* Minkowski dimension *of $\chi$ is defined to be*

$$\dim(\chi) = \inf\{D \geq 0 \mid \forall \varepsilon \in (0, 1) \, \mathcal{C}(\chi, \varepsilon, d) \leq \varepsilon^{-D}\}.$$

Next, we define various notions of degree.

**Definition A.2.** *Let $W : \chi \times \chi \to [0, \infty)$ be a kernel , $X = (X_1, \ldots, X_N)$ sample points, and $G$ the corresponding sampled graph.*

1. *We define the* kernel degree *of $W$ at $x \in \chi$ by*

$$\mathrm{d}_W(x) = \int_\chi W(x, y) d\mu(y). \tag{9}$$

2. *Given a point $x \in \chi$ that need not be in $X$, we define the* graph-kernel degree *of $X$ at $x$ by*

$$\mathrm{d}_X(x) = \frac{1}{N} \sum_{i=1}^N W(x, X_i). \tag{10}$$

3. *The* normalized degree *of $G$ at the node $X_c \in X$ is defined by*

$$\mathrm{d}_G(X_c) = \frac{1}{N} \sum_{i=1}^N W(X_c, X_i). \tag{11}$$

When $x \notin X$, $d_X(x)$ is interpreted as the degree of the node $x$ in the graph $(x, X_1, \ldots, X_n)$ with edge weights sampled from $W$.

Based on the different version of degrees in Definition A.2, we define the corresponding three versions of mean aggregation.

**Definition A.3.** *Given the kernel $W$, we define the* continuous mean aggregation *of the metric-space message signal $U : \chi \times \chi \to \mathbb{R}^F$ by*

$$M_W U = \int_\chi \frac{W(\cdot, y)}{\mathrm{d}_W(\cdot)} U(\cdot, y) d\mu(y).$$

In Definition A.3, $U(x, y)$ represents a message sent from the point $y$ to the point $x$ in the metric space. Given a metric-space signal $f : \chi \to \mathbb{R}^{F'}$ and a message function $\Phi$, we have

$$M_W \Phi(f, f) = \int_\chi \frac{W(\cdot, y)}{\mathrm{d}_W(\cdot)} \Phi\big(f(\cdot), f(y)\big) d\mu(y).$$

**Definition A.4.** *Let $W$ be a kernel $X = X_1, \ldots, X_N$ sample points. For a metric-space message signal $U : \chi \times \chi \to \mathbb{R}^F$, we define the* graph-kernel mean aggregation *by*

$$M_X U = \frac{1}{N} \sum_j \frac{W(\cdot, X_j)}{\mathrm{d}_X(\cdot)} U(\cdot, X_j).$$

Note that in the definition of $M_X$, messages are sent from graph nodes to arbitrary points in the metric space. Hence, $M_X U : \chi \to \mathbb{R}^F$ is a metric-space signal.

**Definition A.5.** *Let $G$ be a graph with nodes $X = X_1, \ldots, X_N$. For a graph message signal $\mathbf{U} : X \times X \to \mathbb{R}^F$, where $\mathbf{U}(X_i, X_j)$ represents a message sent from the node $X_j$ to the node $X_i$, we define the* mean aggregation *by*

$$(M_G \mathbf{U})(X_i) = \frac{1}{N} \sum_j \frac{W(X_i, X_j)}{\mathrm{d}_X(X_i)} \mathbf{U}(X_i, X_j).$$

Note that $M_G \mathbf{U} : X \to \mathbb{R}^F$ is a graph signal.

**Remark A.6.** *Given a graph signal $\mathbf{f} : X \to \mathbb{R}^F$, which can be written as a finite sequence $\mathbf{f} = (\mathbf{f}_i)_i$, and a message function $\Phi : \mathbb{R}^{2F} \to \mathbb{R}^H$, we define*

$$\Phi(\mathbf{f}, \mathbf{f}) := \big(\Phi(\mathbf{f}_i, \mathbf{f}_j)\big)_{i,j=1}^N.$$

Hence, given a graph signal $\mathbf{f} : X \to \mathbb{R}^F$ and the graph messages $\mathbf{U}(X_i, X_j) = \Phi(\mathbf{f}(X_i), \mathbf{f}(X_j))$, we have

$$M_G \mathbf{U} = M_G \Phi(\mathbf{f}, \mathbf{f}) = \frac{1}{N} \sum_j \frac{W(\cdot, X_j)}{\mathrm{d}_X(\cdot)} \Phi\big(\mathbf{f}(\cdot), \mathbf{f}(X_j)\big).$$

Next, we define the different norms used in our analysis.

**Definition A.7.**

1. *For a vector $\mathbf{z} = (z_1, \ldots, z_F) \in \mathbb{R}^F$, we define as usual*

$$\|\mathbf{z}\|_\infty = \max_{1 \le k \le F} |z_k|.$$

2. *For a function $g : \chi \to \mathbb{R}^F$, we define*

$$\|g\|_\infty = \max_{1 \le k \le F} \sup_{x \in \chi} \big|\big(g(x)\big)_k\big|,$$

3. *Given a graph with $N$ nodes, we define the norm $\|\mathbf{f}\|_{2;\infty}$ of graph feature maps $\mathbf{f} = (\mathbf{f}_1, \ldots, \mathbf{f}_N) \in \mathbb{R}^{N \times F}$, with feature dimension $F$, as the root mean square over the infinity norms of the node features, i.e.,*

$$\|\mathbf{f}\|_{2;\infty} = \sqrt{\frac{1}{N} \sum_{i=1}^N \|\mathbf{f}_i\|_\infty^2}.$$

**Definition A.8.** *For a metric-space signal $f : \chi \to \mathbb{R}^F$ and samples $X = (X_1, \ldots, X_N)$ in $\chi$, we define the sampling operator $S^X$ by*

$$S^X f = \big(f(X_i)\big)_{i=1}^N \in \mathbb{R}^{N \times F}.$$

For a metric-space signal $f : \chi \to \mathbb{R}^F$ and a graph signal $\mathbf{f} \in \mathbb{R}^{N \times F}$, we define the distance $\mathrm{dist}$ as $\mathrm{dist}(\mathbf{f}, f) = \|\mathbf{f} - S^X f\|_{2;\infty}$., i.e,

$$\mathrm{dist}(f, \mathbf{f}) = \left( \frac{1}{N} \sum_{i=1}^N \|\mathbf{f}_i - (S^X f)_i\|_\infty^2 \right)^{1/2}. \tag{12}$$

Given a MPNN, we define the *formal bias* of the update and message functions by $\|\Psi^{(l)}(0,0)\|_\infty$ and $\|\Phi^{(l)}(0,0)\|_\infty$ respectively. Furthermore, we say that a function $\Phi : \mathbb{R}^F \to \mathbb{R}^H$ is *Lipschitz continuous* if there exists a $L_\Phi > 0$ such that for every $x, x' \in \mathbb{R}^H$, we have

$$\|\Phi(x) - \Phi(x')\|_\infty \le L_\Phi \|x - x'\|_\infty.$$

Similarly, a function $f : \chi \to \mathbb{R}^F$ is Lipschitz continuous if there exists a $L_f > 0$ such that for every $x, x' \in \chi$, we have

$$\|\Phi(x) - \Phi(x')\|_\infty \le L_f d(x, x').$$

Next we introduce notations for the mappings between consecutive layers of a MPNN.

**Definition A.9.** *Let $\Theta = ((\Phi^{(l)})_{l=1}^T, (\Psi^{(l)})_{l=1}^T)$ be a MPNN with $T$ layers and feature dimensions $(F_l)_{l=1}^T$. For $l = 1, \ldots, T$, we define the mapping from the $(l-1)$'th layer to the $l$'th layer of the gMPNN as*

$$\Lambda_{\Theta_G}^{(l)} : \mathbb{R}^{N \times F_{l-1}} \to \mathbb{R}^{N \times F_l}$$
$$\mathbf{f}^{(l-1)} \mapsto \mathbf{f}^{(l)}.$$

*Similarly, we define $\Lambda_{\Theta_W}^{(l)}$ as the mapping from the $(l-1)$'th layer to the $l$'th layer of the cMPNN $f^{(l-1)} \mapsto f^{(l)}$.*

Definition A.9 leads to the following,

$$\Theta_G^{(T)} = \Lambda_{\Theta_G}^{(T)} \circ \Lambda_{\Theta_G}^{(T-1)} \circ \ldots \circ \Lambda_{\Theta_G}^{(1)}$$

and

$$\Theta_W^{(T)} = \Lambda_{\Theta_W}^{(T)} \circ \Lambda_{\Theta_W}^{(T-1)} \circ \ldots \circ \Lambda_{\Theta_W}^{(1)}$$

Lastly, we formulate the following assumption on the space $\chi$, the kernel $W$, and the MPNN $\Theta$, to which we will refer often in Appendix B.

**Assumption A.10.** *Let $(\chi, d)$ be a metric space and $W : \chi \times \chi \to [0, \infty)$. Let $\Theta$ be a MPNN with message and update functions $\Phi^{(l)} : \mathbb{R}^{2F_l} \to \mathbb{R}^{H_l}$ and $\Psi^{(l)} : \mathbb{R}^{F_l + H_l} \to \mathbb{R}^{F_{l+1}}$, $l = 1, \ldots, T-1$.*

1. *The space $\chi$ is compact, and there exist $D_\chi, C_\chi \ge 0$ such that $\mathcal{C}(\chi, \varepsilon, d) \le C_\chi \varepsilon^{-D_\chi}$ for every $\varepsilon > 0$.* [5]

2. *The diameter of $\chi$ is bounded by 1. Namely, $\operatorname{diam}(\chi) := \sup_{x,y \in \chi} d(x, y) \le 1$.*

3. *The kernel satisfies $\|W\|_\infty < \infty$.*

4. *For every $y \in \chi$, the function $W(\cdot, y)$ is Lipschitz continuous (with respect to its first variable) with Lipschitz constant $L_W$.*

5. *For every $x \in \chi$, the function $W(x, \cdot)$ is Lipschitz continuous (with respect to its second variable) with Lipschitz constant $L_W$.*

6. *There exists a constant $\mathrm{d}_{\min} > 0$ such that for every $x \in \chi$, we have $d_W(x) \ge \mathrm{d}_{\min}$.*

7. *For every $l = 1, \ldots, T$, the message function $\Phi^{(l)}$ and update function $\Psi^{(l)}$ are Lipschitz continuous with Lipschitz constants $L_{\Phi^{(l)}}$ and $L_{\Psi^{(l)}}$ respectively.*

8. *There exists a constant $\mathrm{W}_{\mathrm{diag}} > 0$ such that for every $x \in \chi$, we have $W(x, x) \ge \mathrm{W}_{\mathrm{diag}} > 0$.*

# B   Convergence Analysis

In this section we provide the proofs for Theorem 3.1 from Section 3.

---

[5]The Minkowski dimension $\dim(\chi)$ is a lower bound for all such possible $D_\chi$.

## B.1 Preparation

This section is a preparation for the upcoming proof of Theorem 3.1 from Section 3. An important goal of this section is to formulate and prove Lemma B.5, which provides a uniform concentration of measure of the uniform error between the continuous mean aggregation $M_W$ and the graph-kernel mean aggregation $M_X$. We then show in Corollary B.6 that this uniform bound is preserved by application of an update function. We begin with the following concentration of error lemma which is a slight modification of [Keriven et al., 2020, Lemma 4], and can be derived directly from [Keriven et al., 2020, Lemma 4], by using the assumption $\mathcal{C}(\chi, \varepsilon, d) \leq C_\chi \varepsilon^{-D_\chi}$ instead of $\mathcal{C}(\chi, \varepsilon, d) \leq \varepsilon^{-\dim(\chi)}$.

**Lemma B.1** (Lemma 4, [Keriven et al., 2020].)**.** *Let $(\chi, d, \mu)$ be a metric-measure space and $W$ be a kernel s.t. Assumptions A.10.1-4. are satisfied. Consider a metric-space signal $f : \chi \to \mathbb{R}$ with $\|f\|_\infty < \infty$. Suppose that $X_1, \ldots, X_N$ are drawn i.i.d. from $\mu$ on $\chi$ and let $p \in (0, 1)$. Then, with probability at least $1 - p$, we have*

$$\left\| \frac{1}{N} \sum_{i=1}^N W(\cdot, X_i) f(X_i) - \int_\chi W(\cdot, x) f(x) d\mu(x) \right\|_\infty$$
$$\leq \frac{\|f\|_\infty \Big( \zeta L_W (\sqrt{\log(C_\chi)} + \sqrt{D_\chi}) + (\sqrt{2}\|W\|_\infty + \zeta L_W)\sqrt{\log 2/p} \Big)}{\sqrt{N}},$$

*where*

$$\zeta := \frac{2}{\sqrt{2}} e \Big( \frac{2}{\ln(2)} + 1 \Big) \frac{1}{\sqrt{\ln(2)}} C \tag{13}$$

*and $C$ is the universal constant from Dudley's inequality (see Theorem 8.1.6 [Vershynin, 2018]).*

As a consequence of Lemma B.1, we can derive a sufficient condition on the sample size $N$ which ensures that the graph-kernel degrees are uniformly bounded from below.

**Lemma B.2.** *Let $(\chi, d, \mu)$ be a metric-measure space and $W$ be a kernel s.t. Assumptions A.10.1-4. and A.10.6. are satisfied. Suppose that $X_1, \ldots, X_N$ are drawn i.i.d. from $\mu$ on $\chi$ and let $p \in (0, 1)$. Let*

$$\sqrt{N} \geq 2 \Big( \zeta \frac{L_W}{\mathrm{d}_{\min}} (\sqrt{\log(C_\chi)} + \sqrt{D_\chi}) + \frac{\sqrt{2}\|W\|_\infty + \zeta L_W}{\mathrm{d}_{\min}} \sqrt{\log 2/p} \Big), \tag{14}$$

*where $\zeta$ is defined in (13). Then, with probability at least $1 - p$ the following two inequalities hold: For every $x \in \chi$,*

$$\mathrm{d}_X(x) \geq \frac{\mathrm{d}_{\min}}{2} \tag{15}$$

*and*

$$\left\| \frac{1}{N} \sum_{i=1}^N W(\cdot, X_i) f(X_i) - \int_\chi W(\cdot, x) f(x) d\mu(x) \right\|_\infty$$
$$\leq \frac{\|f\|_\infty \Big( \zeta L_W (\sqrt{\log(C_\chi)} + \sqrt{D_\chi}) + (\sqrt{2}\|W\|_\infty + \zeta L_W)\sqrt{\log 2/p} \Big)}{\sqrt{N}}. \tag{16}$$

*Proof.* By Lemma B.1, with $f = 1$, with probability at least $1 - p$ we have

$$\|\mathrm{d}_X(\cdot) - \mathrm{d}_W(\cdot)\|_\infty \leq \frac{\Big( \zeta L_W (\sqrt{\log(C_\chi)} + \sqrt{D_\chi}) + (\sqrt{2}\|W\|_\infty + \zeta L_W)\sqrt{\log 2/p} \Big)}{\sqrt{N}}.$$

By using the lower bound (14) of $\sqrt{N}$, we have $\|\mathrm{d}_X(\cdot) - \mathrm{d}_W(\cdot)\|_\infty \leq \frac{\mathrm{d}_{\min}}{2}$. Let $x \in \chi$. By Assumption A.10.6, we have $|\mathrm{d}_W(x)| \geq \mathrm{d}_{\min}$, hence $|\mathrm{d}_X(x)| \geq \mathrm{d}_{\min}/2$. $\qquad\square$

The following lemma is a uniform concentration of measure of the Monte Carlo approximation of Lipschitz functions. Related results about uniform law of large numbers for Lipschitz functions can be found in [Vershynin, 2018, Chapter 8.2]. Our result holds for general metric spaces with finite Minkowski dimension.

**Lemma B.3.** *Let $(\chi, d, \mu)$ be a metric-measure space s.t. Assumption A.10.1. is satisfied. Suppose that $X_1, \ldots, X_N$ are drawn i.i.d. from $\mu$ on $\chi$. For every $p > 0$, there exists an event $\mathcal{E}_{\mathrm{Lip}}^p \subset \chi^N$ regarding the choice of $(X_1, \ldots, X_N) \in \chi^N$, with probability $\mu^N(\mathcal{E}_{\mathrm{Lip}}^p) \geq 1 - p$, such that the following uniform bound is satisfied: For every Lipschitz continuous function $F : \chi \to \mathbb{R}^F$ with Lipschitz constant $L_F$, we have*

$$\left\| \frac{1}{N} \sum_{i=1}^{N} F(X_i) - \int_{\chi} F(x) d\mu(x) \right\|_{\infty}$$

$$\leq N^{-\frac{1}{2(D_\chi + 1)}} \left( 2L_F + \frac{C_\chi}{\sqrt{2}} \|F\|_\infty \sqrt{\log(C_\chi) + \frac{D_\chi}{2(D_\chi + 1)} \log(N) + \log(2/p)} \right).$$

For completion, we provide a proof of Lemma B.3.

*Proof.* Let $r > 0$. By Assumption A.10.1, there exists an open covering $(B_j)_{j \in \mathcal{J}}$ of $\chi$ by a family of balls with radius $r$ such that $|\mathcal{J}| \leq C_\chi r^{-D_\chi}$. For $j = 2, \ldots, |\mathcal{J}|$, we define $I_j := B_j \setminus \cup_{i<j} B_i$, and define $I_1 = B_1$. Hence, $(I_j)_{j \in \mathcal{J}}$ is a family of measurable sets such that $I_j \cap I_i = \emptyset$ for all $i \neq j \in \mathcal{J}$, $\bigcup_{j \in \mathcal{J}} I_j = \chi$, and $\mathrm{diam}(I_j) \leq 2r$ for all $j \in \mathcal{J}$, where by convention $\mathrm{diam}(\emptyset) = 0$. For each $j \in \mathcal{J}$, let $z_j$ be the center of the ball $B_j$.

Next, we compute a concentration of error bound on the difference between the measure of $I_j$ and its Monte Carlo approximation, which is uniform in $j \in \mathcal{J}$. Let $j \in \mathcal{J}$ and $q \in (0, 1)$. By Hoeffding's inequality, there is an event $\mathcal{E}_j^q$ with probability $\mu(\mathcal{E}_j) \geq 1 - q$, in which

$$\left\| \frac{1}{N} \sum_{i=1}^{N} \mathbb{1}_{I_j}(X_i) - \mu(I_k) \right\|_{\infty} \leq \frac{1}{\sqrt{2}} \frac{\sqrt{\log(2/q)}}{\sqrt{N}}. \tag{17}$$

Consider the event

$$\mathcal{E}_{\mathrm{Lip}}^{|\mathcal{J}|q} = \bigcap_{j=1}^{|\mathcal{J}|} \mathcal{E}_j^q,$$

with probability $\mu^N(\mathcal{E}_{\mathrm{Lip}}^{|\mathcal{J}|q}) \geq 1 - |\mathcal{J}|q$. In this event, (17) holds for all $j \in \mathcal{J}$. We change the failure probability variable $p = |\mathcal{J}|q$, and denote $\mathcal{E}_{\mathrm{Lip}}^p = \mathcal{E}_{\mathrm{Lip}}^{|\mathcal{J}|q}$.

Next we bound uniformly the Monte Carlo approximation error of the integral of bounded Lipschitz continuous functions $F : \chi \to \mathbb{R}^F$. Let $F : \chi \to \mathbb{R}^F$ be a bounded Lipschitz continuous function with Lipschitz constant $L_F$. We define the step function

$$F^r(y) = \sum_{j \in \mathcal{J}} F(z_j) \mathbb{1}_{I_j}(y).$$

Then,

$$\left\| \frac{1}{N} \sum_{i=1}^{N} F(X_i) - \int_{\chi} F(y) d\mu(y) \right\|_{\infty} \leq \left\| \frac{1}{N} \sum_{i=1}^{N} F(X_i) - \frac{1}{N} \sum_{i=1}^{N} F^r(X_i) \right\|_{\infty}$$

$$+ \left\| \frac{1}{N} \sum_{i=1}^{N} F^r(X_i) - \int_{\chi} F^r(y) d\mu(y) \right\|_{\infty} \tag{18}$$

$$+ \left\| \int_{\chi} F^r(y) d\mu(y) - \int_{\chi} F(y) d\mu(y) \right\|_{\infty}$$

$$=: (1) + (2) + (3).$$

To bound (1), we define for each $X_i$ the unique index $j_i \in \mathcal{J}$ s.t. $X_i \in I_{j_i}$. We calculate,

$$\left\| \frac{1}{N} \sum_{i=1}^{N} F(X_i) - \frac{1}{N} \sum_{i=1}^{N} F^r(X_i) \right\|_{\infty} \leq \frac{1}{N} \sum_{i=1}^{N} \left\| F(X_i) - \sum_{j \in \mathcal{J}} F(z_j) \mathbb{1}_{I_j}(X_i) \right\|_{\infty}$$

$$= \frac{1}{N} \sum_{i=1}^{N} \left\| F(X_i) - F(z_{j_i}) \right\|_{\infty}$$

$$\leq r L_F.$$

We proceed by bounding (2). In the event of $\mathcal{E}_{\mathrm{Lip}}^p$, which holds with probability at least $1-p$, equation (17) holds for all $j \in \mathcal{J}$. In this event, we get

$$\left\| \frac{1}{N} \sum_{i=1}^{N} F^r(X_i) - \int_{\chi} F^r(y) d\mu(y) \right\|_{\infty} = \left\| \sum_{j \in \mathcal{J}} \left( \frac{1}{N} \sum_{i=1}^{N} F(z_j) \mathbb{1}_{I_j}(X_i) - \int_{I_j} F(z_j) dy \right) \right\|_{\infty}$$

$$\leq \sum_{j \in \mathcal{J}} \|F\|_{\infty} \left| \frac{1}{N} \sum_{i=1}^{N} \mathbb{1}_{I_j}(X_i) - \mu(I_j) \right|$$

$$\leq |\mathcal{J}| \|F\|_{\infty} \frac{1}{\sqrt{2}} \frac{\sqrt{\log(2|\mathcal{J}|/p)}}{\sqrt{N}}.$$

Recall that $|\mathcal{J}| \leq C_\chi r^{-D_\chi}$. Then, with probability at least $1-p$

$$\left\| \frac{1}{N} \sum_{i=1}^{N} F^r(X_i) - \int_{\chi} F^r(y) d\mu(y) \right\|_{\infty}$$

$$\leq C_\chi r^{-D_\chi} \|F\|_{\infty} \frac{1}{\sqrt{2}} \frac{\sqrt{\log(C_\chi) - D_\chi \log(r) + \log(2/p)}}{\sqrt{N}}.$$

To bound (3), we calculate

$$\left\| \int_{\chi} F^r(y) d\mu(y) - \int_{\chi} F(y) d\mu(y) \right\|_{\infty} = \left\| \int_{\chi} \sum_{j \in \mathcal{J}} F(z_j) \mathbb{1}_{I_j} d\mu(y) - \int_{\chi} F(y) d\mu(y) \right\|_{\infty}$$

$$\leq \sum_{j \in \mathcal{J}} \int_{I_j} \|F(z_j) - F(y)\|_{\infty} \, d\mu(y)$$

$$\leq r L_F.$$

By plugging the bounds of $(1), (2)$ and $(3)$ into (18), we get

$$\left\| \frac{1}{N} \sum_{i=1}^{N} F(X_i) - \int_{\chi} F(y) d\mu(y) \right\|_{\infty} \leq 2r L_F + C_\chi r^{-D_\chi} \|F\|_{\infty} \frac{1}{\sqrt{2}} \frac{\sqrt{\log(C_\chi) - D_\chi \log(r) + \log(2/p)}}{\sqrt{N}}.$$

Lastly, choosing $r = N^{-\frac{1}{2(D_\chi+1)}}$ gives us an overall error of

$$\left\| \frac{1}{N} \sum_{i=1}^{N} F(X_i) - \int_{\chi} F(y) d\mu(y) \right\|_{\infty}$$

$$\leq N^{-\frac{1}{2(D_\chi+1)}} \left( 2L_F + C_\chi \|F\|_{\infty} \frac{1}{\sqrt{2}} \sqrt{\log(C_\chi) + \frac{D_\chi}{2(D_\chi+1)} \log(N) + \log(2/p)} \right)$$

Since the event $\mathcal{E}_{\mathrm{Lip}}^p$ is independent of the choice of $F : \chi \to \mathbb{R}^F$, the proof is finished. $\square$

The next lemma is based on Lemma B.3, and provides a uniform concentration of measure on the $L^\infty$-error between a non-normalized version of the kernel mean aggregation from Definition A.3 and a non-normalized version of the graph-kernel mean aggregation from Definition A.4.

**Lemma B.4.** *Let $(\chi, d, \mu)$ be a metric-measure space and $W$ be a kernel s.t. Assumptions A.10.1-3 and A.10.5. are satisfied. Let $p \in (0, 1)$. Suppose that $X_1, \ldots, X_N$ are drawn i.i.d. from $\mu$ on $\chi$ such that $(X_1, \ldots, X_N) \in \mathcal{E}_{\text{Lip}}^p$, where the event $\mathcal{E}_{\text{Lip}}^p$ is defined in Lemma B.3. Then, for every $x \in \chi$, $f : \chi \to \mathbb{R}^F$ with Lipschitz constant $L_f$, and $\Phi : \mathbb{R}^{2F} \to \mathbb{R}^H$ with Lipschitz constant $L_\Phi$, we have*

$$\left\| \frac{1}{N} \sum_{i=1}^N W(x, X_i) \Phi\big(f(x), f(X_i)\big) - \int_\chi W(x, y) \Phi\big(f(x), f(y)\big) d\mu(y) \right\|_\infty$$

$$\leq N^{-\frac{1}{2(D_\chi + 1)}} \left( 2 \Big( \|W\|_\infty L_\Phi L_f + L_W \big( L_\Phi \|f\|_\infty + \|\Phi(0, 0)\|_\infty \big) \Big) \right.$$

$$\left. + C_\chi \Big( \|W\|_\infty \big( L_\Phi \|f\|_\infty + \|\Phi(0, 0)\|_\infty \big) \Big) \frac{1}{\sqrt{2}} \sqrt{\log(C_\chi) + \frac{D_\chi}{2(D_\chi + 1)} \log(N) + \log(2/p)} \right).$$

$$\tag{19}$$

*Proof.* For any $x \in \chi$, $f : \chi \to \mathbb{R}^F$ and $\Phi : \mathbb{R}^{2F} \to \mathbb{R}^H$, we define the random variable

$$Y_{x;\Phi} = \frac{1}{N} \sum_{i=1}^N W(x, X_i) \Phi\big(f(x), f(X_i)\big) - \int_\chi W(x, y) \Phi\big(f(x), f(y)\big) d\mu(y)$$

on the sample space $\chi^N$. Applying Lemma B.3 on the integrand $F_x(y) := W(x, y) \Phi\big(f(x), f(y)\big)$, uniformly on the choice of the parameter $x \in \chi$, yields in the event $\mathcal{E}_{\text{Lip}}^p$:

$$\|Y_{x;\Phi}\|_\infty \leq N^{-\frac{1}{2(D_\chi + 1)}} \left( 2 L_{F_x} + C_\chi \|F_x\|_\infty \frac{1}{\sqrt{2}} \sqrt{\log(C_\chi) + \frac{D_\chi}{2(D_\chi + 1)} \log(N) + \log(2/p)} \right).$$

$$\tag{20}$$

So it remains to calculate the Lipschitz constant and the infinity-norm of $F_x$. For this, calculate for $y, y' \in \chi$

$$\begin{aligned}
\|F_x(y) - F_x(y')\|_\infty &= \|W(x, y) \Phi\big(f(x), f(y)\big) - W(x, y') \Phi\big(f(x), f(y')\big)\|_\infty \\
&\leq \|W(x, y) \Phi\big(f(x), f(y)\big) - W(x, y) \Phi\big(f(x), f(y')\big)\|_\infty \\
&\quad + \|W(x, y) \Phi\big(f(x), f(y')\big) - W(x, y') \Phi\big(f(x), f(y')\big)\|_\infty \\
&\leq \big( \|W\|_\infty L_\Phi L_f + L_W (L_\Phi \|f\|_\infty + \|\Phi(0, 0)\|_\infty) \big) d(y, y')
\end{aligned}$$

and

$$\begin{aligned}
\|F_x(\cdot)\|_\infty &= \|W(x, \cdot) \Phi\big(f(x), f(\cdot)\big)\|_\infty \\
&\leq \|W\|_\infty (L_\Phi \|f\|_\infty + \|\Phi(0, 0)\|_\infty).
\end{aligned}$$

$\square$

The next lemma provides a uniform concentration of measure bound on the error between the graph-kernel mean aggregation $M_X$ and the continuous mean aggregation $M_W$.

**Lemma B.5.** *Let $(\chi, d, \mu)$ be a metric-measure space and $W$ be a kernel s.t. Assumptions A.10.1-6. are satisfied. Let $N \in \mathbb{N}$ satisfy (14). Let $\mathcal{E}_{\text{Lip}}^p$ be the event defined in Lemma B.3. There exists an event $\mathcal{F}_{\text{Lip}}^p \subset \mathcal{E}_{\text{Lip}}^p$ regarding the choice of i.i.d $X_1, \ldots, X_N$ from $\mu$ in $\chi$, with probability $\mu(\mathcal{F}_{\text{Lip}}^p) \geq 1 - 2p$, such that condition (15) together with (21) below are satisfied: for every $f : \chi \to \mathbb{R}^F$ with Lipschitz constant $L_f$ and $\Phi : \mathbb{R}^{2F} \to \mathbb{R}^H$ with Lipschitz constant $L_\Phi$*

$$\|(M_X - M_W)\big(\Phi(f, f)\big)\|_\infty \leq 4 \frac{\varepsilon_1}{\sqrt{N} \mathrm{d}_{min}^2} \|W\|_\infty (L_\Phi \|f\|_\infty + \|\Phi(0, 0)\|_\infty)$$

$$+ N^{-\frac{1}{2(D_\chi + 1)}} \left( 2 \Big( \frac{\|W\|_\infty}{\mathrm{d}_{min}} L_\Phi L_f + \frac{L_W}{\mathrm{d}_{min}} \big( L_\Phi \|f\|_\infty + \|\Phi(0, 0)\|_\infty \big) \Big) \right.$$

$$\left. + C_\chi \Big( \frac{\|W\|_\infty}{\mathrm{d}_{min}} \big( L_\Phi \|f\|_\infty + \|\Phi(0, 0)\|_\infty \big) \Big) \frac{1}{\sqrt{2}} \sqrt{\log(C_\chi) + \frac{D_\chi}{2(D_\chi + 1)} \log(N) + \log(2/p)} \right),$$

$$\tag{21}$$

*where*

$$\varepsilon_1 = L_W \left( \sqrt{\log(C_\chi)} + \sqrt{D_\chi} \right) + \left( \sqrt{2}\|W\|_\infty + L_W \right) \sqrt{\log 2/p}. \tag{22}$$

*Proof.* By Lemma B.2, we have with probability at least $1 - p$

$$
\|\mathrm{d}_X - \mathrm{d}_W\|_\infty \leq \frac{\varepsilon_1}{\sqrt{N}} = \zeta \frac{L_W \left( \sqrt{\log(C_\chi)} + \sqrt{D_\chi} \right) + \left( \sqrt{2}\|W\|_\infty + L_W \right) \sqrt{\log 2/p}}{\sqrt{N}}
$$
$$
\leq \frac{\mathrm{d}_{\min}}{2}, \tag{23}
$$

where the second inequality follows from (14). Furthermore, in the same event we have

$$|\mathrm{d}_X(x)|_\infty \geq \frac{\mathrm{d}_{\min}}{2}$$

for all $x \in \chi$. Moreover, $|\mathrm{d}_W(x)|_\infty \geq \mathrm{d}_{\min}$ by Assumption A.10.6. Hence, for all $x \in \chi$, we have

$$
\left| \frac{1}{\mathrm{d}_X(x)} - \frac{1}{\mathrm{d}_W(x)} \right| = \frac{|\mathrm{d}_W(x) - \mathrm{d}_X(x)|}{|\mathrm{d}_X(x)\mathrm{d}_W(x)|}
$$
$$
\leq 4 \frac{\varepsilon_1}{\sqrt{N}\mathrm{d}_{\min}^2}. \tag{24}
$$

Denote that intersection of $\mathcal{E}_{\mathrm{Lip}}^p$ and the event in which (23) occur by $\mathcal{F}_{\mathrm{Lip}}^p$. Let $(X_1, \ldots, X_N)$ be i.i.d samples in $\mathcal{F}_{\mathrm{Lip}}^p$. Define $\tilde{W}(x,y) = \frac{W(x,y)}{\mathrm{d}_W(x)}$. Next we apply Lemma B.4 on the kernel $\tilde{W}$. For this, note that for $x \in \chi$ the kernel $\tilde{W}(x, \cdot)$ is Lipschitz continuous (with respect to the second variable) with Lipschitz constant $L_{\tilde{W}} = \frac{L_W}{\mathrm{d}_{\min}}$, since for $y, y' \in \chi$, we have

$$
\left| \frac{W(x,y)}{\mathrm{d}_W(x)} - \frac{W(x,y')}{\mathrm{d}_W(x)} \right| \leq \frac{L_W}{\mathrm{d}_{min}} d(y, y').
$$

Moreover, for all $y \in \chi$ we have $\|\tilde{W}(\cdot, y)\|_\infty \leq \frac{\|W\|_\infty}{\mathrm{d}_{\min}}$.

Then, we use Lemma B.4 to obtain, for every $f : \chi \to \mathbb{R}^F$ and $\Phi : \mathbb{R}^{2F} \to \mathbb{R}^H$ as specified in the lemma,

$$
\left\| \frac{1}{N} \sum_{i=1}^N \tilde{W}(\cdot, X_i)\Phi\big(f(\cdot), f(X_i)\big) - \int_\chi \tilde{W}(\cdot, y)\Phi\big(f(\cdot), f(y)\big)d\mu(y) \right\|_\infty
$$
$$
\leq N^{-\frac{1}{2(D_\chi+1)}} \left( 2\Big( \|\tilde{W}\|_\infty L_\Phi L_f + L_{\tilde{W}}\big(L_\Phi\|f\|_\infty + \|\Phi(0,0)\|_\infty\big) \Big) \right.
$$
$$
\left. + C_\chi\Big( \|\tilde{W}\|_\infty \big(L_\Phi\|f\|_\infty + \|\Phi(0,0)\|_\infty\big) \Big) \frac{1}{\sqrt{2}} \sqrt{\log(C_\chi) + \frac{D_\chi}{2(D_\chi+1)}\log(N) + \log(2/p)} \right)
$$
$$
\leq N^{-\frac{1}{2(D_\chi+1)}} \left( 2\Big( \frac{\|W\|_\infty}{\mathrm{d}_{\min}} L_\Phi L_f + \frac{L_W}{\mathrm{d}_{\min}}\big(L_\Phi\|f\|_\infty + \|\Phi(0,0)\|_\infty\big) \Big) \right.
$$
$$
\left. + C_\chi\Big( \frac{\|W\|_\infty}{\mathrm{d}_{\min}} \big(L_\Phi\|f\|_\infty + \|\Phi(0,0)\|_\infty\big) \Big) \frac{1}{\sqrt{2}} \sqrt{\log(C_\chi) + \frac{D_\chi}{2(D_\chi+1)}\log(N) + \log(2/p)} \right). \tag{25}
$$

Then, by (24) and (25), for every $f : \chi \to \mathbb{R}^F$ and $\Phi : \mathbb{R}^{2F} \to \mathbb{R}^H$ as specified in the lemma,

$$\|(M_X - M_W)\Phi(f,f)\|_\infty$$

$$= \left\| \frac{1}{N} \sum_{i=1}^N \frac{W(\cdot, X_i)}{\mathrm{d}_X(\cdot)} \Phi\big(f(\cdot), f(X_i)\big) - \int_\chi \frac{W(\cdot, x)}{\mathrm{d}_W(\cdot)} \Phi\big(f(\cdot), f(x)\big) d\mu(x) \right\|_\infty$$

$$\leq \frac{1}{N} \sum_{i=1}^N \big\| W(x, X_i)\Phi\big(f(\cdot), f(X_i)\big) \big\|_\infty \left\| \frac{1}{\mathrm{d}_X(\cdot)} - \frac{1}{\mathrm{d}_W(\cdot)} \right\|_\infty$$

$$+ \left\| \frac{1}{N} \sum_{i=1}^N \tilde{W}(\cdot, X_i)\Phi\big(f(\cdot), f(X_i)\big) - \int_\chi \tilde{W}(\cdot, x)\Phi\big(f(\cdot), f(x)\big) d\mu(x) \right\|_\infty$$

$$\leq 4\frac{\varepsilon_1}{\sqrt{N}\mathrm{d}_{min}^2} \|W\|_\infty (L_\Phi\|f\|_\infty + \|\Phi(0,0)\|_\infty)$$

$$+ N^{-\frac{1}{2(D_\chi+1)}} \left( 2\Big( \frac{\|W\|_\infty}{\mathrm{d}_{min}} L_\Phi L_f + \frac{L_W}{\mathrm{d}_{min}} \big(L_\Phi\|f\|_\infty + \|\Phi(0,0)\|_\infty\big) \Big) \right.$$

$$+ C_\chi \Big( \frac{\|W\|_\infty}{\mathrm{d}_{min}} \big(L_\Phi\|f\|_\infty + \|\Phi(0,0)\|_\infty\big) \Big) \frac{1}{\sqrt{2}} \sqrt{\log(C_\chi) + \frac{D_\chi}{2(D_\chi+1)}\log(N) + \log(2/p)} \Bigg).$$

$\square$

The next corollary shows that Lemma B.5 is preserved by the application of an update function.

**Corollary B.6.** *Let $(\chi, d, \mu)$ be a metric-measure space and $W$ be a kernel s.t. Assumptions A.10.1-6. are satisfied. Let $p > 0$ and $N \in \mathbb{N}$ satisfy (14). Suppose that $X_1, \dots, X_N$ are drawn i.i.d. from $\mu$ on $\chi$. If the event $\mathcal{F}_{\mathrm{Lip}}^p$ from Lemma B.5 occurs, then condition (15) together with (26) below are satisfied: for every $f : \chi \to \mathbb{R}^F$ with Lipschitz constant $L_f$, $\Phi : \mathbb{R}^{2F} \to \mathbb{R}^H$ with Lipschitz constant $L_\Phi$ and $\Psi : \mathbb{R}^{F+H} \to \mathbb{R}^{F'}$ with Lipschitz constant $L_\Psi$*

$$\left\| \Psi\Big( f(\cdot), M_X\big(\Phi(f,f)\big)(\cdot)\Big) - \Psi\Big( f(\cdot), M_W\big(\Phi(f,f)\big)(\cdot)\Big) \right\|_\infty$$

$$\leq L_\Psi \Bigg( 4\frac{\varepsilon_1}{\sqrt{N}\mathrm{d}_{min}^2} \|W\|_\infty (L_\Phi\|f\|_\infty + \|\Phi(0,0)\|_\infty)$$

$$+ N^{-\frac{1}{2(D_\chi+1)}} \left( 2\Big( \frac{\|W\|_\infty}{\mathrm{d}_{min}} L_\Phi L_f + \frac{L_W}{\mathrm{d}_{min}} \big(L_\Phi\|f\|_\infty + \|\Phi(0,0)\|_\infty\big) \Big) \right.$$

$$+ \frac{C_\chi}{\sqrt{2}} \Big( \frac{\|W\|_\infty}{\mathrm{d}_{min}} \big(L_\Phi\|f\|_\infty + \|\Phi(0,0)\|_\infty\big) \Big) \sqrt{\log(C_\chi) + \frac{D_\chi}{2(D_\chi+1)}\log(N) + \log(2/p)} \Bigg) \Bigg),$$

$$(26)$$

*where $\varepsilon_1$ is defined in (22).*

*Proof.* We calculate,

$$\left\| \Psi\Big( f(\cdot), M_X\big(\Phi(f,f)\big)(\cdot)\Big) - \Psi\Big( f(\cdot), M_W\big(\Phi(f,f)\big)(\cdot)\Big) \right\|_\infty$$

$$\leq L_\Psi \big\| M_X\big(\Phi(f,f)\big)(\cdot) - M_W\big(\Phi(f,f)\big)(\cdot) \big\|_\infty,$$

and apply Lemma B.5 to the right-hand-side. $\square$

We continue by providing three lemmas which capture deterministic properties of cMPNNs and gMPNNs. We start by showing that the infinity norm of the output of the $l$-th layer of a cMPNN $f^{(l)}$ can be bounded in terms of the infinity norm of its input $f$.

**Lemma B.7.** *Let $(\chi, d, \mu)$ be a metric-measure space, $W$ be a kernel and $\Theta = ((\Phi^{(l)})_{l=1}^T, (\Psi^{(l)})_{l=1}^T)$ be a MPNN s.t. Assumptions A.10.1-7. are satisfied. Consider a metric-space signal $f : \chi \to \mathbb{R}^F$ with $\|f\|_\infty < \infty$. Then, for $l = 0, \dots, T-1$, the cMPNN output $f^{(l+1)}$ satisfies*

$$\|f^{(l+1)}\|_\infty \leq B_1^{(l+1)} + \|f\|_\infty B_2^{(l+1)},$$

*where*

$$B_1^{(l+1)} = \sum_{k=1}^{l+1} \left( L_{\Psi^{(k)}} \|\Phi^{(k)}(0,0)\|_\infty + \|\Psi^{(k)}(0,0)\|_\infty \right) \prod_{l'=k+1}^{l+1} L_{\Psi^{(l')}} \left( 1 + L_{\Phi^{(l')}} \right) \qquad (27)$$

*and*

$$B_2^{(l+1)} = \prod_{k=1}^{l+1} L_{\Psi^{(k)}} \left( 1 + L_{\Phi^{(k)}} \right). \qquad (28)$$

*Proof.* Let $l = 0, \ldots, T-1$. Then, for $k = 0, \ldots, l$, we have

$$\|f^{(k+1)}(\cdot)\|_\infty = \left\| \Psi^{(k+1)} \left( f^{(k)}(\cdot), M_W \left( \Phi^{(k+1)}(f^{(k)}, f^{(k)}) \right)(\cdot) \right) \right\|_\infty$$

$$\leq \left\| \Psi^{(k+1)} \left( f^{(k)}(\cdot), M_W \left( \Phi^{(k+1)}(f^{(k)}, f^{(k)}) \right)(\cdot) \right) - \Psi^{(k+1)}(0,0) \right\|_\infty + \|\Psi^{(k+1)}(0,0)\|_\infty$$

$$\leq L_{\Psi^{(k+1)}} \left( \|f^{(k)}\|_\infty + \left\| M_W \left( \Phi^{(k+1)}(f^{(k)}, f^{(k)}) \right)(\cdot) \right\|_\infty \right) + \|\Psi^{(k+1)}(0,0)\|_\infty.$$

For the message term, we have

$$\left\| M_W \left( \Phi^{(k+1)}(f^{(k)}, f^{(k)}) \right)(\cdot) \right\|_\infty = \left\| \int_\chi \frac{W(\cdot, y)}{d_W(\cdot)} \Phi^{(k+1)} \left( f^{(k)}(\cdot), f^{(k)}(y) \right) d\mu(y) \right\|_\infty$$

$$\leq \left\| \int_\chi \left| \frac{W(\cdot, y)}{d_W(\cdot)} \right| d\mu(y) \cdot \sup_{y \in \chi} \left| \Phi^{(k+1)}(f^{(k)}(\cdot), f^{(k)}(y)) \right| \right\|_\infty$$

$$\leq L_{\Phi^{(k+1)}} \|f^{(k)}\|_\infty + \|\Phi^{(k+1)}(0,0)\|_\infty.$$

Hence,

$$\|f^{(k+1)}(\cdot)\|_\infty$$
$$\leq L_{\Psi^{(k+1)}} \left( \|f^{(k)}\|_\infty + \left( L_{\Phi^{(k+1)}} \|f^{(k)}\|_\infty + \|\Phi^{(k+1)}(0,0)\|_\infty \right) \right) + \|\Psi^{(k+1)}(0,0)\|_\infty,$$

which we can reorder to

$$\|f^{(k+1)}(\cdot)\|_\infty$$
$$\leq L_{\Psi^{(k+1)}} \left( 1 + L_{\Phi^{(k+1)}} \right) \|f^{(k)}\|_\infty + L_{\Psi^{(k+1)}} \|\Phi^{(k+1)}(0,0)\|_\infty + \|\Psi^{(k+1)}(0,0)\|_\infty.$$

We apply Lemma B.11 to solve this recurrence relation which finishes the proof. □

In the following, we denote by $L_{f^{(l)}}$ the Lipschitz constant of $f^{(l)}$. The next lemma bounds $L_{f^{(l+1)}}$ in terms of $L_f$.

**Lemma B.8.** *Let $(\chi, d, \mu)$ be a metric-measure space, $W$ be a kernel and $\Theta = ((\Phi^{(l)})_{l=1}^T, (\Psi^{(l)})_{l=1}^T)$ be a MPNN s.t. Assumptions A.10.1-7. are satisfied. Consider a Lipschitz continuous metric-space signal $f : \chi \to \mathbb{R}^F$ with $\|f\|_\infty < \infty$ and Lipschitz constant $L_f$. Then, for $l = 0, \ldots, T-1$, the cMPNN output $f^{(l+1)}$ is Lipschitz continuous with Lipschitz constant $L_{f^{(l+1)}}$ satisfying*

$$L_{f^{(l+1)}} \leq \sum_{k=1}^{l+1} \left( \left( L_{\Psi^{(k)}} \frac{L_W}{d_{\min}} (\|\Phi^{(k)}(0,0)\|_\infty + L_{\Phi^{(k)}} \|f^{(k-1)}\|_\infty) + L_{\Psi^{(k)}} \|W\|_\infty (\|\Phi^{(k)}(0,0)\|_\infty \right. \right.$$

$$\left. \left. + L_{\Phi^{(k)}} \|f^{(k-1)}\|_\infty) \frac{L_W}{d_{\min}^2} \right) \prod_{l'=k+1}^{l+1} L_{\Psi^{(l')}} \left( 1 + \frac{\|W\|_\infty}{d_{\min}} L_{\Phi^{(l')}} \right) \right)$$

$$+ L_f \prod_{k=1}^{l+1} L_{\Psi^{(k)}} \left( 1 + \frac{\|W\|_\infty}{d_{\min}} L_{\Phi^{(k)}} \right).$$

*Proof.* Let $l = 0, \ldots, T-1$ and consider $k = 0, \ldots, l$. For $x, x' \in \chi$, we have

$$\|f^{(k+1)}(x) - f^{(k+1)}(x')\|_\infty$$
$$= \left\| \Psi^{(k+1)}\left(f^{(k)}(x), M_W(\Phi^{(k+1)}(f^{(k)}, f^{(k)}))(x)\right) \right.$$
$$\left. - \Psi^{(k+1)}\left(f^{(k)}(x'), M_W(\Phi^{(k+1)}(f^{(k)}, f^{(k)}))(x')\right) \right\|_\infty$$
$$\leq L_{\Psi^{(k+1)}}\left( \left\| f^{(k)}(x) - f^{(k)}(x') \right\|_\infty \right.$$
$$+ \left\| M_W(\Phi^{(k+1)}(f^{(k)}, f^{(k)}))(x) - M_W(\Phi^{(k+1)}(f^{(k)}, f^{(k)}))(x') \right\|_\infty \Big)$$
$$\leq L_{\Psi^{(k+1)}}\left( L_{f^{(k)}} d(x, x') + \| M_W(\Phi^{(k+1)}(f^{(k)}, f^{(k)}))(x) - M_W(\Phi^{(k+1)}(f^{(k)}, f^{(k)}))(x')\|_\infty \right). \tag{29}$$

For the second term, we have

$$\|M_W(\Phi^{(k+1)}(f^{(k)}, f^{(k)}))(x) - M_W(\Phi^{(k+1)}(f^{(k)}, f^{(k)}))(x')\|_\infty$$
$$= \| \int_\chi \frac{W(x,y)}{d_W(x)} \Phi^{(k+1)}\left(f^{(k)}(x), f^{(k)}(y)\right) - \frac{W(x',y)}{d_W(x')} \Phi^{(k+1)}\left(f^{(k)}(x'), f^{(k)}(y)\right) d\mu(y)\|_\infty$$
$$\leq \int_\chi \left\| \frac{W(x,y)}{d_W(x)} \Phi^{(k+1)}\left(f^{(k)}(x), f^{(k)}(y)\right) - \frac{W(x',y)}{d_W(x)} \Phi^{(k+1)}\left(f^{(k)}(x), f^{(k)}(y)\right) \right\|_\infty d\mu(y)$$
$$+ \int_\chi \left\| \frac{W(x',y)}{d_W(x)} \Phi^{(k+1)}\left(f^{(k)}(x), f^{(k)}(y)\right) - \frac{W(x',y)}{d_W(x)} \Phi^{(k+1)}\left(f^{(k)}(x'), f^{(k)}(y)\right) \right\|_\infty d\mu(y) \tag{30}$$
$$+ \int_\chi \left\| \frac{W(x',y)}{d_W(x)} \Phi^{(k+1)}\left(f^{(k)}(x'), f^{(k)}(y)\right) - \frac{W(x',y)}{d_W(x')} \Phi^{(k+1)}\left(f^{(k)}(x'), f^{(k)}(y)\right) \right\|_\infty d\mu(y)$$
$$= (A) + (B) + (C)$$

For $(A)$, we have

$$(A) = \int_\chi \left\| \frac{W(x,y)}{d_W(x)} \Phi^{(k+1)}\left(f^{(k)}(x), f^{(k)}(y)\right) - \frac{W(x',y)}{d_W(x)} \Phi^{(k+1)}\left(f^{(k)}(x), f^{(k)}(y)\right) \right\|_\infty d\mu(y)$$
$$= \int_\chi \frac{|W(x,y) - W(x',y)|}{d_W(x)} \left\| \Phi^{(k+1)}\left(f^{(k)}(x), f^{(k)}(y)\right) \right\|_\infty d\mu(y)$$
$$\leq L_W \frac{d(x,x')}{d_{\min}} \int_\chi \left\| \Phi^{(k+1)}\left(f^{(k)}(x), f^{(k)}(y)\right) \right\|_\infty d\mu(y)$$
$$\leq \frac{L_W}{d_{\min}} \left( \|\Phi^{(k+1)}(0,0)\|_\infty + L_{\Phi^{(k+1)}} \|f^{(k)}\|_\infty \right) d(x, x').$$

For $(B)$, we have

$$(B) = \int_\chi \left\| \frac{W(x',y)}{d_W(x)} \Phi^{(k+1)}\left(f^{(k)}(x), f^{(k)}(y)\right) - \frac{W(x',y)}{d_W(x)} \Phi^{(k+1)}\left(f^{(k)}(x'), f^{(k)}(y)\right) \right\|_\infty d\mu(y)$$
$$= \int_\chi \frac{|W(x',y)|}{|d_W(x)|} \left\| \Phi^{(k+1)}\left(f^{(k)}(x), f^{(k)}(y)\right) - \Phi^{(k+1)}\left(f^{(k)}(x'), f^{(k)}(y)\right) \right\|_\infty d\mu(y)$$
$$\leq \frac{\|W\|_\infty}{d_{\min}} L_{\Phi^{(k+1)}} \int_\chi \left\| \left(f^{(k)}(x), f^{(k)}(y)\right) - \left(f^{(k)}(x'), f^{(k)}(y)\right) \right\|_\infty d\mu(y)$$
$$\leq \frac{\|W\|_\infty}{d_{\min}} L_{\Phi^{(k+1)}} \|f^{(k)}(x) - f^{(k)}(x'))\|_\infty$$
$$\leq \frac{\|W\|_\infty}{d_{\min}} L_{\Phi^{(k+1)}} L_{f^{(k)}} d(x, x').$$

For $(C)$, we have

$$(C) = \int_\chi \left\| \frac{W(x',y)}{\mathrm{d}_W(x)} \Phi^{(k+1)}\big(f^{(k)}(x'), f^{(k)}(y)\big) - \frac{W(x',y)}{\mathrm{d}_W(x')} \Phi^{(k+1)}\big(f^{(k)}(x'), f^{(k)}(y)\big) \right\|_\infty d\mu(y)$$

$$= \int_\chi |W(x',y)| \left| \frac{1}{\mathrm{d}_W(x)} - \frac{1}{\mathrm{d}_W(x')} \right| \left\| \Phi^{(k+1)}\big(f^{(k)}(x'), f^{(k)}(y)\big) \right\|_\infty d\mu(y)$$

$$\leq \|W\|_\infty (\|\Phi^{(k+1)}(0,0)\|_\infty + L_{\Phi^{(k+1)}} \|f^{(k)}\|_\infty) \frac{L_W}{\mathrm{d}_{\min}^2} d(x,x'),$$

where the last inequality holds since

$$\left| \frac{1}{\mathrm{d}_W(x)} - \frac{1}{\mathrm{d}_W(x')} \right| \leq \frac{|\mathrm{d}_W(x') - \mathrm{d}_W(x)|}{|\mathrm{d}_W(x)\mathrm{d}_W(x')|}$$

$$\leq \frac{1}{\mathrm{d}_{\min}^2} |\mathrm{d}_W(x') - \mathrm{d}_W(x)|$$

$$\leq \frac{1}{\mathrm{d}_{\min}^2} \int_\chi |W(x',y) - W(x,y)| d\mu(y)$$

$$\leq \frac{1}{\mathrm{d}_{\min}^2} \int_\chi L_W d(x,x') d\mu(y)$$

$$\leq \frac{L_W}{\mathrm{d}_{\min}^2} d(x,x').$$

Hence, by plugging (30) and our bounds for $(A), (B)$ and $(C)$ into (29), we have

$$\|f^{(k+1)}(x) - f^{(k+1)}(x')\|_\infty$$

$$\leq L_{\Psi^{(k+1)}} \Big( L_{f^{(k)}} d(x,x') + \|M_W\big(\Phi^{(k+1)}(f^{(k)}, f^{(k)})\big)(x) - M_W\big(\Phi^{(k+1)}(f^{(k)}, f^{(k)})\big)(x')\|_\infty \Big)$$

$$\leq L_{\Psi^{(k+1)}} \Big( L_{f^{(k)}} d(x,x') + (A) + (B) + (C) \Big)$$

$$\leq L_{\Psi^{(k+1)}} \Big( L_{f^{(k)}} + \frac{L_W}{\mathrm{d}_{\min}} (\|\Phi^{(k+1)}(0,0)\|_\infty + L_{\Phi^{(k+1)}} \|f^{(k)}\|_\infty)$$

$$+ \frac{\|W\|_\infty}{\mathrm{d}_{\min}} L_{\Phi^{(k+1)}} L_{f^{(k)}} + \|W\|_\infty (\|\Phi^{(k+1)}(0,0)\|_\infty + L_{\Phi^{(k+1)}} \|f^{(k)}\|_\infty) \frac{L_W}{\mathrm{d}_{\min}^2} \Big) d(x,x').$$

Hence,

$$L_{f^{(k+1)}} \leq L_{\Psi^{(k+1)}} \frac{L_W}{\mathrm{d}_{\min}} (\|\Phi^{(k+1)}(0,0)\|_\infty + L_{\Phi^{(k+1)}} \|f^{(k)}\|_\infty) + L_{\Psi^{(k+1)}} \left( 1 + \frac{\|W\|_\infty}{\mathrm{d}_{\min}} L_{\Phi^{(k+1)}} \right) L_{f^{(k)}}$$

$$+ L_{\Psi^{(k+1)}} \|W\|_\infty (\|\Phi^{(k+1)}(0,0)\|_\infty + L_{\Phi^{(k+1)}} \|f^{(k)}\|_\infty) \frac{L_W}{\mathrm{d}_{\min}^2}.$$

We finish the proof by solving the recurrence relation with Lemma B.11. $\qquad\square$

**Corollary B.9.** *Consider the same setting as in Lemma B.8. Then, for $l = 0, \dots, T-1$,*

$$L_{f^{(l)}} \leq Z_1^{(l)} + Z_2^{(l)} \|f\|_\infty + Z_3^{(l)} L_f,$$

*where $Z_1^{(l)}$, $Z_2^{(l)}$ and $Z_3^{(l)}$ are independent of $f$ and defined as*

$$Z_1^{(l)} = \sum_{k=1}^l \left( \left( L_{\Psi^{(k)}} \frac{L_W}{\mathrm{d}_{\min}} \|\Phi^{(k)}(0,0)\|_\infty + L_{\Psi^{(k)}} \|W\|_\infty \|\Phi^{(k)}(0,0)\|_\infty \frac{L_W}{\mathrm{d}_{\min}^2} \right) \right.$$

$$\left. + B_1^{(k-1)} \left( L_{\Psi^{(k)}} \frac{L_W}{\mathrm{d}_{\min}} L_{\Phi^{(k)}} + L_{\Psi^{(k)}} \|W\|_\infty L_{\Phi^{(k)}} \frac{L_W}{\mathrm{d}_{\min}^2} \right) \right) \prod_{l'=k+1}^l L_{\Psi^{(l')}} \left( 1 + \frac{\|W\|_\infty}{\mathrm{d}_{\min}} L_{\Phi^{(l')}} \right),$$

$$Z_2^{(l)} = \sum_{k=1}^l B_2^{(k-1)} \left( L_{\Psi^{(k)}} \frac{L_W}{\mathrm{d}_{\min}} L_{\Phi^{(k)}} + L_{\Psi^{(k)}} \|W\|_\infty L_{\Phi^{(k)}} \frac{L_W}{\mathrm{d}_{\min}^2} \right) \prod_{l'=k+1}^l L_{\Psi^{(l')}} \left( 1 + \frac{\|W\|_\infty}{\mathrm{d}_{\min}} L_{\Phi^{(l')}} \right),$$

$$Z_3^{(l)} = \prod_{k=1}^l L_{\Psi^{(k)}} \left( 1 + \frac{\|W\|_\infty}{\mathrm{d}_{\min}} L_{\Phi^{(k)}} \right),$$

$$(31)$$

*where $B_1^{(k)}$ and $B_2^{(k)}$ are defined in (27) and (28).*

*Proof.* By Lemma B.8, we have

$$
\begin{aligned}
L_{f^{(l)}} \leq & \sum_{k=1}^{l} \Bigg( \Big( L_{\Psi^{(k)}} \frac{L_W}{\mathrm{d}_{\min}} (\|\Phi^{(k)}(0,0)\|_\infty + L_{\Phi^{(k)}} \|f^{(k-1)}\|_\infty) + L_{\Psi^{(k)}} \|W\|_\infty (\|\Phi^{(k)}(0,0)\|_\infty \\
& + L_{\Phi^{(k)}} \|f^{(k-1)}\|_\infty) \frac{L_W}{\mathrm{d}_{\min}^2} \Big) \prod_{l'=k+1}^{l} L_{\Psi^{(l')}} \Big( 1 + \frac{\|W\|_\infty}{\mathrm{d}_{\min}} L_{\Phi^{(l')}} \Big) \Bigg) \\
& + L_f \prod_{k=1}^{l} L_{\Psi^{(k)}} \Big( 1 + \frac{\|W\|_\infty}{\mathrm{d}_{\min}} L_{\Phi^{(k)}} \Big) \\
= & \sum_{k=1}^{l} \Big( L_{\Psi^{(k)}} \frac{L_W}{\mathrm{d}_{\min}} \|\Phi^{(k)}(0,0)\|_\infty + L_{\Psi^{(k)}} \|W\|_\infty \|\Phi^{(k)}(0,0)\|_\infty \frac{L_W}{\mathrm{d}_{\min}^2} \Big) \\
& \prod_{l'=k+1}^{l} L_{\Psi^{(l')}} \Big( 1 + \frac{\|W\|_\infty}{\mathrm{d}_{\min}} L_{\Phi^{(l')}} \Big) \\
& + \sum_{k=1}^{l} \|f^{(k-1)}\|_\infty \Big( L_{\Psi^{(k)}} \frac{L_W}{\mathrm{d}_{\min}} L_{\Phi^{(k)}} + L_{\Psi^{(k)}} \|W\|_\infty L_{\Phi^{(k)}} \frac{L_W}{\mathrm{d}_{\min}^2} \Big) \\
& \prod_{l'=k+1}^{l} L_{\Psi^{(l')}} \Big( 1 + \frac{\|W\|_\infty}{\mathrm{d}_{\min}} L_{\Phi^{(l')}} \Big) \\
& + L_f \prod_{k=1}^{l} L_{\Psi^{(k)}} \Big( 1 + \frac{\|W\|_\infty}{\mathrm{d}_{\min}} L_{\Phi^{(k)}} \Big) \\
\leq & \sum_{k=1}^{l} \Big( L_{\Psi^{(k)}} \frac{L_W}{\mathrm{d}_{\min}} \|\Phi^{(k)}(0,0)\|_\infty + L_{\Psi^{(k)}} \|W\|_\infty \|\Phi^{(k)}(0,0)\|_\infty \frac{L_W}{\mathrm{d}_{\min}^2} \Big) \\
& \prod_{l'=k+1}^{l} L_{\Psi^{(l')}} \Big( 1 + \frac{\|W\|_\infty}{\mathrm{d}_{\min}} L_{\Phi^{(l')}} \Big) \\
& + \sum_{k=1}^{l} (B_1^{(k-1)} + B_2^{(k-1)} \|f\|_\infty) \Big( L_{\Psi^{(k)}} \frac{L_W}{\mathrm{d}_{\min}} L_{\Phi^{(k)}} + L_{\Psi^{(k)}} \|W\|_\infty L_{\Phi^{(k)}} \frac{L_W}{\mathrm{d}_{\min}^2} \Big) \\
& \prod_{l'=k+1}^{l} L_{\Psi^{(l')}} \Big( 1 + \frac{\|W\|_\infty}{\mathrm{d}_{\min}} L_{\Phi^{(l')}} \Big) \\
& + L_f \prod_{k=1}^{l} L_{\Psi^{(k)}} \Big( 1 + \frac{\|W\|_\infty}{\mathrm{d}_{\min}} L_{\Phi^{(k)}} \Big),
\end{aligned}
$$

where the last inequality holds by Lemma B.7. □

We continue with the following simple lemma which bounds the infinity norm of the output of a gMPNN.

**Lemma B.10.** *Let $(\chi, d, \mu)$ be a metric-measure space, $W$ be a kernel and $\Theta = ((\Phi^{(l)})_{l=1}^T, (\Psi^{(l)})_{l=1}^T)$ be a MPNN s.t. Assumptions A.10.1-8. are satisfied. Consider a metric-space signal $f : \chi \to \mathbb{R}^F$ with $\|f\|_\infty < \infty$. Consider a graph $(G, \mathbf{f}) \sim (W, f)$ with $N$ nodes and corresponding graph features. Then,*

$$
\|\Theta_G(\mathbf{f})\|_{2;\infty}^2 \leq N^T (A' + A'' \|f\|_\infty^2),
$$

*where*

$$A' = \sum_{l=1}^{T} \left( 2(L_{\Psi^{(l)}})^2 \frac{2}{W_{\text{diag}}} \|W\|_{\infty} \|\Phi^{(l)}(0,0)\|_{\infty}^2 + 2\|\Psi^{(l)}(0,0)\|_{\infty}^2 \right)$$

$$\prod_{l'=l+1}^{T} 2(L_{\Psi^{(l')}})^2 \left( \frac{2}{W_{\text{diag}}} \|W\|_{\infty}(L_{\Phi^{(l')}})^2 + 1 \right)$$

*and*

$$A'' = \prod_{l=1}^{T} 2(L_{\Psi^{(l)}})^2 \left( \frac{2}{W_{\text{diag}}} \|W\|_{\infty}(L_{\Phi^{(l)}})^2 + 1 \right).$$

*Proof.* Let $l = 0, \ldots, T-1$. We have

$$\|\mathbf{f}^{(l+1)}\|_{2;\infty}^2 = \frac{1}{N} \sum_{i=1}^{N} \|\mathbf{f}_i^{(l+1)}\|_{\infty}^2,$$

where $\mathbf{f}_i^{(l+1)} = \Psi^{(l+1)}(\mathbf{f}_i^{(l)}, \mathbf{m}_i^{(l+1)})$ with $\mathbf{m}_i^{(l+1)} = M_G\big(\Phi^{(l+1)}(\mathbf{f}^{(l)}, \mathbf{f}^{(l)})\big)(X_i)$. By using the Lipschitz continuity of $\Psi^{(l+1)}$, we get

$$\begin{aligned}
\|\mathbf{f}_i^{(l+1)}\|_{\infty}^2 &\leq 2\big(\|\Psi^{(l+1)}(\mathbf{f}_i^{(l)}, \mathbf{m}_i^{(l+1)}) - \Psi^{(l+1)}(0,0)\|_{\infty}^2 + \|\Psi^{(l+1)}(0,0)\|_{\infty}^2\big) \\
&\leq 2\big((L_{\Psi^{(l+1)}})^2(\|\mathbf{f}_i^{(l)}\|_{\infty}^2 + \|\mathbf{m}_i^{(l+1)}\|_{\infty}^2) + \|\Psi^{(l+1)}(0,0)\|_{\infty}^2\big)
\end{aligned} \tag{32}$$

For the message term we calculate

$$\begin{aligned}
\|\mathbf{m}_i^{(l+1)}\|_{\infty}^2 &= \left\| \frac{1}{\sum_{j=1}^{N} W(X_i, X_j)} \sum_{j=1}^{N} W(X_i, X_j)\Phi^{(l+1)}(\mathbf{f}_i^{(l)}, \mathbf{f}_j^{(l)}) \right\|_{\infty}^2 \\
&= \left| \frac{1}{\sum_{j=1}^{N} W(X_i, X_j)} \right|^2 \left\| \sum_{j=1}^{N} \sqrt{W(X_i, X_j)} \cdot \sqrt{W(X_i, X_j)}\Phi^{(l+1)}(\mathbf{f}_i^{(l)}, \mathbf{f}_j^{(l)}) \right\|_{\infty}^2 \\
&\leq \left| \frac{1}{\sum_{j=1}^{N} W(X_i, X_j)} \right|^2 \sum_{j=1}^{N} |W(X_i, X_j)| \sum_{j=1}^{N} |W(X_i, X_j)| \|\Phi^{(l+1)}(\mathbf{f}_i^{(l)}, \mathbf{f}_j^{(l)})\|_{\infty}^2 \\
&\leq \left| \frac{1}{\sum_{j=1}^{N} W(X_i, X_j)} \right| \|W\|_{\infty} \sum_{j=1}^{N} \|\Phi^{(l+1)}(\mathbf{f}_i^{(l)}, \mathbf{f}_j^{(l)})\|_{\infty}^2 \\
&\leq \frac{\|W\|_{\infty}}{W_{\text{diag}}} \sum_{j=1}^{N} \|\Phi^{(l+1)}(\mathbf{f}_i^{(l)}, \mathbf{f}_j^{(l)})\|_{\infty}^2
\end{aligned}$$

where the inequality follows from Cauchy-Schwarz inequality. Per assumption, we have $|W(X_i, X_i)| \geq W_{\text{diag}}$ and for every $i = 1, \ldots, N$,

$$\begin{aligned}
\|\Phi^{(l+1)}(\mathbf{f}_i^{(l)}, \mathbf{f}_j^{(l)})\|_{\infty}^2 &= \|\Phi^{(l+1)}(\mathbf{f}_i^{(l)}, \mathbf{f}_j^{(l)}) - \Phi^{(l+1)}(0,0) + \Phi^{(l+1)}(0,0)\|_{\infty}^2 \\
&\leq 2\Big(\|\Phi^{(l+1)}(\mathbf{f}_i^{(l)}, \mathbf{f}_j^{(l)}) - \Phi^{(l+1)}(0,0)\|_{\infty}^2 + \|\Phi^{(l+1)}(0,0)\|_{\infty}^2\Big) \\
&\leq 2\Big((L_{\Phi^{(l+1)}})^2(\|\mathbf{f}_i^{(l)}\|_{\infty}^2 + \|\mathbf{f}_j^{(l)}\|_{\infty}^2) + \|\Phi^{(l+1)}(0,0)\|_{\infty}^2\cdot\Big)
\end{aligned}$$

Hence,

$$\begin{aligned}
\|\mathbf{m}_i^{(l+1)}\|_{\infty}^2 &\leq \frac{2\|W\|_{\infty}}{W_{\text{diag}}} \frac{N}{N} \sum_{j=1}^{N} \Big((L_{\Phi^{(l+1)}})^2(\|\mathbf{f}_i^{(l)}\|_{\infty}^2 + \|\mathbf{f}_j^{(l)}\|_{\infty}^2) + \|\Phi^{(l+1)}(0,0)\|_{\infty}^2\Big) \\
&\leq N\frac{2\|W\|_{\infty}}{W_{\text{diag}}} \Big((L_{\Phi^{(l+1)}})^2\|\mathbf{f}_i^{(l)}\|_{\infty}^2 + (L_{\Phi^{(l+1)}})^2\|\mathbf{f}^{(l)}\|_{2;\infty}^2 + \|\Phi^{(l+1)}(0,0)\|_{\infty}^2\Big).
\end{aligned} \tag{33}$$

By (32) and (33), we have

$$\|\mathbf{f}^{(l+1)}\|_{2;\infty}^2 \leq \frac{1}{N}\sum_{i=1}^{N} 2\Big((L_{\Psi^{(l+1)}})^2(\|\mathbf{f}_i^{(l)}\|_\infty^2 + \|\mathbf{m}_i^{(l+1)}\|_\infty^2) + \|\Psi^{(l+1)}(0,0)\|_\infty^2\Big)$$

$$\leq \frac{1}{N}\sum_{i=1}^{N} 2\bigg((L_{\Psi^{(l+1)}})^2\Big(\|\mathbf{f}_i^{(l)}\|_\infty^2 + N\frac{2\|W\|_\infty}{\mathrm{W}_{\mathrm{diag}}}((L_{\Phi^{(l+1)}})^2\|\mathbf{f}_i^{(l)}\|_\infty^2$$

$$+ (L_{\Phi^{(l+1)}})^2\|\mathbf{f}^{(l)}\|_{2;\infty}^2 + \|\Phi^{(l+1)}(0,0)\|_\infty^2)\Big) + \|\Psi^{(l+1)}(0,0)\|_\infty^2\bigg)$$

$$= 2(L_{\Psi^{(l+1)}})^2\Big(\frac{1}{N}\sum_{i=1}^{N}\|\mathbf{f}_i^{(l)}\|_\infty^2 + N\frac{2\|W\|_\infty}{\mathrm{W}_{\mathrm{diag}}}((L_{\Phi^{(l+1)}})^2\frac{1}{N}\sum_{i=1}^{N}\|\mathbf{f}_i^{(l)}\|_\infty^2$$

$$+ (L_{\Phi^{(l+1)}})^2\|\mathbf{f}^{(l)}\|_{2;\infty}^2 + \|\Phi^{(l+1)}(0,0)\|_\infty^2)\Big) + 2\|\Psi^{(l+1)}(0,0)\|_\infty^2$$

$$= 2(L_{\Psi^{(l+1)}})^2\Big(\|\mathbf{f}^{(l)}\|_{2;\infty}^2 + N\frac{2\|W\|_\infty}{\mathrm{W}_{\mathrm{diag}}}((L_{\Phi^{(l+1)}})^2\|\mathbf{f}^{(l)}\|_{2;\infty}^2 + \|\Phi^{(l+1)}(0,0)\|_\infty^2)\Big)$$

$$+ 2\|\Psi^{(l+1)}(0,0)\|_\infty^2$$

$$= 2(L_{\Psi^{(l+1)}})^2\Big(N\frac{2\|W\|_\infty}{\mathrm{W}_{\mathrm{diag}}}(L_{\Phi^{(l+1)}})^2 + 1\Big)\|\mathbf{f}^{(l)}\|_{2;\infty}^2$$

$$+ 2(L_{\Psi^{(l+1)}})^2 N\frac{2\|W\|_\infty}{\mathrm{W}_{\mathrm{diag}}}\|\Phi^{(l+1)}(0,0)\|_\infty^2 + 2\|\Psi^{(l+1)}(0,0)\|_\infty^2$$

Hence, by $\|\mathbf{f}\|_{2;\infty}^2 \leq \|f\|_\infty^2$ and Lemma B.11, we have

$$\|\mathbf{f}^{(T)}\|_{2;\infty}^2 \leq \sum_{l=1}^{T}\Big(2(L_{\Psi^{(l)}})^2 N\frac{2\|W\|_\infty}{\mathrm{W}_{\mathrm{diag}}}\|\Phi^{(l)}(0,0)\|_\infty^2 + 2\|\Psi^{(l)}(0,0)\|_\infty^2\Big)$$

$$\prod_{l'=l+1}^{T} 2(L_{\Psi^{(l')}})^2\Big(N\frac{2\|W\|_\infty}{\mathrm{W}_{\mathrm{diag}}}(L_{\Phi^{(l')}})^2 + 1\Big)$$

$$+ \|f\|_\infty^2\prod_{l=1}^{T}\Big(2(L_{\Psi^{(l)}})^2\Big(N\frac{2\|W\|_\infty}{\mathrm{W}_{\mathrm{diag}}}(L_{\Phi^{(l)}})^2 + 1\Big)\Big)$$

$$\leq N^T\sum_{l=1}^{T}\Big(2(L_{\Psi^{(l)}})^2\frac{2\|W\|_\infty}{\mathrm{W}_{\mathrm{diag}}}\|\Phi^{(l)}(0,0)\|_\infty^2 + 2\|\Psi^{(l)}(0,0)\|_\infty^2\Big)$$

$$\prod_{l'=l+1}^{T} 2(L_{\Psi^{(l')}})^2\Big(\frac{2}{\mathrm{W}_{\mathrm{diag}}^2}\|W\|_\infty^2(L_{\Phi^{(l')}})^2 + 1\Big)$$

$$+ \|f\|_\infty^2 N^T\prod_{l}^{T}\Big(2(L_{\Psi^{(l)}})^2\Big(\frac{2\|W\|_\infty}{\mathrm{W}_{\mathrm{diag}}}(L_{\Phi^{(l)}})^2 + 1\Big)\Big).$$

$$\square$$

We finish this subsection with the following easily verifiable lemma that provides a general solution for certain recurrence relations.

**Lemma B.11.** *Let $(\eta^{(l)})_{l=0}^{T}$ be a sequence of real numbers satisfying $\eta^{(l+1)} \leq a^{(l+1)}\eta^{(l)} + b^{(l+1)}$ for $l = 0, \ldots, T-1$, for some real numbers $a^{(l)}, b^{(l)}, l = 1, \ldots, T$. Then*

$$\eta^{(T)} \leq \sum_{l=1}^{T} b^l \prod_{l'=l+1}^{T} a^{(l')} + \eta^{(0)}\prod_{l=1}^{T} a^{(l)},$$

*where we define the product $\prod_{T+1}^{T}$ as 1.*

## B.2 Proof of Theorem 3.1

The idea of the Proof of Theorem 3.1 is as follows. We first use Corollary B.6 to bound the error between a cMPNN and a gMPNN layer-wise, when the input of layer $l$ of the gMPNN is exactly the sampled graph signal from the output of layer $l-1$ of the cMPNN. This is shown in Corollary B.12. Then, we use this to provide a recurrence relation for the true error between a cMPNN and the corresponding gMPNN in Lemma B.13. We solve this recurrence relation in Corollary B.14, where we have an error bound that depends only on the parameters of the MPNN, the regularity of the kernel and the regularity of the continuous output metric-space signals of the layers of the cMPNN. We remove the last dependency in Theorem B.15. We then analyze the additional error by a final pooling layer, leading to the formulation and final proof of Theorem 3.1, rewritten as Theorem B.18.

**Corollary B.12.** *Let $(\chi, d, \mu)$ be a metric-measure space and $W$ be a kernel s.t. Assumptions A.10.1-6 are satisfied. Let $p \in (0, \frac{1}{2})$. Consider a graph $(G, \mathbf{f}) \sim (W, f)$ with $N$ nodes and corresponding graph features, where $N$ satisfies (14). If the event $\mathcal{F}_{\mathrm{Lip}}^p$ from Lemma B.5 occurs, then condition (15) together with (34) below are satisfied: For every MPNN $\Theta$ satisfying Assumption A.10.7. and $f : \chi \to \mathbb{R}^F$ with Lipschitz constant $L_f$, we have*

$$\mathrm{dist}\left(\Lambda_{\Theta_G}^{(l+1)}(S^X f^{(l)}), \Lambda_{\Theta_W}^{(l+1)}(f^{(l)})\right) \leq Q^{(l+1)} \tag{34}$$

*for all $l = 0, \ldots, T-1$, where $f^{(l)} = \Theta_W^{(l)} f$ as defined in (5), and $\Lambda_{\Theta_G}^{(l+1)}$ and $\Lambda_{\Theta_W}^{(l+1)}$ are defined in Definition A.9. Here,*

$$\begin{aligned}
Q^{(l+1)} = L_{\Psi^{(l+1)}} &\left(4 \frac{\varepsilon_1}{\sqrt{N} \mathrm{d}_{min}^2} \|W\|_\infty (L_{\Phi^{(l+1)}} \|f^{(l)}\|_\infty + \|\Phi^{(l+1)}(0,0)\|_\infty) \right. \\
&+ N^{-\frac{1}{2(D_\chi+1)}} \left(2\Big(\frac{\|W\|_\infty}{\mathrm{d}_{min}} L_{\Phi^{(l+1)}} L_{f^{(l)}} + \frac{L_W}{\mathrm{d}_{min}}\big(L_{\Phi^{(l+1)}} \|f^{(l)}\|_\infty + \|\Phi^{(l+1)}(0,0)\|_\infty\big)\Big) \right. \\
&+ \frac{C_\chi}{\sqrt{2}} \Big(\frac{\|W\|_\infty}{\mathrm{d}_{min}} \big(L_{\Phi^{(l+1)}} \|f^{(l)}\|_\infty + \|\Phi^{(l+1)}(0,0)\|_\infty\big)\Big) \\
&\left.\left. \cdot \sqrt{\log(C_\chi) + \frac{D_\chi}{2(D_\chi+1)} \log(N) + \log(2/p)}\right)\right),
\end{aligned} \tag{35}$$

*and* $\mathrm{dist}$ *is defined in (12).*

*Proof.* Let $l = 0, \ldots, T - 1$. We have,

$$\left(\text{dist}\left(\Lambda_{\Theta_G}^{(l+1)}(S^X f^{(l)}), \Lambda_{\Theta_W}^{(l+1)}(f^{(l)})\right)\right)^2$$

$$= \|\Lambda_{\Theta_G}^{(l+1)}(S^X f^{(l)}) - S^X \Lambda_{\Theta_W}^{(l+1)}(f^{(l)})\|_{2;\infty}^2$$

$$= \frac{1}{N} \sum_{i=1}^{N} \|\Lambda_{\Theta_G}^{(l+1)}(S^X f^{(l)})(X_i) - S^X \Lambda_{\Theta_W}^{(l+1)}(f^{(l)})(X_i)\|_\infty^2$$

$$= \frac{1}{N} \sum_{i=1}^{N} \left\| \Psi^{(l+1)}\left(f^{(l)}(X_i), M_G\left(\Phi^{(l+1)}(S^X f^{(l)}, S^X f^{(l)})\right)(X_i)\right) \right.$$

$$\left. - \Psi^{(l+1)}\left(f^{(l)}(X_i), M_W\left(\Phi^{(l+1)}(f^{(l)}, f^{(l)})\right)(X_i)\right) \right\|_\infty^2$$

$$= \frac{1}{N} \sum_{i=1}^{N} \left\| \Psi^{(l+1)}\left(f^{(l)}(X_i), M_X\left(\Phi^{(l+1)}(f^{(l)}, f^{(l)})\right)(X_i)\right) \right.$$

$$\left. - \Psi^{(l+1)}\left(f^{(l)}(X_i), M_W\left(\Phi^{(l+1)}(f^{(l)}, f^{(l)})\right)(X_i)\right) \right\|_\infty^2$$

$$\leq L_{\Psi^{(l+1)}}^2 \left( 4 \frac{\varepsilon_1}{\sqrt{N} \mathrm{d}_{min}^2} \|W\|_\infty (L_{\Phi^{(l+1)}} \|f^{(l)}\|_\infty + \|\Phi^{(l+1)}(0,0)\|_\infty) \right.$$

$$+ N^{-\frac{1}{2(D_\chi + 1)}} \left( 2\left( \frac{\|W\|_\infty}{\mathrm{d}_{min}} L_{\Phi^{(l+1)}} L_{f^{(l)}} + \frac{L_W}{\mathrm{d}_{min}} \left( L_{\Phi^{(l+1)}} \|f^{(l)}\|_\infty + \|\Phi^{(l+1)}(0,0)\|_\infty \right) \right) \right.$$

$$+ \frac{C_\chi}{\sqrt{2}} \left( \frac{\|W\|_\infty}{\mathrm{d}_{min}} \left( L_{\Phi^{(l+1)}} \|f^{(l)}\|_\infty + \|\Phi^{(l+1)}(0,0)\|_\infty \right) \right)$$

$$\left. \cdot \sqrt{\log(C_\chi) + \frac{D_\chi}{2(D_\chi + 1)} \log(N) + \log(2/p)} \right) \right)^2 ,$$

where the final inequality holds, by applying Corollary B.6. $\qquad\square$

**Lemma B.13.** *Let $(\chi, d, \mu)$ be a metric-measure space and $W$ be a kernel s.t. Assumptions A.10.1-6. are satisfied. Let $p \in (0, \frac{1}{2})$. Consider a graph $(G, \mathbf{f}) \sim (W, f)$ with $N$ nodes and corresponding graph features, where $N$ satisfies (14). Denote, for $l = 1, \ldots, T$,*

$$\varepsilon^{(l)} = \text{dist}(\Theta_G^{(l)}(\mathbf{f}), \Theta_W^{(l)}(f)),$$

*and $\varepsilon^{(0)} = \text{dist}(\mathbf{f}, f)$. If the event $\mathcal{F}_{\text{Lip}}^p$ from Lemma B.5 occurs, then, for every MPNN $\Theta$ satisfying Assumption A.10.7. and $f : \chi \to \mathbb{R}^F$ with Lipschitz constant $L_f$, the following recurrence relation holds:*

$$\varepsilon^{(l)} \leq K^{(l+1)} \varepsilon^{(l)} + Q^{(l+1)}$$

*for $l = 0, \ldots, T - 1$. Here, $Q^{(l+1)}$ is defined in (35), and*

$$K^{(l+1)} = \sqrt{(L_{\Psi^{(l+1)}})^2 + \frac{8\|W\|_\infty^2}{\mathrm{d}_{min}^2} (L_{\Phi^{(l+1)}})^2 (L_{\Psi^{(l+1)}})^2}. \tag{36}$$

*Proof.* In the event $\mathcal{F}_{\text{Lip}}^p$, by Corollary B.12, we have for every MPNN $\Theta$ satisfying Assumption A.10.7. and $f : \chi \to \mathbb{R}^F$ with Lipschitz constant $L_f$,

$$\text{dist}\left(\Lambda_{\Theta_G}^{(l+1)}(S^X f^{(l)}), \Lambda_{\Theta_W}^{(l+1)}(f^{(l)})\right) \leq Q^{(l+1)} \tag{37}$$

for $l = 0, \ldots, T - 1$, and

$$|\mathrm{d}_X(x)| \geq \frac{\mathrm{d}_{min}}{2} \tag{38}$$

for all $x \in \chi$. Let $l = 0, \ldots, T-1$. We have

$$
\begin{aligned}
&\operatorname{dist}(\Theta_G^{(l+1)}(\mathbf{f}), \Theta_W^{(l+1)}(f)) \\
&= \|\Theta_G^{(l+1)}(\mathbf{f}) - S^X \Theta_W^{(l+1)}(f)\|_{2;\infty} \\
&\leq \|\Theta_G^{(l+1)}(\mathbf{f}) - \Lambda_{\Theta_G}^{(l+1)}(S^X f^{(l)})\|_{2;\infty} + \|\Lambda_{\Theta_G}^{(l+1)}(S^X f^{(l)}) - S^X \Theta_{\Theta_W}^{(l+1)}(f)\|_{2;\infty} \\
&= \|\Lambda_G^{(l+1)}(\mathbf{f}^{(l)}) - \Lambda_G^{(l+1)}(S^X f^{(l)})\|_{2;\infty} + \|\Lambda_{\Theta_G}^{(l+1)}(S^X f^{(l)}) - S^X \Lambda_{\Theta_W}^{(l+1)}(f^{(l)})\|_{2;\infty} \\
&\leq \|\Lambda_{\Theta_G}^{(l+1)}(\mathbf{f}^{(l)}) - \Lambda_{\Theta_G}^{(l+1)}(S^X f^{(l)})\|_{2;\infty} + Q^{(l+1)}.
\end{aligned}
\tag{39}
$$

We bound the first term on the right-hand-side of (39) as follows.

$$
\begin{aligned}
&\|\Lambda_{\Theta_G}^{(l+1)}(\mathbf{f}^{(l)}) - \Lambda_{\Theta_G}^{(l+1)}(S^X f^{(l)})\|_{2;\infty}^2 \\
&= \frac{1}{N} \sum_{i=1}^{N} \left\| \Psi^{(l+1)}\left( \mathbf{f}_i^{(l)}, M_G\big(\Phi^{(l+1)}(\mathbf{f}^{(l)}, \mathbf{f}^{(l)})\big)(X_i) \right) \right. \\
&\qquad \left. - \Psi^{(l+1)}\left( (S^X f^{(l)})_i, M_G\big(\Phi^{(l+1)}(S^X f^{(l)}, S^X f^{(l)})\big)(X_i) \right) \right\|_{\infty}^2 \\
&\leq \frac{1}{N} (L_{\Psi^{(l+1)}})^2 \sum_{i=1}^{N} \left\| \left( \mathbf{f}_i^{(l)}, M_G\big(\Phi^{(l+1)}(\mathbf{f}^{(l)}, \mathbf{f}^{(l)})\big)(X_i) \right) \right. \\
&\qquad \left. - \left( (S^X f^{(l)})_i, M_G\big(\Phi^{(l+1)}(S^X f^{(l)}, S^X f^{(l)})\big)(X_i) \right) \right\|_{\infty}^2 \\
&\leq \frac{1}{N} (L_{\Psi^{(l+1)}})^2 \left( \sum_{i=1}^{N} \left\| \mathbf{f}_i^{(l)} - (S^X f^{(l)})_i \right\|_{\infty}^2 \right. \\
&\quad + \sum_{i=1}^{N} \left\| M_G\big(\Phi^{(l+1)}(\mathbf{f}^{(l)}, \mathbf{f}^{(l)})\big)(X_i) - M_G\big(\Phi^{(l+1)}(S^X f^{(l)}, S^X f^{(l)})\big)(X_i) \right\|_{\infty}^2 \right) \\
&\leq (L_{\Psi^{(l+1)}})^2 \left( \big(\operatorname{dist}(\mathbf{f}^{(l)}, f^{(l)})\big)^2 \right. \\
&\quad + \frac{1}{N} \sum_{i=1}^{N} \left\| M_G\big(\Phi^{(l+1)}(\mathbf{f}^{(l)}, \mathbf{f}^{(l)})\big)(X_i) - M_G\big(\Phi^{(l+1)}(S^X f^{(l)}, S^X f^{(l)})\big)(X_i) \right\|_{\infty}^2 \right) \\
&\leq (L_{\Psi^{(l+1)}})^2 \left( (\varepsilon^{(l)})^2 \right. \\
&\quad + \frac{1}{N} \sum_{i=1}^{N} \left\| M_G\big(\Phi^{(l+1)}(\mathbf{f}^{(l)}, \mathbf{f}^{(l)})\big)(X_i) - M_G\big(\Phi^{(l+1)}(S^X f^{(l)}, S^X f^{(l)})\big)(X_i) \right\|_{\infty}^2 \right).
\end{aligned}
\tag{40}
$$

Now, for every $i = 1, \ldots, N$, we have

$$\left\| M_G\Big(\Phi^{(l+1)}\big(\mathbf{f}^{(l)}, \mathbf{f}^{(l)}\big)\Big)(X_i) - M_G\Big(\Phi^{(l+1)}\big(S^X f^{(l)}, S^X f^{(l)}\big)\Big)(X_i) \right\|_\infty^2$$

$$= \left\| \frac{1}{N} \sum_{j=1}^N \frac{W(X_i, X_j)}{\mathrm{d}_X(X_i)} \Phi^{(l+1)}\big(\mathbf{f}^{(l)}(X_i), \mathbf{f}^{(l)}(X_j)\big) \right.$$

$$\left. - \frac{1}{N} \sum_{j=1}^N \frac{W(X_i, X_j)}{\mathrm{d}_X(X_i)} \Phi^{(l+1)}\big(S^X f^{(l)}(X_i), S^X f^{(l)}(X_j)\big) \right\|_\infty^2$$

$$= \left\| \frac{1}{N} \sum_{j=1}^N \frac{W(X_i, X_j)}{\mathrm{d}_X(X_i)} \Big(\Phi^{(l+1)}\big(\mathbf{f}^{(l)}(X_i), \mathbf{f}^{(l)}(X_j)\big) - \Phi^{(l+1)}\big(S^X f^{(l)}(X_i), S^X f^{(l)}(X_j)\big)\Big) \right\|_\infty^2$$

$$\leq \frac{1}{N^2} \sum_{j=1}^N \left| \frac{W(X_i, X_j)}{\mathrm{d}_X(X_i)} \right|^2 \sum_{j=1}^N \left\| \Big(\Phi^{(l+1)}\big(\mathbf{f}^{(l)}(X_i), \mathbf{f}^{(l)}(X_j)\big) - \Phi^{(l+1)}\big(S^X f^{(l)}(X_i), S^X f^{(l)}(X_j)\big)\Big) \right\|_\infty^2$$

$$\leq \frac{4\|W\|_\infty^2}{\mathrm{d}_{\min}^2} \frac{1}{N} \sum_{j=1}^N \left\| \Big(\Phi^{(l+1)}\big(\mathbf{f}^{(l)}(X_i), \mathbf{f}^{(l)}(X_j)\big) - \Phi^{(l+1)}\big(S^X f^{(l)}(X_i), S^X f^{(l)}(X_j)\big)\Big) \right\|_\infty^2,$$

$$(41)$$

where the second-to-last inequality holds by the Cauchy–Schwarz inequality and the last inequality holds by (38). Now, for the term on the right-hand-side of (41), we have

$$\frac{1}{N} \sum_{j=1}^N \left\| \Phi^{(l+1)}\big(\mathbf{f}^{(l)}(X_i), \mathbf{f}^{(l)}(X_j)\big) - \Phi^{(l+1)}\big(S^X f^{(l)}(X_i), S^X f^{(l)}(X_j)\big) \right\|_\infty^2$$

$$\leq (L_{\Phi^{(l+1)}})^2 \frac{1}{N} \sum_{j=1}^N \left( \left\| \mathbf{f}^{(l)}(X_i) - S^X f^{(l)}(X_i) \right\|_\infty^2 + \left\| \mathbf{f}^{(l)}(X_j) - S^X f^{(l)}(X_j) \right\|_\infty^2 \right) \qquad (42)$$

$$\leq (L_{\Phi^{(l+1)}})^2 \left\| \mathbf{f}^{(l)}(X_i) - S^X f^{(l)}(X_i) \right\|_\infty^2 + (L_{\Phi^{(l+1)}})^2 (\varepsilon^{(l)})^2.$$

Hence, by inserting (42) into (41) and (41) into (40), we have

$$\|\Lambda_{\Theta_G}^{(l+1)}(\mathbf{f}^{(l)}) - \Lambda_{\Theta_G}^{(l+1)}(S^X f^{(l)})\|_{2;\infty}^2$$

$$\leq (L_{\Psi^{(l+1)}})^2 \Big( (\varepsilon^{(l)})^2 + \frac{1}{N} \sum_{i=1}^N \big\| M_G\big(\Phi^{(l)}(\mathbf{f}^{(l)}, \mathbf{f}^{(l)})\big)(X_i) - M_G\big(\Phi^{(l)}(S^X f^{(l)}, S^X f^{(l)})\big)(X_i) \big\|_\infty^2 \Big)$$

$$\leq (L_{\Psi^{(l+1)}})^2 \Big( (\varepsilon^{(l)})^2 + \frac{4\|W\|_\infty^2}{\mathrm{d}_{\min}^2} (L_{\Phi^{(l+1)}})^2 \big(\frac{1}{N} \sum_{i=1}^N \big\| \mathbf{f}^{(l)}(X_i) - S^X f^{(l)}(X_i) \big\|_\infty^2 + (\varepsilon^{(l)})^2 \big) \Big)$$

$$\leq (L_{\Psi^{(l+1)}})^2 \Big( (\varepsilon^{(l)})^2 + \frac{4\|W\|_\infty^2}{\mathrm{d}_{\min}^2} (L_{\Phi^{(l+1)}})^2 \big( (\varepsilon^{(l)})^2 + (\varepsilon^{(l)})^2 \big) \Big)$$

$$\leq (L_{\Psi^{(l+1)}})^2 \Big( (\varepsilon^{(l)})^2 + \frac{8\|W\|_\infty^2}{\mathrm{d}_{\min}^2} (L_{\Phi^{(l+1)}})^2 (\varepsilon^{(l)})^2 \Big).$$

By inserting this into (39), we conclude

$$\mathrm{dist}(\Theta_G^{(l+1)}(\mathbf{f}), \Theta_W^{(l+1)}(f)) \leq (L_{\Psi^{(l+1)}})^2 \big(1 + \frac{8\|W\|_\infty^2}{\mathrm{d}_{\min}^2} (L_{\Phi^{(l+1)}})^2\big)(\varepsilon^{(l)})^2 + Q^{(l+1)}.$$

$$\square$$

**Corollary B.14.** *Let $(\chi, d, \mu)$ be a metric-measure space and $W$ be a kernel s.t. Assumptions A.10.1-6. are satisfied. Let $p \in (0, \frac{1}{2})$. Consider a graph $(G, \mathbf{f}) \sim (W, f)$ with $N$ nodes and corresponding graph features, where $N$ satisfies (14). If the event $\mathcal{F}_{\mathrm{Lip}}^p$ from Lemma B.5 occurs, then, for every MPNN $\Theta$ satisfying Assumption A.10.7. and every Lipschitz continuous $f : \chi \to \mathbb{R}^F$ with Lipschitz constant $L_f$,*

$$\mathrm{dist}\big(\Theta_G(f(X)), \Theta_W(f)\big) \leq \sum_{l=1}^T Q^{(l)} \prod_{l'=l+1}^T K^{(l')},$$

where $Q^{(l)}$ and $K^{(l')}$ are defined in ([35](#)) and ([36](#)), respectively.

*Proof.* By Lemma [B.13](#), for every MPNN $\Theta$ satisfying Assumption [A.10.7](#). and every Lipschitz continuous $f : \chi \to \mathbb{R}^F$ with Lipschitz constant $L_f$, the recurrence relation

$$\varepsilon^{(l+1)} \le K^{(l+1)}\varepsilon^{(l)} + Q^{(l+1)}$$

holds for $l = 0, \dots, T-1$. We use that $\varepsilon^{(0)} = 0$ and $\varepsilon^{(T)} = \text{dist}\big(\Theta_G(f(X)), \Theta_W(f)\big)$, and solve this recurrence relation by Lemma [B.11](#) to finish the proof. $\square$

**Theorem B.15.** *Let $(\chi, d, \mu)$ be a metric-measure space and $W$ be a kernel s.t. Assumptions [A.10.1-6](#). are satisfied. Let $p \in (0, \frac{1}{2})$. Consider a graph $(G, \mathbf{f}) \sim (W, f)$ with $N$ nodes and corresponding graph features, where $N$ satisfies ([14](#)). If the event $\mathcal{F}^p_{\text{Lip}}$ from Lemma [B.5](#) occurs, then for every MPNN $\Theta$ satisfying Assumption [A.10.7](#) and $f : \chi \to \mathbb{R}^F$ with Lipschitz constant $L_f$,*

$$\begin{aligned}
&\text{dist}\big(\Theta_G(f(X)), \Theta_W(f)\big) \\
&\le N^{-\frac{1}{2}} \left(\Omega_1 + \Omega_2 \log(2/p) + \Omega_3\|f\|_\infty + \Omega_4\|f\|_\infty \log(2/p)\right) \\
&\quad + N^{-\frac{1}{2(D_\chi+1)}} \left(\Omega_5 + \Omega_6\|f\|_\infty + \Omega_7 L_f\right) \\
&\quad + N^{-\frac{1}{2(D_\chi+1)}} \sqrt{\log(C_\chi) + \frac{D_\chi}{2(D_\chi+1)} \log(N) + \log(2/p)} \cdot (\Omega_8 + \Omega_9\|f\|_\infty),
\end{aligned}$$

*where $\Omega_i$, for $i = 1, \dots, 9$, are constants of the MPNN $\Theta$, defined in ([48](#)), which depend only on the Lipschitz constants of the message and update functions $\{L_{\Phi^{(l)}}, L_{\Psi^{(l)}}\}_{l=1}^T$, and the formal biases $\{\|\Phi^{(l)}(0,0)\|_\infty\}_{l=1}^T$.*

*Proof.* In the event $\mathcal{F}^p_{\text{Lip}}$, by Corollary [B.14](#), for every MPNN $\Theta$ satisfying Assumption [A.10.7](#). and $f : \chi \to \mathbb{R}^F$ with Lipschitz constant $L_f$,

$$\text{dist}\big(\Theta_G(f(X)), \Theta_W(f)\big) \le \sum_{l=1}^T Q^{(l)} \prod_{l'=l+1}^T K^{(l')}, \tag{43}$$

where

$$\begin{aligned}
Q^{(l)} = L_{\Psi^{(l)}} \Bigg( & 4\frac{\varepsilon_1}{\sqrt{N}\text{d}_{min}^2} \|W\|_\infty (L_{\Phi^{(l)}}\|f^{(l-1)}\|_\infty + \|\Phi^{(l)}(0,0)\|_\infty) \\
& + N^{-\frac{1}{2(D_\chi+1)}} \bigg( 2\Big(\frac{\|W\|_\infty}{\text{d}_{min}} L_{\Phi^{(l)}} L_{f^{(l-1)}} + \frac{L_W}{\text{d}_{min}}\big(L_{\Phi^{(l)}}\|f^{(l-1)}\|_\infty + \|\Phi^{(l)}(0,0)\|_\infty\big)\Big) \\
& + \frac{C_\chi}{\sqrt{2}} \Big(\frac{\|W\|_\infty}{\text{d}_{min}}\big(L_{\Phi^{(l)}}\|f^{(l-1)}\|_\infty + \|\Phi^{(l)}(0,0)\|_\infty\big)\Big) \\
& \cdot \sqrt{\log(C_\chi) + \frac{D_\chi}{2(D_\chi+1)} \log(N) + \log(2/p)} \bigg) \Bigg),
\end{aligned}$$

and

$$(K^{(l')})^2 = (L_{\Psi^{(l')}})^2 + \frac{8\|W\|_\infty^2}{\text{d}_{\min}^2} (L_{\Phi^{(l')}})^2 (L_{\Psi^{(l')}})^2.$$

We plug the definition of $Q^{(l)}$ into the right-hand-side of (43), to get

$$\text{dist}\big(\Theta_G(f(X)), \Theta_W(f)\big)$$
$$\leq \sum_{l=1}^{T} L_{\Psi^{(l)}} \left( 4\frac{\varepsilon_1}{\sqrt{N}\mathrm{d}_{min}^2} \|W\|_\infty (L_{\Phi^{(l)}}\|f^{l-1}\|_\infty + \|\Phi^{(l)}(0,0)\|_\infty) \right.$$
$$+ N^{-\frac{1}{2(D_\chi+1)}} \Big( \frac{2\|W\|_\infty}{\mathrm{d}_{min}} L_{\Phi^{(l)}} L_{f^{(l-1)}} + \frac{2L_W}{\mathrm{d}_{min}}(L_{\Phi^{(l)}}\|f^{(l-1)}\|_\infty + \|\Phi^{(l)}(0,0)\|_\infty)$$
$$+ \frac{C_\chi}{\sqrt{2}}\Big(\frac{\|W\|_\infty}{\mathrm{d}_{min}}(L_{\Phi^{(l)}}\|f^{(l-1)}\|_\infty + \|\Phi^{(l)}(0,0)\|_\infty)\Big)$$
$$\left. \sqrt{\log(C_\chi) + \frac{D_\chi}{2(D_\chi+1)}\log(N) + \log(2/p)}\Big) \right) \prod_{l'=l+1}^{T} K^{(l')}. \tag{44}$$

By Lemma B.7, we have

$$\|f^{(l)}\|_\infty \leq B_1^{(l)} + B_2^{(l)}\|f\|_\infty, \tag{45}$$

where $B_1^{(l)}, B_2^{(l)}$ are independent of $f$. Furthermore, we have

$$L_{f^{(l)}} \leq Z_1^{(l)} + Z_2^{(l)}\|f\|_\infty + Z_3^{(l)}L_f, \tag{46}$$

where $Z_1^{(l)}, Z_2^{(l)}$ and $Z_3^{(l)}$ are independent of $f$, and defined in (31). We plug the bound of $L_{f^{(l-1)}}$ from (46) into (43)

$$\text{dist}\big(\Theta_G(f(X)), \Theta_W(f)\big)$$
$$\leq \sum_{l=1}^{T} L_{\Psi^{(l)}} \left( 4\frac{\varepsilon_1}{\sqrt{N}\mathrm{d}_{min}^2} \|W\|_\infty (L_{\Phi^{(l)}}\|f^{(l-1)}\|_\infty + \|\Phi^{(l)}(0,0)\|_\infty) + N^{-\frac{1}{2(D_\chi+1)}} \right.$$
$$\cdot \Big( \frac{2\|W\|_\infty}{\mathrm{d}_{min}} L_{\Phi^{(l)}}(Z_1^{(l-1)} + Z_2^{(l-1)}\|f\|_\infty + Z_3^{(l-1)}L_f) + \frac{2L_W}{\mathrm{d}_{min}}(L_{\Phi^{(l)}}\|f^{(l-1)}\|_\infty + \|\Phi^{(l)}(0,0)\|_\infty)$$
$$+ \frac{C_\chi}{\sqrt{2}}\Big(\frac{\|W\|_\infty}{\mathrm{d}_{min}}(L_{\Phi^{(l)}}\|f^{(l-1)}\|_\infty + \|\Phi^{(l)}(0,0)\|_\infty)\Big)$$
$$\left. \cdot \sqrt{\log(C_\chi) + \frac{D_\chi}{2(D_\chi+1)}\log(N) + \log(2/p)}\Big) \right) \prod_{l'=l+1}^{T} K^{(l')}.$$

We insert the bound of $\|f^{(l-1)}\|_\infty$ from (45) in the above expression, to get

$$\leq \sum_{l=1}^{T} L_{\Psi^{(l)}} \left( 4\frac{\varepsilon_1}{\sqrt{N}\mathrm{d}_{min}^2} \|W\|_\infty \big(L_{\Phi^{(l)}}(B_1^{(l-1)} + B_2^{(l-1)}\|f\|_\infty) + \|\Phi^{(l)}(0,0)\|_\infty\big) + N^{-\frac{1}{2(D_\chi+1)}} \right.$$
$$\cdot \Big( \frac{2\|W\|_\infty}{\mathrm{d}_{min}} L_{\Phi^{(l)}}(Z_1^{(l-1)} + Z_2^{(l-1)}\|f\|_\infty + Z_3^{(l-1)}L_f) + \frac{2L_W}{\mathrm{d}_{min}}(B_1^{(l-1)} + B_2^{(l-1)}\|f\|_\infty)$$
$$+ \frac{C_\chi}{\sqrt{2}}\Big(\frac{\|W\|_\infty}{\mathrm{d}_{min}}\big(L_{\Phi^{(l)}}(B_1^{(l-1)} + B_2^{(l-1)}\|f\|_\infty) + \|\Phi^{(l)}(0,0)\|_\infty\big)\Big)$$
$$\left. \sqrt{\log(C_\chi) + \frac{D_\chi}{2(D_\chi+1)}\log(N) + \log(2/p)}\Big) \right) \prod_{l'=l+1}^{T} K^{(l')}. \tag{47}$$

We insert the bound for $\varepsilon_1$, defined in (22) as

$$\varepsilon_1 = \zeta\Big(L_W\big(\sqrt{\log(C_\chi)} + \sqrt{D_\chi}\big) + \big(\sqrt{2}\|W\|_\infty + L_W\big)\sqrt{\log 2/p}\Big),$$

into ([47](#)) to get

$$
\begin{aligned}
\leq \sum_{l=1}^{T} L_{\Psi^{(l)}} \Bigg( & 4 \frac{\zeta \big( L_W \big( \sqrt{\log(C_\chi)} + \sqrt{D_\chi} \big) + \big( \sqrt{2}\|W\|_\infty + L_W \big) \sqrt{\log 2/p} \big)}{\sqrt{N} \mathrm{d}_{min}^2} \\
& \cdot \|W\|_\infty \big( L_{\Phi^{(l)}} (B_1^{(l-1)} + B_2^{(l-1)} \|f\|_\infty) + \|\Phi^{(l)}(0,0)\|_\infty \big) \\
& + N^{-\frac{1}{2(D_\chi+1)}} \Big( \frac{2\|W\|_\infty}{\mathrm{d}_{min}} L_{\Phi^{(l)}} (Z_1^{(l-1)} + Z_2^{(l-1)} \|f\|_\infty + Z_3^{(l-1)} L_f) \\
& + \frac{2 L_W}{\mathrm{d}_{min}} \big( B_1^{(l-1)} + B_2^{(l-1)} \|f\|_\infty \big) \\
& + \frac{C_\chi}{\sqrt{2}} \Big( \frac{\|W\|_\infty}{\mathrm{d}_{min}} \big( L_{\Phi^{(l)}} (B_1^{(l-1)} + B_2^{(l-1)} \|f\|_\infty) + \|\Phi^{(l)}(0,0)\|_\infty \big) \\
& \cdot \sqrt{\log(C_\chi) + \frac{D_\chi}{2(D_\chi+1)} \log(N) + \log(2/p)} \Big) \Bigg) \prod_{l'=l+1}^{T} K^{(l')}.
\end{aligned}
$$

Then, rearranging the terms yields

$$
\begin{aligned}
= & \sum_{l=1}^{T} L_{\Psi^{(l)}} 4 \frac{\zeta L_W \big( \sqrt{\log(C_\chi)} + \sqrt{D_\chi} \big)}{\sqrt{N} \mathrm{d}_{min}^2} \|W\|_\infty \big( L_{\Phi^{(l)}} B_1^{(l-1)} + \|\Phi^{(l)}(0,0)\|_\infty \big) \prod_{l'=l+1}^{T} K^{(l')} \\
+ & \sum_{l=1}^{T} L_{\Psi^{(l)}} 4 \frac{\zeta (\sqrt{2}\|W\|_\infty + L_W) \sqrt{\log 2/p}}{\sqrt{N} \mathrm{d}_{min}^2} \|W\|_\infty \big( L_{\Phi^{(l)}} B_1^{(l-1)} + \|\Phi^{(l)}(0,0)\|_\infty \big) \prod_{l'=l+1}^{T} K^{(l')} \\
+ & \sum_{l=1}^{T} L_{\Psi^{(l)}} 4 \frac{\zeta L_W \big( \sqrt{\log(C_\chi)} + \sqrt{D_\chi} \big)}{\sqrt{N} \mathrm{d}_{min}^2} \|W\|_\infty \big( L_{\Phi^{(l)}} B_2^{(l-1)} \|f\|_\infty \big) \prod_{l'=l+1}^{T} K^{(l')} \\
+ & \sum_{l=1}^{T} L_{\Psi^{(l)}} 4 \frac{\zeta (\sqrt{2}\|W\|_\infty + L_W) \sqrt{\log 2/p}}{\sqrt{N} \mathrm{d}_{min}^2} \|W\|_\infty \big( L_{\Phi^{(l)}} B_2^{(l-1)} \|f\|_\infty \big) \prod_{l'=l+1}^{T} K^{(l')} \\
+ & \sum_{l=1}^{T} L_{\Psi^{(l)}} N^{-\frac{1}{2(D_\chi+1)}} \Big( \frac{2\|W\|_\infty}{\mathrm{d}_{min}} L_{\Phi^{(l)}} Z_1^{(l-1)} + \frac{2 L_W}{\mathrm{d}_{min}} B_1^{(l-1)} \Big) \prod_{l'=l+1}^{T} K^{(l')} \\
+ & \sum_{l=1}^{T} L_{\Psi^{(l)}} N^{-\frac{1}{2(D_\chi+1)}} \Big( \frac{2\|W\|_\infty}{\mathrm{d}_{min}} L_{\Phi^{(l)}} Z_2^{(l-1)} \|f\|_\infty + \frac{2 L_W}{\mathrm{d}_{min}} B_2^{(l-1)} \|f\|_\infty \Big) \prod_{l'=l+1}^{T} K^{(l')} \\
+ & \sum_{l=1}^{T} L_{\Psi^{(l)}} N^{-\frac{1}{2(D_\chi+1)}} 2 \frac{\|W\|_\infty}{\mathrm{d}_{min}} L_{\Phi^{(l)}} Z_3^{(l-1)} L_f \prod_{l'=l+1}^{T} K^{(l')} \\
+ & \sum_{l=1}^{T} L_{\Psi^{(l)}} N^{-\frac{1}{2(D_\chi+1)}} \frac{C_\chi}{\sqrt{2}} \frac{\|W\|_\infty}{\mathrm{d}_{min}} (L_{\Phi^{(l)}} B_1^{(l-1)} + \|\Phi^{(l)}(0,0)\|_\infty) \\
& \cdot \sqrt{\log(C_\chi) + \frac{D_\chi}{2(D_\chi+1)} \log(N) + \log(2/p)} \Big) \prod_{l'=l+1}^{T} K^{(l')} \\
+ & \sum_{l=1}^{T} L_{\Psi^{(l)}} N^{-\frac{1}{2(D_\chi+1)}} \frac{C_\chi}{\sqrt{2}} \frac{\|W\|_\infty}{\mathrm{d}_{min}} L_{\Phi^{(l)}} B_2^{(l-1)} \|f\|_\infty \\
& \cdot \sqrt{\log(C_\chi) + \frac{D_\chi}{2(D_\chi+1)} \log(N) + \log(2/p)} \Big) \prod_{l'=l+1}^{T} K^{(l')}
\end{aligned}
$$

$$
\begin{aligned}
=: \ & \Omega_1 \frac{1}{\sqrt{N}} + \Omega_2 \frac{\log(2/p)}{\sqrt{N}} + \Omega_3 \frac{\|f\|_\infty}{\sqrt{N}} + \Omega_4 \frac{\|f\|_\infty \log(2/p)}{\sqrt{N}} \\
& + N^{-\frac{1}{2(D_\chi+1)}} \left( \Omega_5 + \Omega_6 \|f\|_\infty + \Omega_7 L_f \right) \\
& + N^{-\frac{1}{2(D_\chi+1)}} \sqrt{\log(C_\chi) + \frac{D_\chi}{2(D_\chi+1)} \log(N) + \log(2/p)} \cdot (\Omega_8 + \Omega_9 \|f\|_\infty),
\end{aligned}
$$

where we define

$$
\Omega_1 = \sum_{l=1}^{T} L_{\Psi^{(l)}} 4 \frac{\zeta L_W \left( \sqrt{\log(C_\chi)} + \sqrt{D_\chi} \right)}{\mathrm{d}_{min}^2} \|W\|_\infty \left( L_{\Phi^{(l)}} B_1^{(l-1)} + \|\Phi^{(l)}(0,0)\|_\infty \right) \prod_{l'=l+1}^{T} K^{(l')}
$$

$$
\Omega_2 = \sum_{l=1}^{T} L_{\Psi^{(l)}} 4 \frac{\zeta (\sqrt{2}\|W\|_\infty + L_W)}{\mathrm{d}_{min}^2} \|W\|_\infty \left( L_{\Phi^{(l)}} B_1^{(l-1)} + \|\Phi^{(l)}(0,0)\|_\infty \right) \prod_{l'=l+1}^{T} K^{(l')}
$$

$$
\Omega_3 = \sum_{l=1}^{T} L_{\Psi^{(l)}} 4 \frac{\zeta L_W \left( \sqrt{\log(C_\chi)} + \sqrt{D_\chi} \right)}{\mathrm{d}_{min}^2} \|W\|_\infty \left( L_{\Phi^{(l)}} B_2^{(l-1)} \right) \prod_{l'=l+1}^{T} K^{(l')}
$$

$$
\Omega_4 = \sum_{l=1}^{T} L_{\Psi^{(l)}} 4 \frac{\zeta (\sqrt{2}\|W\|_\infty + L_W) \sqrt{\log 2/p}}{\mathrm{d}_{min}^2} \|W\|_\infty \left( L_{\Phi^{(l)}} B_2^{(l-1)} \right) \prod_{l'=l+1}^{T} K^{(l')}
$$

$$
\Omega_5 = \sum_{l=1}^{T} L_{\Psi^{(l)}} \left( \frac{2\|W\|_\infty}{\mathrm{d}_{min}} L_{\Phi^{(l)}} Z_1^{(l-1)} + \frac{2 L_W}{\mathrm{d}_{min}} B_1^{(l-1)} \right) \prod_{l'=l+1}^{T} K^{(l')}
$$

$$
\Omega_6 = \sum_{l=1}^{T} L_{\Psi^{(l)}} \left( \frac{2\|W\|_\infty}{\mathrm{d}_{min}} L_{\Phi^{(l)}} Z_2^{(l-1)} + \frac{2 L_W}{\mathrm{d}_{min}} B_2^{(l-1)} \right) \prod_{l'=l+1}^{T} K^{(l')}
$$

$$
\Omega_7 = \sum_{l=1}^{T} L_{\Psi^{(l)}} 2 \frac{\|W\|_\infty}{\mathrm{d}_{min}} L_{\Phi^{(l)}} Z_3^{(l-1)} \prod_{l'=l+1}^{T} K^{(l')}
$$

$$
\Omega_8 = \sum_{l=1}^{T} L_{\Psi^{(l)}} \frac{C_\chi}{\sqrt{2}} \frac{\|W\|_\infty}{\mathrm{d}_{min}} \left( L_{\Phi^{(l)}} B_1^{(l-1)} + \|\Phi^{(l)}(0,0)\|_\infty \right) \prod_{l'=l+1}^{T} K^{(l')}
$$

$$
\Omega_9 = \sum_{l=1}^{T} L_{\Psi^{(l)}} \frac{C_\chi}{\sqrt{2}} \frac{\|W\|_\infty}{\mathrm{d}_{min}} L_{\Phi^{(l)}} B_2^{(l-1)} \prod_{l'=l+1}^{T} K^{(l')},
$$

(48)

where $Z_1^{(l-1)}, Z_2^{(l-1)}, Z_3^{(l-1)}$ are defined in (31), $B_1^{(l-1)}$ and $B_2^{(l-1)}$ are defined in (27) and (28), and

$$
K^{(l')} = \sqrt{(L_{\Psi^{(l')}})^2 + \frac{8\|W\|_\infty^2}{\mathrm{d}_{min}^2} (L_{\Phi^{(l')}})^2 (L_{\Psi^{(l')}})^2}.
$$

$\square$

Next we study the convergence of MPNNs after global pooling. We give the following lemma.

**Lemma B.16.** *Let $(\chi, d, \mu)$ be a metric-measure space and $W$ be a kernel s.t. Assumptions A.10.1-6. are satisfied. Suppose that $X_1, \ldots, X_N$ are drawn i.i.d. from $\mu$ on $\chi$ such that $(X_1, \ldots, X_N) \in \mathcal{E}_{\mathrm{Lip}}^p$, where the event $\mathcal{E}_{\mathrm{Lip}}^p$ is defined in Lemma B.3. Then, for every MPNN $\Theta$ satisfying Assumption A.10.7*

and $f : \chi \to \mathbb{R}^F$ with Lipschitz constant $L_f$,

$$\left\| \frac{1}{N} \sum_{i=1}^{N} \left( S^X \Theta_W(f) \right)(X_i) - \int_\chi \Theta_W(f)(y) d\mu(y) \right\|_\infty$$

$$\leq N^{-\frac{1}{2(D_\chi+1)}} \left( 2(Z_1^{(T)} + Z_2^{(T)} \|f\|_\infty + Z_3^{(T)} L_f) + \frac{C_\chi}{\sqrt{2}} (B_1^{(T)} + B_2^{(T)} \|f\|_\infty) \right. \tag{49}$$

$$\left. \cdot \sqrt{\log(C_\chi) + \frac{D_\chi}{2(D_\chi+1)} \log(N) + \log(2/p)} \right).$$

Here, $Z_1^{(T)}, Z_2^{(T)}, Z_3^{(T)}$ and $B_1^{(T)}, B_2^{(T)}$ are defined in (45) and (46).

*Proof.* By Lemma B.7, we have

$$\|\Theta_W^{(T)}(f)\|_\infty \leq B_1^{(T)} + \|f\|_\infty B_2^{(T)}$$

and, by Corollary B.8, we have

$$L_{\Theta_W^{(T)}(f)} \leq Z_1^{(T)} + Z_2^{(T)} \|f\|_\infty + Z_3^{(T)} L_f$$

for all MPNNs $\Theta$ and metric-space signals $f$ considered. Hence, by Lemma B.3, equation (49) holds. $\qquad\square$

**Corollary B.17.** *Let $(\chi, d, \mu)$ be a metric-measure space and $W$ be a kernel s.t. Assumptions A.10.1-6. are satisfied. Consider a graph $(G, \mathbf{f}) \sim (W, f)$ with $N$ nodes and corresponding graph features, where $N$ satisfies (14). If the event $\mathcal{F}_{\text{Lip}}^p$ from Lemma B.5 occurs, then for every MPNN $\Theta$ satisfying Assumption A.10.7 and every $f : \chi \to \mathbb{R}^F$ with Lipschitz constant $L_f$,*

$$\left\| \Theta_G^P(\mathbf{f}) - \Theta_W^P(f) \right\|_\infty^2 \leq \frac{S_1 + S_2 \|f\|_\infty^2}{N} + \frac{R_1 + R_2 \|f\|_\infty^2 + R_3 L_f^2}{N^{\frac{1}{D_\chi+1}}} + \frac{T_1 + T_2 \|f\|_\infty^2}{N^{\frac{1}{D_\chi+1}}} \log(N)$$

$$+ \frac{S_3 + S_4 \|f\|_\infty^2}{N} \log^2(2/p) + \frac{R_4 + R_5 \|f\|_\infty^2}{N^{\frac{1}{D_\chi+1}}} \log(2/p),$$

*where the constants are defined in (51) below.*

*Proof.* We have

$$\left\|\Theta_G^P(\mathbf{f}) - \Theta_W^P(f)\right\|_\infty$$

$$= \left\|\frac{1}{N}\sum_{i=1}^N \Theta_G(\mathbf{f})(X_i) - \int_\chi \Theta_W(f)(y)d\mu(y)\right\|_\infty$$

$$\leq \left\|\frac{1}{N}\sum_{i=1}^N \Theta_G(\mathbf{f})(X_i) - \frac{1}{N}\sum_{i=1}^N \left(S^X\Theta_W(f)\right)(X_i)\right\|_\infty$$

$$+ \left\|\frac{1}{N}\sum_{i=1}^N \left(S^X\Theta_W(f)\right)(X_i) - \int_\chi \Theta_W(f)(y)d\mu(y)\right\|_\infty$$

$$\leq \frac{1}{N}\sum_{i=1}^N \left\|\Theta_G(\mathbf{f})(X_i) - \left(S^X\Theta_W(f)\right)(X_i)\right\|_\infty$$

$$+ \left\|\frac{1}{N}\sum_{i=1}^N \left(S^X\Theta_W(f)\right)(X_i) - \int_\chi \Theta_W(f)(y)d\mu(y)\right\|_\infty$$

$$\leq \frac{1}{N}\sum_{i=1}^N \left\|\Theta_G(\mathbf{f})(X_i) - \left(S^X\Theta_W(f)\right)(X_i)\right\|_\infty$$

$$+ N^{-\frac{1}{2(D_\chi+1)}}\left(2(Z_1^{(T)} + Z_2^{(T)}\|f\|_\infty + Z_3^{(T)}L_f) + \frac{C_\chi}{\sqrt{2}}(B_1^{(T)} + B_2^{(T)}\|f\|_\infty)\right.$$

$$\left.\cdot\sqrt{\log(C_\chi) + \frac{D_\chi}{2(D_\chi+1)}\log(N) + \log(2/p)}\right)$$

$$= \mathrm{dist}\left(\Theta_G(\mathbf{f}), \Theta_W(f)\right)$$

$$+ N^{-\frac{1}{2(D_\chi+1)}}\left(2(Z_1^{(T)} + Z_2^{(T)}\|f\|_\infty + Z_3^{(T)}L_f) + \frac{C_\chi}{\sqrt{2}}(B_1^{(T)} + B_2^{(T)}\|f\|_\infty)\right.$$

$$\left.\cdot\sqrt{\log(C_\chi) + \frac{D_\chi}{2(D_\chi+1)}\log(N) + \log(2/p)}\right),$$

where the last inequality holds by Lemma B.16. Together with Theorem B.15, we get

$$\left\|\Theta_G^P(\mathbf{f}) - \Theta_W^P(f)\right\|_\infty$$

$$\leq \frac{\Omega_1 + \Omega_2\log(2/p) + \Omega_3\|f\|_\infty + \Omega_4\|f\|_\infty\log(2/p)}{N^{\frac{1}{2}}}$$

$$+ \frac{\Omega_5 + \Omega_6\|f\|_\infty + \Omega_7 L_f}{N^{\frac{1}{2(D_\chi+1)}}}$$

$$+ \frac{\Omega_8 + \Omega_9\|f\|_\infty}{N^{\frac{1}{2(D_\chi+1)}}}\sqrt{\log(C_\chi) + \frac{D_\chi}{2(D_\chi+1)}\log(N) + \log(2/p)} \qquad (50)$$

$$+ N^{-\frac{1}{2(D_\chi+1)}}\left(2(Z_1^{(T)} + Z_2^{(T)}\|f\|_\infty + Z_3^{(T)}L_f) + \frac{C_\chi}{\sqrt{2}}(B_1^{(T)} + B_2^{(T)}\|f\|_\infty)\right.$$

$$\left.\cdot\sqrt{\log(C_\chi) + \frac{D_\chi}{2(D_\chi+1)}\log(N) + \log(2/p)}\right).$$

Now we use the inequality

$$\left(\sum_{i=1}^n a_i\right)^2 \leq n\sum_{i=1}^n a_i^2$$

for any $a_i \in \mathbb{R}_+$, $i = 1, \ldots, N$, and square both sides of (50) to get

$$
\begin{aligned}
\big\| &\Theta_G^P(\mathbf{f}) - \Theta_W^P(f) \big\|_\infty^2 \\
&\leq 14 \frac{\Omega_1^2 + \Omega_3^2 \|f\|_\infty^2}{N} + 14 \frac{\Omega_5^2 + \Omega_6^2 \|f\|_\infty^2 + \Omega_7^2 L_f^2}{N^{\frac{1}{D_\chi+1}}} \\
&\quad + 14 \frac{\Omega_8^2 + \Omega_9^2 \|f\|_\infty^2}{N^{\frac{1}{D_\chi+1}}} \left( \log(C_\chi) + \frac{D_\chi}{2(D_\chi+1)} \log(N) \right) \\
&\quad + 56 \frac{(Z_1^{(T)})^2 + (Z_2^{(T)})^2 \|f\|_\infty^2 + (Z_3^{(T)})^2 L_f^2}{N^{\frac{1}{D_\chi+1}}} \\
&\quad + 7 \frac{\left( C_\chi^2 (B_1^{(T)})^2 + C_\chi^2 (B_2^{(T)})^2 \|f\|_\infty^2 \right) \left( \log(C_\chi) + \frac{D_\chi}{2(D_\chi+1)} \log(N) \right)}{N^{\frac{1}{D_\chi+1}}} \\
&\quad + 14 \frac{(\Omega_2^2 + \Omega_4^2 \|f\|_\infty^2) \log^2(2/p)}{N} + 14 \frac{\Omega_8^2 + \Omega_9^2 \|f\|_\infty^2}{N^{\frac{1}{D_\chi+1}}} \log(2/p) \\
&\quad + 7 \frac{(C_\chi^2 (B_1^{(T)})^2 + C_\chi^2 (B_2^{(T)})^2 \|f\|_\infty^2) \log(2/p)}{N^{\frac{1}{D_\chi+1}}} \\
&=: \frac{S_1 + S_2 \|f\|_\infty^2}{N} + \frac{R_1 + R_2 \|f\|_\infty^2 + R_3 L_f^2}{N^{\frac{1}{D_\chi+1}}} + \frac{T_1 + T_2 \|f\|_\infty^2}{N^{\frac{1}{D_\chi+1}}} \log(N) \\
&\quad + \frac{S_3 + S_4 \|f\|_\infty^2}{N} \log^2(2/p) + \frac{R_4 + R_5 \|f\|_\infty^2}{N^{\frac{1}{D_\chi+1}}} \log(2/p),
\end{aligned}
$$

where

$$
\begin{aligned}
S_1 &= 14\Omega_1^2 \\
S_2 &= 14\Omega_3^2 \\
S_3 &= 14\Omega_2^2 \\
S_4 &= 14\Omega_4^2 \\
R_1 &= 14\Omega_5^2 + 14\Omega_8^2 \log(C_\chi) + 56(Z_1^{(T)})^2 + 7C_\chi^2 (B_1^{(T)})^2 \log(C_\chi) \\
R_2 &= 14\Omega_6^2 + 14\Omega_9^2 \log(C_\chi) + 56(Z_2^{(T)})^2 + 7C_\chi^2 (B_2^{(T)})^2 \log(C_\chi) \\
R_3 &= 14\Omega_7^2 + 56(Z_3^{(T)})^2 \\
R_4 &= 14\Omega_8^2 + 7C_\chi^2 (B_1^{(T)})^2 \\
R_5 &= 14\Omega_9^2 + 7C_\chi^2 (B_2^{(T)})^2 \\
T_1 &= 14\Omega_8^2 \frac{D_\chi}{2(D_\chi+1)} + 7C_\chi^2 (B_1^{(T)})^2 \frac{D_\chi}{2(D_\chi+1)} \\
T_2 &= 14\Omega_9^2 \frac{D_\chi}{2(D_\chi+1)} + 7C_\chi^2 (B_2^{(T)})^2 \frac{D_\chi}{2(D_\chi+1)},
\end{aligned}
\tag{51}
$$

and $\Omega_1, \ldots, \Omega_9$ are defined in (48), and $B_1^{(T)}$ and $B_2^{(T)}$ are defined in (27) and (28). $\qquad\square$

We now write a version of Theorem 3.1 (about the convergence error of MPNNs) with detailed constants, and prove it.

**Theorem B.18.** *Let $(\chi, d, \mu)$ be a metric-measure space and $W$ be a kernel s.t. Assumptions A.10.1-6. and Assumptions A.10.8 are satisfied. Consider a graph $(G, \mathbf{f}) \sim (W, f)$ with $N$ nodes and*

*corresponding graph features. Then, for every $f : \chi \to \mathbb{R}^F$ with Lipschitz constant $L_f$,*

$$
\mathbb{E}_{X_1,\ldots,X_N \sim \mu^N}\left[\sup_{\Theta \in \mathrm{Lip}_{L,B}} \left\|\Theta_G^P(\mathbf{f}) - \Theta_W^P(f)\right\|_\infty^2\right]
$$

$$
\leq 6\sqrt{\pi}\left(\frac{S_1 + S_3 + (S_2 + S_4)\|f\|_\infty^2}{N} + \frac{R_1 + R_4 + (R_2 + R_5)\|f\|_\infty^2 + R_3 L_f^2}{N^{\frac{1}{D_\chi + 1}}}\right.
$$

$$
\left. + \frac{(T_1 + T_2\|f\|_\infty^2)\log(N)}{N^{\frac{1}{D_\chi + 1}}}\right) + \mathcal{O}\left(\exp(-N)N^{\frac{3}{2}T - \frac{3}{2}}\right),
$$

*where the constants are defined in (51).*

*Proof.* For any $p > 0$, we have with probability at least $1 - 2p$ for every $\Theta \in \mathrm{Lip}_{L,B}$, by Corollary B.17, that

$$
\left\|\Theta_G^P(\mathbf{f}) - \Theta_W^P(f)\right\|_\infty^2 \leq H_1 + H_2\log(2/p) + H_3\log^2(2/p)
$$

if (14) holds, where

$$
H_1 = \frac{S_1 + S_2\|f\|_\infty^2}{N} + \frac{R_1 + R_2\|f\|_\infty^2 + R_3 L_f^2}{N^{\frac{1}{D_\chi + 1}}} + \frac{T_1 + T_2\|f\|_\infty^2}{N^{\frac{1}{D_\chi + 1}}}\log(N),
$$

$$
H_2 = \frac{R_4 + R_5\|f\|_\infty^2}{N^{\frac{1}{D_\chi + 1}}} \text{ and } H_3 = \frac{S_3 + S_4\|f\|_\infty^2}{N}.
$$

Further, for every $p \in (0, 1/2)$, we consider $k > 0$ such that $p = 2\exp(-k^2)$. This means, if $p$ respectively $k$ satisfies (14), we have with probability at least $1 - 4\exp(-k^2)$ for every $\Theta \in \mathrm{Lip}_{L,B}$,

$$
\left\|\Theta_G^P(\mathbf{f}) - \Theta_W^P(f)\right\|_\infty^2 \leq H_1 + H_2 k + H_3 k^2.
$$

If $k$ does not satisfy (14), we get

$$
k > N_0 = D_1 + D_2\sqrt{N},
$$

where $D_1 \in \mathbb{R}$ and $D_2 > 0$ are the matching constants in (14). By Lemma B.10 and Lemma B.7, we get in this case

$$
\left\|\Theta_G^P(\mathbf{f}) - \Theta_W^P(f)\right\|_\infty^2 = \left\|\frac{1}{N}\sum_{i=1}^N \Theta_G(\mathbf{f})_i - \int_\chi \Theta_W(f)(y)d\mu(y)\right\|_\infty^2
$$

$$
\leq \frac{4}{N}\sum_{i=1}^N \|\Theta_G(\mathbf{f})_i\|_\infty^2 + 2\left\|\int_\chi \Phi_W(f)(y)d\mu(y)\right\|_\infty^2 \qquad (52)
$$

$$
\leq \frac{4}{N}\|\Theta_G(\mathbf{f})\|_{2;\infty}^2 + 2\|\Theta_W(f)\|_\infty^2
$$

$$
\leq \frac{4}{N}N^T(A' + A''\|f\|_\infty^2) + 2(B_1^{(T)} + \|f\|_\infty B_2^{(T)})^2 =: q(N),
$$

where the first inequality holds by applying the triangle inequality and Cauchy-Schwarz.

We then calculate the expected value by partitioning the integral over the event space into the following sum.

$$
\mathbb{E}_{X_1,\ldots,X_N \sim \mu^N}\left[\sup_{\Theta \in \mathrm{Lip}_{L,B}}\left\|\Theta_G^P(\mathbf{f}) - \Theta_W^P(f)\right\|_\infty^2\right]
$$

$$
\leq \sum_{k=0}^{N_0}\mathbb{P}\left(H_1 + H_2 k + H_3 k^2 \leq \sup_{\Theta \in \mathrm{Lip}_{L,B}}\left\|\Theta_G^P(\mathbf{f}) - \Theta_W^P(f)\right\|_\infty^2 < H_1 + H_2(k+1) + H_3(k+1)^2\right)
$$

$$
\cdot \left(H_1 + H_2(k+1) + H_3(k+1)^2\right)
$$

$$
+ \sum_{k=N_0}^\infty \mathbb{P}\left(H_1 + H_2 k + H_3 k^2 \leq \sup_{\Theta \in \mathrm{Lip}_{L,B}}\left\|\Theta_G^P(\mathbf{f}) - \Theta_W^P(f)\right\|_\infty^2 < H_1 + H_2(k+1) + H_3(k+1)^2\right)
$$

$$
\cdot q(N)
$$

(53)

To bound the second sum, note that it is a finite sum, since $\left\|\Theta_G^P(\mathbf{f}) - \Theta_W^P(f)\right\|_\infty^2$ is bounded by $q(N)$, which is defined in (52). The summands are zero if $H_1 + H_2 k + H_3 k^2 > q(N)$, which holds for $k > \sqrt{\frac{q(N)}{H_3}}$. Hence, we calculate with the right-hand-side of (53) by

$$
\begin{aligned}
&\leq 2 \sum_{k=0}^{N_0} 2 \exp(-k^2) \cdot \left(H_1 + H_2(k+1) + H_3(k+1)^2\right) \\
&+ \sum_{k=N_0}^{\left\lceil \sqrt{\frac{q(N)}{H_3}} \right\rceil} 4 \exp(-N_0^2) \cdot q(N) \\
&\leq 2 \int_0^\infty 2 \exp(-k^2) \cdot \left(H_1 + H_2(k+1) + H_3(k+1)^2\right) \\
&+ 4 \exp(-N_0^2) q(N) \left\lceil \sqrt{\frac{q(N)}{H_3}} \right\rceil,
\end{aligned}
\tag{54}
$$

where $q(N) = O(N^{T-1})$ is a polynomial in $N$ as defined above. The first term on the right-hand-side is bounded by using

$$
\int_0^\infty 2(t+1)^2 e^{-t^2} \, dt, \ \int_0^\infty 2(t+1) e^{-t^2} \, dt, \ \int_0^\infty 2 e^{-t^2} \, dt \leq 3\sqrt{\pi}.
$$

For the second term we remember that $N_0 = D_1 + D_2\sqrt{N}$. Hence,

$$
\begin{aligned}
&\mathbb{E}_{X_1,\ldots,X_N \sim \mu^N} \left[ \sup_{\Theta \in \mathrm{Lip}_{L,B}} \left\|\Theta_G^P(\mathbf{f}) - \Theta_W^P(f)\right\|_\infty^2 \right] \\
&\leq 6\sqrt{\pi}(H_1 + H_2 + H_3) + \mathcal{O}(\exp(-N) N^{\frac{3}{2}T - \frac{3}{2}}).
\end{aligned}
$$

$\square$

## C Generalization Analysis

In this section, we provide details on our generalization analysis of MPNNs. In Subsection C.1, we detail the data distribution from the graph classification task, which was introduced in Subsection 2.4. In Subsection C.2, we provide a detailed version and a proof for Theorem 3.3 (about the generalization bound of MPNNs). This is followed by a derivation of the asymptotics of our generalization bound in Subsection C.3 and a comparison of the asymptotics of our generalization bound with other related generalization bounds in Subsection C.4.

### C.1 The Probability Space of the Dataset

Recall that the measure on the space $\chi^j$ is denoted by $\mu^j$. Given a class $j$ and $N \in \mathbb{N}$, the space of graphs with $N$ nodes from class $j$ is defined to be $(\chi^j)^N$. The measure on $(\chi^j)^N$ is defined to be $(\mu^j)^N$, namely, the direct product of the measure $\mu^j$ with itself $N$ times. The space $\mathcal{G}_j$ of graphs of any size, which are sampled from class $j$, is defined to be

$$
\mathcal{G}_j := \bigcup_{n \in \mathbb{N}} (\chi^j)^N.
$$

The measure on $\mathcal{G}_j$ is denoted by $\mu_{\mathcal{G}_j}$, and defined as follows.

**Definition C.1.** *A set of graphs $S \subset \mathcal{G}_j$ is called measurable, if for each $N \in \mathbb{N}$, the restriction*

$$
S_N := \{G \in S \mid G \text{ has } N \text{ nodes}\} \subset (\chi^j)^N
$$

*is measurable with respect to $(\mu^j)^N$. The measure of a measurable set $S \subset \mathcal{G}_j$ is defined to be*

$$
\mu_{\mathcal{G}_j}(S) := \sum_{N=1}^\infty \nu(N)(\mu^j)^N(S_N),
$$

*where $\nu(N)$ is the probability of choosing a graph with $N$ nodes (see Subsection 2.4).*

The space of graphs of either of the classes $j = 1, \ldots, \Gamma$ is defined to be

$$\mathcal{G} := \bigcup_{j=1}^{\Gamma} \mathcal{G}_j.$$

The measure on $\mathcal{G}$ is denoted by $\mu_{\mathcal{G}}$, and defined as follows.

**Definition C.2.** *A set of graphs $S \subset \mathcal{G}$ is called measurable, if for each $j = 1, \ldots, \Gamma$, the restriction*

$$S_j := \{G \in S \mid G \text{ is sampled from class } j\} \subset \mathcal{G}_j$$

*is measurable with respect to $\mu_{\mathcal{G}_j}$. The measure of a measurable $S \subset \mathcal{G}$ is defined to be*

$$\mu_{\mathcal{G}}(S) = \sum_{j=1}^{\Gamma} \gamma_j \mu_{\mathcal{G}_j}(S_j),$$

*where $\gamma_j$ is the probability of choosing class $j$ (see Subsection 2.4).*

With these notations, the space of graph datasets of size $m$ is defined to be $\mathcal{G}^m$ with the direct product measure $\mu_{\mathcal{G}}^m$. We denote a random graph sampled from the space of graphs by $(G, \mathbf{f}, y) \sim \mu_{\mathcal{G}}$. Here, $y$ denotes the class of the graph, namely, the value $y$ such that $(G, \mathbf{f})$ is sampled from class $y$.

The next lemma is direct, and given without proof.

**Lemma C.3.** *The spaces $\{\mathcal{G}, \mu_{\mathcal{G}}\}$ and $\{\mathcal{G}_j, \mu_{\mathcal{G}_j}\}$, $j = 1, \ldots, \Gamma$, are measure spaces, and $\mu_{\mathcal{G}}$ and $\mu_{\mathcal{G}_j}$, $j = 1, \ldots, \Gamma$, are probability measures.*

Let us next derive a re-parameterization of the space of datasets $\mathcal{G}^m$. Given $\mathcal{T} \sim \mu_{\mathcal{G}}^m$, for every $j = 1, \ldots, \Gamma$, let $m_j$ denote the number of graphs in $\mathcal{T}$ that fall into the class $j$. Note that $\mathbf{m} = (m_1, \ldots, m_\Gamma)$ has a multinomial distribution with parameters $m$ and $\boldsymbol{\gamma} = (\gamma_1, \ldots, \gamma_\Gamma)$, which we denote by $\mathrm{MN}_{m,\boldsymbol{\gamma}}$. Conditioning the choice of the graphs on the choice of $\mathbf{m}$, we can formulate the data sampling procedure as first sampling $\mathbf{m}$ from $\mathrm{MN}_{m,\boldsymbol{\gamma}}$, and then sampling $\{G_i^j, \mathbf{f}_i^j\}_{i=1}^{m_j} \sim (\mu_{\mathcal{G}_j})^{m_j}$, $j = 1 \ldots, \Gamma$ independently of each other. Now, the measure $\mu_{\mathcal{G}}^m$ of the space of datasets can be parameterized as follows.

First, we define the following measure space. Let $\mathbf{m} = (m_1, \ldots, m_\Gamma)$ satisfy $\sum_{j=1}^{\Gamma} m_j = m$. We define the space

$$\mathcal{G}^{\mathbf{m}} := \prod_{j=1}^{\Gamma} \mathcal{G}_j^{m_j},$$

with the measure

$$\mu_{\mathcal{G}^{\mathbf{m}}} := \prod_{j=1}^{\Gamma} \mu_{\mathcal{G}_j}^{m_j}. \tag{55}$$

The space $\mathcal{G}^{\mathbf{m}}$ is interpreted as the space of datasets with exactly $m_j$ samples in each class $j$.

We can now show the following parametrization of the measure space $\mathcal{G}^m$ of datasets of size $m$. The lemma is direct, and given without proof.

**Lemma C.4.** *A set of datasets $S \subset \mathcal{G}^m$ is measurable, if and only if for every $\mathbf{m} = (m_1, \ldots, m_\Gamma)$ with $\sum_{j=1}^{\Gamma} m_j = m$, the restriction*

$$S_{\mathbf{m}} = \{\mathcal{T} \in S \mid \forall 1 \leq j \leq \Gamma, \ \mathcal{T} \text{ contains } m_j \text{ graphs from class } j\} \subset \mathcal{G}^{\mathbf{m}}$$

*is measurable with respect to $\mu_{\mathcal{G}^{\mathbf{m}}}$.*

*With these notations, $\mu_{\mathrm{G}}^m$ is decomposed as follows: $\mathcal{G}^m = \bigcup_{\mathbf{m}} \mathcal{G}^{\mathbf{m}}$, and for every measurable set of datasets $S \subset \mathcal{G}^m$,*

$$\mu_{\mathrm{G}}^m(S) = \sum_{\mathbf{m}: \, m_1 + \ldots + m_\Gamma = m} \mu_{\mathrm{MN}_{m,\boldsymbol{\gamma}}}(\mathbf{m}) \sum_{j=1}^{\Gamma} \sum_{i=1}^{m_j} \mu_{\mathrm{G}_j}(S_{\mathbf{m}}).$$

## C.2 Proof of Theorem 3.3

The following corollary computes the expected robustness of a random graph, of arbitrary size, sampled from $\mu_{\mathcal{G}_j}$, and is a direct result of Definition C.1 and Theorem B.18.

**Corollary C.5.** *Let $\{(W^j, f^j)\}$ be a RGM on the corresponding metric-measure space $(\chi^j, d^j, \mu^j)$ that satisfies Assumptions A.10.1.-6. and A.10.8. Let $\mu_{\mathcal{G}_j}$ be the distribution from Definition C.1. Then,*

$$\mathbb{E}_{(G^j, \mathbf{f}^j) \sim \mu_{\mathcal{G}_j}} \Big[ \sup_{\Theta \in \mathrm{Lip}_{L,B}} \big\| \Theta_{G^j}^P(\mathbf{f}^j) - \Theta_{W^j}^P(f^j) \big\|_\infty^2 \Big]$$

$$\leq 6\sqrt{\pi} \Bigg( \big( S_1^{(j)} + S_3^{(j)} + (S_2^{(j)} + S_4^{(j)}) \|f^j\|_\infty^2 \big) \mathbb{E}_{N \sim \nu} \big[ N^{-1} \big]$$

$$+ \big( R_1^{(j)} + R_4^{(j)} + (R_2^{(j)} + R_5^{(j)}) \|f^j\|_\infty^2 + R_3^{(j)} L_{f^j}^2 \big) \mathbb{E}_{N \sim \nu} \Big[ N^{-\frac{1}{D_{\chi^j}+1}} \Big]$$

$$+ \big( T_1^{(j)} + T_2^{(j)} \|f^j\|_\infty^2 \big) \mathbb{E}_{N \sim \nu} \Big[ \log(N) N^{-\frac{1}{D_{\chi^j}+1}} \Big] \Bigg) + \mathcal{O} \Big( \mathbb{E}_{N \sim \nu} \Big[ \exp(-N) N^{\frac{3T}{2}-\frac{3}{2}} \Big] \Big),$$

*where $S_l^{(j)}, R_l^{(j)}, T_l^{(j)}$ are the according constants from Theorem B.18 for each class $j$ and are defined in (51).*

When sampling a dataset $\mathcal{T} \sim p^m$, the numbers of samples $m_j$ that fall in class $\chi^j$, for $j = 1, \ldots, \Gamma$, are distributed multinomially. We hence recall a concentration of measure result for multinomial variables.

**Lemma C.6** (Proposition A.6 in [Vaart and Wellner, 1996], Bretagnolle-Huber-Carol inequality)**.** *If the random vector $(m_1, \ldots m_\Gamma)$ is multinomially distributed with parameters $m$ and $\gamma_1, \ldots, \gamma_\Gamma$, then*

$$\mathbb{P} \left( \sum_{i=1}^{\Gamma} |m_i - m\gamma_i| \geq 2\sqrt{m}\lambda \right) \leq 2^\Gamma \exp(-2\lambda^2)$$

*for any $\lambda > 0$.*

We now write a version of Theorem 3.3 (about the generalization error of MPNNs) with detailed constants, and prove it.

**Theorem C.7.** *Let $\{(W^j, f^j)\}_{j=1}^\Gamma$ be a collection of RGMs on corresponding metric-measure spaces $\{(\chi^j, d^j, \mu^j)\}_{j=1}^\Gamma$ such that each one satisfies Assumptions A.10.1.-6. and A.10.8. Let $\mu_{\mathcal{G}}$ denote the data distribution from Definition C.2. Let $\mathcal{T} = \big( (G_1, \mathbf{f}_1, y_1), \ldots, (G_m, \mathbf{f}_m, y_m) \big) \sim \mu_{\mathcal{G}}^m$ be a dataset of graphs. Then,*

$$\mathbb{E}_{\mathcal{T} \sim \mu_{\mathcal{G}}^m} \Bigg[ \sup_{\Theta \in \mathrm{Lip}_{L,B}} \bigg( \frac{1}{m} \sum_{i=1}^{m} \mathcal{L}(\Theta_{G_i}^P(\mathbf{f}_i), y_i) - \mathbb{E}_{(G, \mathbf{f}, y) \sim \mu_{\mathcal{G}}} \big[ \mathcal{L}(\Theta_G^P(\mathbf{f}), y) \big] \bigg)^2 \Bigg]$$

$$\leq 2^\Gamma \frac{8 \|\mathcal{L}\|_\infty^2}{m} \pi + \frac{6\sqrt{\pi}}{m} 2^\Gamma \Gamma \sum_{j=1}^{\Gamma} \gamma_j L_{\mathcal{L}}^2 \Bigg( \sqrt{\pi} \Big( \big( S_1^{(j)} + S_3^{(j)} + (S_2^{(j)} + S_4^{(j)}) \|f^j\|_\infty^2 \big) \mathbb{E}_{N \sim \nu} \big[ N^{-1} \big]$$

$$+ \big( R_1^{(j)} + R_4^{(j)} + (R_2^{(j)} + R_5^{(j)}) \|f^j\|_\infty^2 + R_3^{(j)} L_{f^j}^2 \big) \mathbb{E}_{N \sim \nu} \Big[ N^{-\frac{1}{D_{\chi^j}+1}} \Big]$$

$$+ \big( T_1^{(j)} + T_2^{(j)} \|f^j\|_\infty^2 \big) \mathbb{E}_{N \sim \nu} \Big[ \log(N) N^{-\frac{1}{D_{\chi^j}+1}} \Big] \Big) + \mathcal{O} \Big( \mathbb{E}_{N \sim \nu} \Big[ \exp(-N) N^{\frac{3}{2}T-\frac{3}{2}} \Big] \Big) \Bigg),$$

*where $S_l^{(j)}, R_l^{(j)}, T_l^{(j)}$ are the according constants from Theorem B.18 for each class $j$ and are defined in (51).*

*Proof.* Given $\mathbf{m} = (m_1, \ldots, m_\Gamma)$ with $\sum_{j=1}^\Gamma m_j = m$, recall that $\mathcal{G}^{\mathbf{m}}$ is the space of datasets with fixed number of samples $m_j$ from each class $j = 1, \ldots, \Gamma$. The probability measure on $\mathcal{G}^{\mathbf{m}}$ is given

by $\mu_{\mathcal{G}^{\mathbf{m}}}$ (see (55)). Similarly to the notation of Lemma C.4, we denote the conditional choice of the dataset on the choice of $\mathbf{m}$ by

$$\mathcal{T}_{\mathbf{m}} := \big\{ \{ G_i^j, \mathbf{f}_i^j \}_{i=1}^{m_j} \big\}_{j=1}^{\Gamma} \sim \mu_{\mathcal{G}^{\mathbf{m}}}.$$

Given $k \in \mathbb{Z}$, denote by $\mathcal{M}_k$ the set of all $\mathbf{m} = (m_1, \ldots, m_\Gamma) \in \mathbb{N}_0^\Gamma$ with $\sum_{j=1}^{\Gamma} m_j = m$, such that $2\sqrt{m}k \le \sum_{j=1}^{\Gamma} |m_j - m\gamma_j| < 2\sqrt{m}(k+1)$. Using these notations, we decompose the expected generalization error as follows.

$$\mathbb{E}_{\mathcal{T} \sim \mu_{\mathcal{G}}^m} \left[ \sup_{\Theta \in \mathrm{Lip}_{L,B}} \left( \frac{1}{m} \sum_{i=1}^{m} \mathcal{L}(\Theta_{G_i}^P(\mathbf{f}_i), y_i) - \mathbb{E}_{(G,\mathbf{f},y)\sim\mu_{\mathcal{G}}} \left[ \mathcal{L}(\Theta_G^P(\mathbf{f}), y) \right] \right)^2 \right]$$

$$= \mathbb{E}_{\mathcal{T} \sim \mu_{\mathcal{G}}^m} \left[ \sup_{\Theta \in \mathrm{Lip}_{L,B}} \left( \frac{1}{m} \sum_{j=1}^{\Gamma} \sum_{i=1}^{m_j} \mathcal{L}(\Theta_{G_i^j}^P(\mathbf{f}_i^j), y_j) - \mathbb{E}_{(G,\mathbf{f},y)\sim\mu_{\mathcal{G}}} \left[ \mathcal{L}(\Theta_G^P(\mathbf{f}), y) \right] \right)^2 \right]$$

$$= \mathbb{E}_{\mathcal{T} \sim \mu_{\mathcal{G}}^m} \left[ \sup_{\Theta \in \mathrm{Lip}_{L,B}} \left( \sum_{j=1}^{\Gamma} \left( \frac{1}{m} \sum_{i=1}^{m_j} \mathcal{L}(\Theta_{G_i^j}^P(\mathbf{f}_i^j), y_j) - \gamma_j \mathbb{E}_{(G^j,\mathbf{f}^j)\sim\mu_{\mathcal{G}_j}} \left[ \mathcal{L}(\Theta_{G^j}^P(\mathbf{f}^j), y_j) \right] \right) \right)^2 \right]$$

$$\le \sum_{k} \mathbb{P}\big( \mathbf{m} \in \mathcal{M}_k \big) \times \sup_{\mathbf{m} \in \mathcal{M}_k} \mathbb{E}_{\mathcal{T}_{\mathbf{m}} \sim \mu_{\mathcal{G}^{\mathbf{m}}}} \left[ \sup_{\Theta \in \mathrm{Lip}_{L,B}} \left( \sum_{j=1}^{\Gamma} \left( \frac{1}{m} \sum_{i=1}^{m_j} \mathcal{L}(\Theta_{G_i^j}^P(\mathbf{f}_i^j), y_j) \right. \right. \right.$$
$$\left. \left. \left. - \frac{1}{m} \sum_{i=1}^{m\gamma_j} \mathbb{E}_{(G^j,\mathbf{f}^j)\sim\mu_{\mathcal{G}_j}} \left[ \mathcal{L}(\Theta_{G^j}^P(\mathbf{f}^j), y_j) \right] \right) \right)^2 \right]$$

$$(56)$$

We bound the last term of (56) as follows. For $j = 1, \ldots, \Gamma$, if $m_j \le m\gamma_j$, we add "ghost samples", i.e., we add additional i.i.d. sampled graphs $(G_{m_j}^j, \mathbf{f}_{m_j}^j), \ldots, (G_{m\gamma_j}^j, \mathbf{f}_{m\gamma_j}^j) \sim (W^j, f^j)$. By convention, for any two $l, q \in \mathbb{N}_0$ with $l < q$, we define

$$\sum_{j=q}^{l} c_j = - \sum_{j=l}^{q} c_j.$$

for any sequence $c_j$ of reals, and define $\sum_{j=q}^{q} c_j = 0$. With these notations, we have

$$
\mathbb{E}_{\mathcal{T}_\mathbf{m} \sim \mu_\mathcal{G}\mathbf{m}} \left[ \sup_{\Theta \in \mathrm{Lip}_{L,B}} \left( \sum_{j=1}^{\Gamma} \left( \frac{1}{m} \sum_{i=1}^{m_j} \mathcal{L}(\Theta_{G_i^j}^P(\mathbf{f}_i^j), y_j) \right. \right. \right.
$$
$$
\left. \left. \left. - \frac{1}{m} \sum_{i=1}^{m\gamma_j} \mathbb{E}_{(G^j, \mathbf{f}^j) \sim \mu_{\mathcal{G}_j}} \left[ \mathcal{L}(\Theta_{G^j}^P(\mathbf{f}^j), y_j) \right] \right) \right)^2 \right]
$$

$$
= \mathbb{E}_{\mathcal{T}_\mathbf{m} \sim \mu_\mathcal{G}\mathbf{m}} \left[ \sup_{\Theta \in \mathrm{Lip}_{L,B}} \left( \sum_{j=1}^{\Gamma} \left( \frac{1}{m} \sum_{i=1}^{m\gamma_j} \mathcal{L}(\Theta_{G_i^j}^P(\mathbf{f}_i^j), y_j) + \frac{1}{m} \sum_{i=m\gamma_j}^{m_j} \mathcal{L}(\Theta_{G_i^j}^P(\mathbf{f}_i^j), y_j) \right. \right. \right.
$$
$$
\left. \left. \left. - \frac{1}{m} \sum_{i=1}^{m\gamma_j} \mathbb{E}_{(G^j, \mathbf{f}^j) \sim \mu_{\mathcal{G}_j}} \left[ \mathcal{L}(\Theta_{G^j}^P(\mathbf{f}^j), y_j) \right] \right) \right)^2 \right] \tag{57}
$$

$$
\leq \mathbb{E}_{\mathcal{T}_\mathbf{m} \sim \mu_\mathcal{G}\mathbf{m}} \left[ \sup_{\Theta \in \mathrm{Lip}_{L,B}} 2 \left( \sum_{j=1}^{\Gamma} \left( \frac{1}{m} \sum_{i=1}^{m\gamma_j} \mathcal{L}(\Theta_{G_i^j}^P(\mathbf{f}_i^j), y_j) \right. \right. \right.
$$
$$
\left. \left. \left. - \frac{1}{m} \sum_{i=1}^{m\gamma_j} \mathbb{E}_{(G^j, \mathbf{f}^j) \sim \mu_{\mathcal{G}_j}} \left[ \mathcal{L}(\Theta_{G^j}^P(\mathbf{f}^j), y_j) \right] \right) \right)^2 \right]
$$

$$
+ \mathbb{E}_{\mathcal{T}_\mathbf{m} \sim \mu_\mathcal{G}\mathbf{m}} \left[ 2 \left( \sum_{j=1}^{\Gamma} \left( \frac{1}{m} |m\gamma_j - m_j| \|\mathcal{L}\|_\infty \right) \right)^2 \right].
$$

Let us first bound the last term of the above bound. Since any $\mathbf{m} \in \mathcal{M}_\mathbf{k}$ satisfies $\sum_{j=1}^{\Gamma} |m_j - m\gamma_j| < 2\sqrt{m}(k+1)$, we have

$$
\mathbb{E}_{\mathcal{T}_\mathbf{m} \sim \mu_\mathcal{G}\mathbf{m}} \left[ 2 \left( \sum_{j=1}^{\Gamma} \left( \frac{1}{m} |m\gamma_j - m_j| \|\mathcal{L}\|_\infty \right) \right)^2 \right] \leq \frac{2}{m^2} \|\mathcal{L}\|_\infty^2 \left( \sum_{j=1}^{\Gamma} |m\gamma_j - m_j| \right)^2
$$

$$
\leq \frac{2}{m^2} \|\mathcal{L}\|_\infty^2 4m(k+1)^2 = \frac{8\|\mathcal{L}\|_\infty^2}{m}(k+1)^2.
$$

Hence, by Lemma C.6,

$$
\sum_k \mathbb{P}(\mathbf{m} \in \mathcal{M}_k) \times \sup_{\mathbf{m} \in \mathcal{M}_k} \mathbb{E}_{\mathcal{T}_\mathbf{m} \sim \mu_\mathcal{G}\mathbf{m}} \left[ 2 \left( \sum_{j=1}^{\Gamma} \left( \frac{1}{m} |m\gamma_j - m_j| \|\mathcal{L}\|_\infty \right) \right)^2 \right]
$$

$$
\leq \sum_k \mathbb{P}(\mathbf{m} \in \mathcal{M}_k) \times \frac{8\|\mathcal{L}\|_\infty^2}{m}(k+1)^2
$$

$$
\leq \sum_k 2^\Gamma \exp(-2k^2) \frac{8\|\mathcal{L}\|_\infty^2}{m}(k+1)^2
$$

$$
\leq \int_0^\infty 2^\Gamma \exp(-2k^2) \frac{8\|\mathcal{L}\|_\infty^2}{m}(k+1)^2 dk
$$

$$
= 2^\Gamma \frac{8\|\mathcal{L}\|_\infty^2}{m} \int_0^\infty \exp(-2k^2)(k+1)^2 dk
$$

$$
\leq 2^\Gamma \frac{8\|\mathcal{L}\|_\infty^2}{m} \pi.
$$

To bound the first term of the right-hand-side of (57), we have

$$
\mathbb{E}_{\mathcal{T}_{\mathbf{m}} \sim \mu_{\mathcal{G}^{\mathbf{m}}}} \left[ \sup_{\Theta \in \mathrm{Lip}_{L,B}} \left( \sum_{j=1}^{\Gamma} \left( \frac{1}{m} \sum_{i=1}^{m\gamma_j} \mathcal{L}(\Theta_{G_i^j}^P(\mathbf{f}_i^j), y_j) \right. \right. \right.
$$

$$
\left. \left. \left. - \frac{1}{m} \sum_{i=1}^{m\gamma_j} \mathbb{E}_{(G^j, \mathbf{f}^j) \sim \mu_{\mathcal{G}_j}} \left[ \sup_{\Theta \in \mathrm{Lip}_{L,B}} \mathcal{L}(\Theta_{G^j}^P(\mathbf{f}^j), y_j) \right] \right) \right)^2 \right]
$$

$$
\leq \Gamma \sum_{j=1}^{\Gamma} \mathbb{E}_{\mathcal{T}_{\mathbf{m}} \sim \mu_{\mathcal{G}^{\mathbf{m}}}} \left[ \sup_{\Theta \in \mathrm{Lip}_{L,B}} \left( \frac{1}{m} \sum_{i=1}^{m\gamma_j} \mathcal{L}(\Theta_{G_i^j}^P(\mathbf{f}_i^j), y_j) \right. \right.
$$

$$
\left. \left. - \frac{1}{m} \sum_{i=1}^{m\gamma_j} \mathbb{E}_{(G^j, \mathbf{f}^j) \sim \mu_{\mathcal{G}_j}} \left[ \sup_{\Theta \in \mathrm{Lip}_{L,B}} \mathcal{L}(\Theta_{G^j}^P(\mathbf{f}^j), y_j) \right] \right)^2 \right]
$$

$$
= \Gamma \sum_{j=1}^{\Gamma} \mathrm{Var}_{(G^j, \mathbf{f}^j) \sim \mu_{\mathcal{G}_j}} \left[ \sup_{\Theta \in \mathrm{Lip}_{L,B}} \frac{1}{m} \sum_{i=1}^{\gamma_j \cdot m} \mathcal{L}(\Theta_{G^j}^P(\mathbf{f}^j), y_j) \right]
$$

$$
= \Gamma \sum_{j=1}^{\Gamma} \frac{\gamma_j}{m} \mathrm{Var}_{(G^j, \mathbf{f}^j) \sim \mu_{\mathcal{G}_j}} \left[ \sup_{\Theta \in \mathrm{Lip}_{L,B}} \mathcal{L}(\Theta_{G^j}^P(\mathbf{f}^j), y_j) \right]
$$

$$
\leq \Gamma \sum_{j=1}^{\Gamma} \frac{\gamma_j}{m} \mathbb{E}_{(G^j, \mathbf{f}^j) \sim \mu_{\mathcal{G}_j}} \left[ \sup_{\Theta \in \mathrm{Lip}_{L,B}} \left| \mathcal{L}(\Theta_{G^j}^P(\mathbf{f}^j), y_j) - \mathcal{L}(\Theta_{W^j}^P(f^j), y_j) \right|^2 \right]
$$

$$
\leq \Gamma \sum_{j=1}^{\Gamma} \frac{\gamma_j}{m} \mathbb{E}_{(G^j, \mathbf{f}^j) \sim \mu_{\mathcal{G}_j}} \left[ \sup_{\Theta \in \mathrm{Lip}_{L,B}} L_{\mathcal{L}}^2 \| \Theta_{G^j}^P(\mathbf{f}^j) - \Theta_{W^j}^P(f^j) \|_\infty^2 \right].
$$

We now apply Corollary C.5 to get

$$
\leq \Gamma \sum_{j=1}^{\Gamma} \frac{\gamma_j}{m} L_{\mathcal{L}}^2 \left( 6\sqrt{\pi} \left( (S_1 + S_3 + (S_2 + S_4)\|f^j\|_\infty^2) \mathbb{E}_{N\sim\nu} \left[ N^{-1} \right] \right. \right.
$$

$$
+ (R_1 + R_4 + (R_2 + R_5)\|f^j\|_\infty^2 + R_3 L_{f^j}^2) \mathbb{E}_{N\sim\nu} \left[ N^{-\frac{1}{D_{\chi^j}+1}} \right]
$$

$$
\left. \left. + (T_1 + T_2\|f^j\|_\infty^2) \mathbb{E}_{N\sim\nu} \left[ \frac{\log(N)}{N^{\frac{1}{D_\chi^j+1}}} \right] \right) + \mathcal{O}\left( \mathbb{E}_{N\sim\nu} \left[ \exp(-N) N^{\frac{3}{2}T - \frac{3}{2}} \right] \right) \right).
$$

Hence, by Lemma C.6,

$$
\sum_k \mathbb{P}\big(\mathbf{m} \in \mathcal{M}_k\big) \times \sup_{\mathbf{m} \in \mathcal{M}_k} \mathbb{E}_{\mathcal{T}_{\mathbf{m}} \sim \mu_{\mathcal{G}^{\mathbf{m}}}} \left[ \sup_{\Theta \in \mathrm{Lip}_{L,B}} \left( \sum_{j=1}^{\Gamma} \left( \frac{1}{m} \sum_{i=1}^{m\gamma_j} \mathcal{L}(\Theta_{G_i^j}^P(\mathbf{f}_i^j), y_j) \right. \right. \right.
$$
$$
\left. \left. \left. - \frac{1}{m} \sum_{i=1}^{m\gamma_j} \mathbb{E}_{(G^j, \mathbf{f}^j) \sim \mu_{\mathcal{G}_j}} \big[ \mathcal{L}(\Theta_{G^j}^P(\mathbf{f}^j), y_j) \big] \right) \right)^2 \right]
$$
$$
\leq \frac{\sqrt{\pi}}{2} 2^{\Gamma} \sum_{j=1}^{\Gamma} \frac{\gamma_j}{m} \mathbb{E}_{\mathcal{T}_{\mathbf{m}} \sim \mu_{\mathcal{G}^{\mathbf{m}}}} \left[ \sup_{\Theta \in \mathrm{Lip}_{L,B}} \left( \sum_{j=1}^{\Gamma} \left( \frac{1}{m} \sum_{i=1}^{m\gamma_j} \mathcal{L}(\Theta_{G_i^j}^P(\mathbf{f}_i^j), y_j) \right. \right. \right.
$$
$$
\left. \left. \left. - \frac{1}{m} \sum_{i=1}^{m\gamma_j} \mathbb{E}_{(G^j, \mathbf{f}^j) \sim \mu_{\mathcal{G}_j}} \big[ \mathcal{L}(\Theta_{G^j}^P(\mathbf{f}^j), y_j) \big] \right) \right)^2 \right]
$$
$$
\leq \frac{\sqrt{\pi}}{2} 2^{\Gamma} \Gamma \sum_{j=1}^{\Gamma} \frac{\gamma_j}{m} L_{\mathcal{L}}^2 \left( 6\sqrt{\pi} \left( (S_1^{(j)} + S_3^{(j)} + (S_2^{(j)} + S_4^{(j)}) \|f^j\|_\infty^2) \mathbb{E}_{N \sim \nu} \left[ N^{-1} \right] \right. \right.
$$
$$
+ (R_1^{(j)} + R_4^{(j)} + (R_2^{(j)} + R_5^{(j)}) \|f^j\|_\infty^2 + R_3^{(j)} L_{f^j}^2) \mathbb{E}_{N \sim \nu} \left[ N^{-\frac{1}{D_{\chi^j}+1}} \right]
$$
$$
\left. \left. + (T_1^{(j)} + T_2^{(j)} \|f^j\|_\infty^2) \mathbb{E}_{N \sim \nu} \left[ \frac{\log(N)}{N^{\frac{1}{D_\chi^j+1}}} \right] \right) + \mathcal{O} \left( \mathbb{E}_{N \sim \nu} \left[ \exp(-N) N^{\frac{3}{2}T - \frac{3}{2}} \right] \right) \right),
$$

where $S_l^{(j)}, R_l^{(j)}, T_l^{(j)}$ are the according constants from Theorem B.18 for each class $j$ and are defined in (51). All in all, we get

$$
\mathbb{E}_{\mathcal{T} \sim \mu_{\mathcal{G}}^m} \left[ \sup_{\Theta \in \mathrm{Lip}_{L,B}} \left( \frac{1}{m} \sum_{j=1}^{\Gamma} \sum_{i=1}^{m_j} \mathcal{L}(\Theta_{G_i^j}^P(\mathbf{f}_i^j), y_j) - \mathbb{E}_{(G, \mathbf{f}, y) \sim \mu_{\mathcal{G}}} \big[ \mathcal{L}(\Theta_G^P(\mathbf{f}), y) \big] \right)^2 \right]
$$
$$
\leq 2^{\Gamma} \frac{8 \|\mathcal{L}\|_\infty^2}{m} \pi + \frac{\sqrt{\pi}}{m} 2^{\Gamma} \Gamma \sum_{j=1}^{\Gamma} \gamma_j L_{\mathcal{L}}^2 \left( 6\sqrt{\pi} \left( (S_1^{(j)} + S_3^{(j)} + (S_2^{(j)} + S_4^{(j)}) \|f^j\|_\infty^2) \mathbb{E}_{N \sim \nu} \left[ N^{-1} \right] \right. \right.
$$
$$
+ (R_1^{(j)} + R_4^{(j)} + (R_2^{(j)} + R_5^{(j)}) \|f^j\|_\infty^2 + R_3^{(j)} L_{f^j}^2) \mathbb{E}_{N \sim \nu} \left[ N^{-\frac{1}{D_{\chi^j}+1}} \right]
$$
$$
\left. \left. + (T_1^{(j)} + T_2^{(j)} \|f^j\|_\infty^2) \mathbb{E}_{N \sim \nu} \left[ \frac{\log(N)}{N^{\frac{1}{D_\chi^j+1}}} \right] \right) + \mathcal{O} \left( \mathbb{E}_{N \sim \nu} \left[ \exp(-N) N^{\frac{3}{2}T - \frac{3}{2}} \right] \right) \right).
$$

We define

$$
C = 6\sqrt{\pi} \max_{j=1,\ldots,\Gamma} \left( \sum_{i=1}^{4} S_i^{(j)} + \sum_{i=1}^{5} R_i^{(j)} + \sum_{i=1}^{2} T_i^{(j)} \right), \tag{58}
$$

leading to

$$
\mathbb{E}_{\mathcal{T} \sim \mu_{\mathcal{G}}^m} \left[ \sup_{\Theta \in \mathrm{Lip}_{L,B}} \left( R_{emp}(\Theta^P) - R_{exp}(\Theta^P) \right)^2 \right]
$$
$$
\leq \frac{2^{\Gamma} 8 \|\mathcal{L}\|_\infty^2 \pi}{m} + \frac{2^{\Gamma} \Gamma L_{\mathcal{L}}^2 C}{m} \sum_j \gamma_j \big( 1 + \|f^j\|_\infty^2 + L_{f^j}^2 \big)
$$
$$
\cdot \left( \mathbb{E}_{N \sim \nu} \left[ \frac{1}{N} + \frac{1 + \log(N)}{N^{1/D_{\chi^j}+1}} + \mathcal{O} \left( \exp(-N) N^{\frac{3}{2}T - \frac{3}{2}} \right) \right] \right).
$$

$\square$

## C.3 Asymptotics of the Generalization Bound

In this subsection, we derive the asymptotic dependency of our generalization bound in Theorem 3.3 with respect to the uniform Lipschitz bound $L$ of the message and update function, the depth $T$, the maximal hidden dimension $h$ and the average graph size, that we denote in this section by abuse of notation $N$. Since we bound the expected square generalization error, and most other related generalization bounds are formulated in high probability, we transform our bound in expectation to a bound in high probability, using, e.g., Markov's Inequality (and then taking the square root of the square error). By this, the comparison with other generalization bounds formulated in high probability are valid. Hence, we focus on the constant $\sqrt{C}$, where $C$ is the constant from Theorem 3.3. We reformulated Theorem 3.3 as Theorem C.7, where we observed that $C \leq 6\sqrt{\pi} \max_{j=1,\dots,\Gamma} \left( \sum_{i=1}^{4} S_i^{(j)} + \sum_{i=1}^{5} R_i^{(j)} + \sum_{i=1}^{2} T_i^{(j)} \right)$, where $S_l^{(j)}, R_l^{(j)}, T_l^{(j)}$ are the according constants from Theorem B.18 for each class $j$ and are defined in (51). For a better presentation, we drop the class-superscript by setting $S_l = \max_j S_l^{(j)}$, for $l = 1, \dots, 4$, $R_l = \max_j R_l^{(j)}$, for $l = 1, \dots, 5$ and $T_l = \max_j T_l^{(j)}$, for $l = 1, 2$. Further, denote $C_\chi = \max_j C_{\chi^j}$, $D_\chi = \max_j D_{\chi^j}$, $L_W = \max_j L_{W^j}$ and $\|W\|_\infty = \max_j \|W^j\|_\infty$.

The constants $R_i, S_i$ and $T_i$ are bounded by a polynomial of order 2 in $\Omega_j$, for $j = 1, \dots, 9$, defined in (48). The constants $\Omega_j$, $j = 1, \dots, 9$, depend on a polynomial of degree one in $Z_1^{(l)}, Z_2^{(l)}, Z_3^{(l)}$, $B_1^{(l)}, B_2^{(l)}$ and on a polynomial of degree at most $T - 1$ in $K^{(l)}$ for $l = 1, \dots, T-1$. Here, $Z_1^{(l)}, Z_2^{(l)}, Z_3^{(l)}$ are defined in (31), $B_1^{(l)}$ and $B_2^{(l)}$ are defined in (27) and (28), and

$$K^{(l')} = \sqrt{(L_{\Psi^{(l')}})^2 + \frac{8\|W\|_\infty^2}{d_{\min}^2}(L_{\Phi^{(l')}})^2(L_{\Psi^{(l')}})^2}.$$

Hence, our strategy is as follows. We first work out the asymptotic behaviour of $Z_1^{(l)}, Z_2^{(l)}, Z_3^{(l)}$, $B_1^{(l)}, B_2^{(l)}$ and $K^{(l)}$ for $l = 1, \dots, T-1$ with respect to the parameters. Then, we derive the asymptotics of $\Omega_j$, $j = 1, \dots, 9$. These already agree with the asymptotic of $\sqrt{C}$. For this, we write $A \lesssim x^k$ if $A$ is bounded by a polynomial of order $k$ in $x$.

We begin with observing that $K^{(l')} \lesssim L^2 \frac{\|W\|_\infty}{d_{\min}}$. Since we only consider MPNNs $\Theta \in \mathrm{Lip}_{L,B}$, we have for $l = 1, \dots, T$,

$$B_1^{(l)} \leq \sum_{k=1}^{l} \left( L_{\Psi^{(k)}} \frac{\|W\|_\infty}{d_{\min}} \|\Phi^{(k)}(0,0)\|_\infty + \|\Psi^{(k)}(0,0)\|_\infty \right) \prod_{l'=k+1}^{l} L_{\Psi^{(l')}} \left( 1 + \frac{\|W\|_\infty}{d_{\min}} L_{\Phi^{(l')}} \right)$$

$$\lesssim \sum_{k=1}^{l} LB \frac{\|W\|_\infty}{d_{\min}} \left( \frac{\|W\|_\infty}{d_{\min}} L^2 \right)^{l-k} \lesssim \frac{\|W\|_\infty^l}{d_{\min}^l} L^{2l} B.$$

and

$$B_2^{(l)} \leq \prod_{k=1}^{l} L_{\Psi^{(k)}} \left( 1 + \frac{\|W\|_\infty}{d_{\min}} L_{\Phi^{(k)}} \right)$$

$$\lesssim \frac{\|W\|_\infty^l}{d_{\min}^l} L^{2l}.$$

For $l = 1, \ldots, T$, the constants $Z_1^{(l)}$, are defined in (31). We have

$$
\begin{aligned}
Z_1^{(l)} &\leq \sum_{k=1}^{l} \Bigg( \Big( L_{\Psi^{(k)}} \frac{L_W}{\mathrm{d}_{\min}} \|\Phi^{(k)}(0,0)\|_\infty + L_{\Psi^{(k)}} \|W\|_\infty \|\Phi^{(k)}(0,0)\|_\infty \frac{L_W}{\mathrm{d}_{\min}^2} \Big) \\
&\quad + B_1^{(k-1)} \Big( L_{\Psi^{(k)}} \frac{L_W}{\mathrm{d}_{\min}} L_{\Phi^{(k)}} + L_{\Psi^{(k)}} \|W\|_\infty L_{\Phi^{(k)}} \frac{L_W}{\mathrm{d}_{\min}^2} \Big) \Bigg) \prod_{l'=k+1}^{l} L_{\Psi^{(l')}} \Big( 1 + \frac{\|W\|_\infty}{\mathrm{d}_{\min}} L_{\Phi^{(l')}} \Big) \\
&\lesssim \sum_{k=1}^{l} B_1^{(k-1)} L^2 \frac{L_W}{\mathrm{d}_{\min}^2} \Big( \frac{\|W\|_\infty}{\mathrm{d}_{\min}} L^2 \Big)^{l-k} \\
&\lesssim \sum_{k=1}^{l} B \|W\|_\infty \frac{\|W\|_\infty^{k-1}}{\mathrm{d}_{\min}^{k-1}} (L^2)^{k-1} L^2 \frac{L_W}{\mathrm{d}_{\min}^2} \Big( \frac{\|W\|_\infty}{\mathrm{d}_{\min}} L^2 \Big)^{l-k} \\
&\lesssim B \frac{\|W\|_\infty^l L_W}{\mathrm{d}_{\min}^{l+1}} L^{2l}.
\end{aligned}
$$

We have

$$
\begin{aligned}
Z_2^{(l)} &\leq \sum_{k=1}^{l} B_2^{(k)} \Big( L_{\Psi^{(k)}} \frac{L_W}{\mathrm{d}_{\min}} L_{\Phi^{(k)}} + L_{\Psi^{(k)}} \|W\|_\infty L_{\Phi^{(k)}} \frac{L_W}{\mathrm{d}_{\min}^2} \Big) \prod_{l'=k+1}^{l} L_{\Psi^{(l')}} \Big( 1 + \frac{\|W\|_\infty}{\mathrm{d}_{\min}} L_{\Phi^{(l')}} \Big) \\
&\lesssim \sum_{k=1}^{l} B_2^{(k-1)} L^2 \frac{L_W}{\mathrm{d}_{\min}} \|W\|_\infty \Big( \frac{\|W\|_\infty}{\mathrm{d}_{\min}} L^2 \Big)^{l-k} \\
&\lesssim \sum_{k=1}^{l} \frac{\|W\|_\infty^{k-1}}{\mathrm{d}_{\min}^{k-1}} (L^2)^{k-1} L^2 \frac{L_W}{\mathrm{d}_{\min}} \|W\|_\infty \Big( \frac{\|W\|_\infty}{\mathrm{d}_{\min}} L^2 \Big)^{l-k} \\
&\lesssim \frac{L_W \|W\|_\infty^l}{\mathrm{d}_{\min}^l} L^{2l}.
\end{aligned}
$$

We have

$$
\begin{aligned}
Z_3^{(l)} &\leq \prod_{k=1}^{l} L_{\Psi^{(k)}} \Big( 1 + \frac{\|W\|_\infty}{\mathrm{d}_{\min}} L_{\Phi^{(k)}} \Big) \\
&\lesssim \frac{\|W\|_\infty^l}{\mathrm{d}_{\min}^l} L^{2l}.
\end{aligned}
$$

For $i = 1, \ldots, 9$, the constant $\Omega_i$ depends on $K^{(l)}$ for which we have

$$
K^{(l')} \leq \sqrt{ (L_{\Psi^{(l')}})^2 + \frac{8\|W\|_\infty^2}{\mathrm{d}_{\min}^2} (L_{\Phi^{(l')}})^2 (L_{\Psi^{(l')}})^2 } \lesssim \frac{\|W\|_\infty}{\mathrm{d}_{\min}} L^2
$$

For $\Omega_1$, we calculate

$$
\begin{aligned}
\Omega_1 &\leq \sum_{l=1}^{T} L_{\Psi^{(l)}} 4 \frac{\zeta L_W \big( \sqrt{\log(C_\chi)} + \sqrt{D_\chi} \big)}{\mathrm{d}_{min}^2} \|W\|_\infty \big( L_{\Phi^{(l)}} B_1^{(l-1)} + \|\Phi^{(l)}(0,0)\|_\infty \big) \prod_{l'=l+1}^{T} K^{(l')} \\
&\lesssim \big( \sqrt{\log(C_\chi)} + \sqrt{D_\chi} \big) \sum_{l=1}^{T} L^2 B_1^{(l-1)} \frac{L_W \|W\|_\infty}{\mathrm{d}_{\min}^2} (L^2)^{T-l} \\
&\lesssim \big( \sqrt{\log(C_\chi)} + \sqrt{D_\chi} \big) B L^{2T} \frac{L_W \|W\|_\infty^T}{\mathrm{d}_{\min}^{T+1}}
\end{aligned}
$$

Similar calculations lead to

$$
\Omega_i \lesssim \big( \sqrt{\log(C_\chi)} + \sqrt{D_\chi} \big) B (L^2)^T \frac{L_W \|W\|_\infty^T}{\mathrm{d}_{\min}^{T+1}}.
$$

Hence,

$$GE \lesssim \frac{2^{\Gamma/2}}{\sqrt{m}} + \frac{2^{\Gamma/2}\big(\sqrt{\log(C_\chi)} + \sqrt{D_\chi}\big)BL^{2T}L_W\|W\|_\infty^T}{\sqrt{m}\mathrm{d}_{\min}^{T+1}}\mathbb{E}_{N\sim\nu}\left[\frac{\sqrt{\log(N)}}{N^{\frac{1}{2(D_\chi+1)}}}\right]. \qquad (59)$$

## C.4 Generalization Bound Comparison

In this subsection, we compare our generalization bound, especially the asymptotics derived in the previous subsection, with other related generalization bounds. Since related work does neither consider the same network architecture, nor the same data distribution as our work, we emphasize the setting of each of the cited results. We then write the asymptotics of the cited bounds in terms of the maximal hidden dimension $h$, depth $T$, Lipschitz bound $L$ of the message and update functions, maximal node $d$ degree and graph size $N$. We recall (59), where we derived the asymptotics of our generalization bound from Theorem 3.3 with respect to $T, L$ and $N$ as

$$\mathcal{O}\left(\mathbb{E}_{N\sim\nu}\left[\frac{\sqrt{\log(N)}}{N^{\frac{1}{2(D_\chi+1)}}}\right]\right), \qquad \mathcal{O}\left(L^{2T}\right)$$

and $\mathcal{O}(1)$ with respect to $h$.

### C.4.1 PAC-Bayesian Approach based Bound

The generalization analysis of [Liao et al., 2021] considers MPNNs with sum aggregation for a $K$-class graph classification setting. The authors differentiate between the input node feature vectors $\mathbf{x}_v$, which is an unchanged input for every layer, and the node embedding/representation in the $l$-th layer $\mathbf{f}^{(l)}$, where they take $\mathbf{f}^{(0)} = 0$. More formally, the MPNNs takes the following form.

**Definition C.8.** *Let $G$ be a graph with graph features $\mathbf{x}$. A MPNN (in [Liao et al., 2021]) with $T$ layers is defined by taking the input feature representation $\mathbf{f}^{(0)} = 0$, and mapping it to the features $\mathbf{f}^{(l)}$ in the $l$-th layer, which are defined recursively by*

$$\mathbf{f}_v^{(l)} = \Psi\left(W_1\mathbf{x}_v + W_2\rho\left(\sum_{u\in\mathcal{N}(v)}\Phi(\mathbf{f}_u^{(l-1)})\right)\right), \qquad (60)$$

*where $\rho$, $\Psi$ and $\Phi$ are nonlinear transformations, and $W_1$ and $W_2$ are linear transformations. This is followed by a global pooling layer, which takes as an input $\mathbf{f}^{(T-1)} \in \mathbb{R}^{N\times K}$, and returns the vector*

$$\frac{1}{N}\mathbf{1}_N\mathbf{f}^{(T-1)}W_T \in \mathbb{R}^{1\times K},$$

*where $W_T$ is a linear transformation. Here $\mathbf{1}_N$ denotes the vector $(1,\ldots,1) \in \mathbb{R}^{1\times N}$, where $N$ is the number of nodes in the graph.*

The message and update functions in Definition C.8 are the same in every layer. It is assumed that $\Psi, \rho$ and $\Phi$ have Lipschitz constants $L_\Psi, L_\rho$ and $L_\Phi$. Furthermore it is assumed that $W_1, W_2$ and $W_T$ have bounded norms, i.e., $\|W_1\|_2 \leq B_1, \|W_2\|_2 \leq B_2$ and $\|W_T\|_2 \leq B_T$.

The expected multiclass margin loss is then defined as

$$R_{\mathcal{D},\gamma}(\Theta) = \mathbb{P}_{(G,\mathbf{x},\mathbf{y})\sim\mathcal{D}}\left(\left(\Theta_G^P(\mathbf{x})\right)_{\mathbf{y}} \leq \gamma + \max_{j\neq\mathbf{y}}\left(\Theta_G^P(\mathbf{x})\right)_j\right),$$

where $\mathcal{D}$ is the unknown data distribution, $\gamma > 0$ and $\Theta_G^P$ is the MPNN after pooling. Accordingly, the empirical loss is defined as

$$R_{\mathcal{T},\gamma}(\Theta) = \frac{1}{m}\sum_{(G_i,\mathbf{x}_i,\mathbf{y}_i)\in\mathcal{T}}\mathbb{1}\left(\left(\Theta_{G_i}^P(\mathbf{x}_i)\right)_{\mathbf{y}_i} \leq \gamma + \max_{j\neq\mathbf{y}_i}\left(\Theta_{G_i}^P(\mathbf{x}_i)\right)_j\right),$$

where the summand $\mathbb{1}\left(\left(\Theta_{G_i}^P(\mathbf{x}_i)\right)_{\mathbf{y}_i} \leq \gamma + \max_{j\neq\mathbf{y}_i}\left(\Theta_{G_i}^P(\mathbf{x}_i)\right)_j\right)$ is equal to 1 if $\left(\Theta_{G_i}^P(\mathbf{x}_i)\right)_{\mathbf{y}_i} \leq \gamma + \max_{j\neq\mathbf{y}_i}\left(\Theta_{G_i}^P(\mathbf{x}_i)\right)_j$ and otherwise 0.

Furthermore, the following assumptions hold for the training set and the considered MPNNs

**Assumption C.9.**

1. *The training set $\mathcal{T} = \{(G_1, \mathbf{x}_1, \mathbf{y}_1), \dots, (G_m, \mathbf{x}_m, \mathbf{y}_m)\}$ is drawn i.i.d. from some distribution $\mathcal{D}$, where all graphs are simple and have node degrees at most $d - 1$.*

2. *The maximum hidden dimension across all layers is $h$.*

3. *The node features are drawn in an $l^2$-ball with radius $B$ from the node feature space $\mathcal{X}$.*

The generalization bound is formulated in terms of the following constants: $\zeta = \min\left(\|W_1\|_2, \|W_2\|_2, \|W_T\|_2\right)$, $|w|_2^2 = \|W_1\|_F^2 + \|W_2\|_F^2 + \|W_T\|_F^2$, $\lambda = \|W_1\|_2\|W_T\|_2$, $\xi = L_\Psi \frac{(d\mathcal{C})^{l-1}-1}{d\mathcal{C}-1}$, and the percolation complexity $\mathcal{C} = L_\Psi L_\rho L_\Phi \|W_2\|_2$. We summarize the main result [Liao et al., 2021, Theorem 3.4] as follows.

**Theorem.** *Let $T > 1$. Then for any $\delta, \gamma > 0$, with probability at least $1 - \delta$ over the choice of the training set $\mathcal{T} \sim \mathcal{D}^m$ of $m$ graphs, for any $T$-layered MPNN $\Theta$, we have,*

1. *If $d\mathcal{C} = 1$, then*

$$R_{\mathcal{D},0}(\Theta) \leq R_{\mathcal{T},\gamma}(\Theta)$$
$$+ \mathcal{O}\left(\sqrt{\frac{B^2 \max\left(\zeta^{-6}, \lambda^3 L_\Psi^3\right)(T+1)^4 h \log(Th)|w|_2^2 + \log\frac{m}{\delta}}{\gamma^2 m}}\right).$$

2. *If $d\mathcal{C} \neq 1$, then*

$$R_{\mathcal{D},0}(\Theta) \leq R_{\mathcal{T},\gamma}(\Theta)$$
$$+ \mathcal{O}\left(\sqrt{\frac{B^2\left(\max\left(\zeta^{-(T+1)}, (\lambda\xi)^{(T+1)/T}\right)\right)^2 T^2 h \log(Th)|w|_2^2 + \log\frac{m(T+1)}{\delta}}{\gamma^2 m}}\right).$$

We only consider the non-degenerative case $d\mathcal{C} \neq 1$, as it is the generic case, which can again be split into two cases. As the authors in [Liao et al., 2021] mention, these two cases correspond to $\max(\zeta^{-1}, (\lambda\xi)^{\frac{1}{T}}) = \zeta^{-1}$ (case A) and $\max(\zeta^{-1}, (\lambda\xi)^{\frac{1}{T}}) = (\lambda\xi)^{\frac{1}{T}}$ (case B). In practice case B occurs more often, where the generalization bound depends on the parameters with orders $\mathcal{O}\left(d^{\frac{(T+1)(T-2)}{T}}\right)$, $\mathcal{O}\left(\sqrt{h\log h}\right)$ and $\mathcal{O}\left(\lambda^{1+\frac{1}{T}}\xi^{1+\frac{1}{T}}\sqrt{\|W_1\|_F^2 + \|W_2\|_F^2 + \|W_l\|_F^2}\right)$. In case A, the generalization bound depends on the parameters with orders $\mathcal{O}\left(\sqrt{h\log h}\right)$ and $\mathcal{O}\left(\zeta^{-(T+1)}\sqrt{\|W_1\|_F^2 + \|W_2\|_F^2 + \|W_l\|_F^2}\right)$.

We now describe the architecture in Definition C.8 in terms of the message passing framework from (1). For $l = 1, \dots, T$, we denote by $\mathbf{m}_i^{(l)}$ and $\mathbf{f}_i^{(l)}$ the message and graph feature of node $i$ in the $l$-th layer, respectively. Given a simple graph $G$ with node features $(\mathbf{x}_i)_i$, we set $\mathbf{f}_i = \mathbf{x}_i$ as the input for the MPNN. Then the message function in the first layer is given by $\Phi^{(1)}(\mathbf{f}_i, \mathbf{f}_j) = \mathbf{f}_i$. We recall that the message in MPNNs with sum aggregation is calculated as $\mathbf{m}_i^{(1)} = \sum_{j \in \mathcal{N}(i)} \Phi^{(1)}(\mathbf{f}_i, \mathbf{f}_j)$. The update function in the first layer is given by $\Psi^{(1)}(\mathbf{f}_i, \mathbf{m}_i^{(1)}) = \left(\Phi(W_1 \mathbf{f}_i), \mathbf{f}_i\right)$. For $l = 2, \dots, T-1$, the message functions are defined as

$$\Phi^{(l)}(\mathbf{f}_i^{(l-1)}, \mathbf{f}_j^{(l-1)}) = \Phi(\mathbf{f}_i^{(l-1)})$$

and the update functions are defined as

$$\Psi^{(l)}(\mathbf{f}_i^{(l-1)}, \mathbf{m}_i^{(l)}) = \Psi\left(W_1(\mathbf{f}_i^{(l-1)})_2 + W_2\rho(\mathbf{m}_i^{(l)}), (\mathbf{f}_i^{(l-1)})_2\right),$$

where $(\mathbf{f}_i^{(l-1)})_2$ stays unchanged through all layers, and is equal to the input graph features $\mathbf{x}_i$. The aggregation scheme is given by sum aggregation. Finally, the pooling in Definition C.8 can be described by a graph MPNN layer with update function $W_T$ followed by average pooling. With this construction of message and update functions the MPNN $\Theta = \left((\Psi^{(l)})_{l=1}^T, (\Phi^{(l)})_{l=1}^T\right)$ with sum aggregation matches the architecture in Definition C.8.

We summarize the Lipschitz bounds for the message and update functions by $L_{\Phi^{(1)}} = 1$, $L_{\Phi^{(T)}} = 1$, $L_{\Psi^{(1)}} = L_\Phi$, $L_{\Phi^{(T)}} = \|W_T\|_2$ and $L_{\Phi^{(l)}} = L_\Phi$, $L_{\Psi^{(l)}} = L_\Psi \big( \|W_1\|_2 + \|W_2\|_2 L_\rho \big)$ for $l = 2, \ldots, T - 1$. For deriving our generalization bound in Theorem 3.1, we assume that there exists a uniform Lipschitz bound for the message and update functions, denoted by $L$. Hence, we assume that $L_\Phi \leq L$, $L_\Psi \|W_1\|_2 + L_\Psi \|W_2\|_2 L_\rho \leq L$ and $\|W_T\|_2 \leq L$.

For simplicity and better comparison with our generalization bound, we make use of the following upper bounds,

$$
\begin{aligned}
\mathcal{C} &= L_\Psi L_\rho L_\Phi \|W_2\| \leq L^2, \\
\xi &= L_\Psi \frac{(d\mathcal{C})^{T-1} - 1}{d\mathcal{C} - 1} \leq L(L^2)^{T-2}, \\
\zeta &= \min(\|W_1\|_2, \|W_2\|_2, \|W_l\|_2) \leq L \text{ and} \\
\lambda &= \|W_1\|_2 \|W_l\|_2 \leq L.
\end{aligned}
\tag{61}
$$

This leads to

$$
\begin{aligned}
\mathcal{O}\left( \lambda^{1+\frac{1}{T}} \xi^{1+\frac{1}{T}} \sqrt{\|W_1\|_F^2 + \|W_2\|_F^2 + \|W_l\|_F^2} \right) &= \mathcal{O}\left( L^{1+\frac{1}{T}} (L(L^2)^{T-2})^{1+\frac{1}{T}} L \right) \\
&= \mathcal{O}\left( L^{2T-2/T+1} \right).
\end{aligned}
$$

Hence, the asympotics of the generalization bound in [Liao et al., 2021] with respect to the maximal hidden dimension $h$, the Lipschitz bound $L$, the depth $T$ and the maximum node degree $d$ can be summarized respectively as

$$
\mathcal{O}\left( d^{\frac{(T+1)(T-2)}{T}} \right), \quad \mathcal{O}\left( \sqrt{h \log h} \right) \quad \text{and} \quad \mathcal{O}\left( L^{2T-2/T+1} \right).
$$

### C.4.2  Rademacher Complexity based Bound

We next analyze the bound derived in [Garg et al., 2020]. Since [Garg et al., 2020] consider the same architecture, defined in Definition C.8, as [Liao et al., 2021], we adopt the notation from Subsection C.4.1. The authors in [Garg et al., 2020] consider a binary graph classification task with the same Assumptions C.9 on the training set and the MPNN as in [Liao et al., 2021]. The main result can be summarized as follows.

**Theorem.** *Let $T > 1$. Then for any $\delta, \gamma > 0$, with probability at least $1 - \delta$ over the choice of the training set $\mathcal{T} \sim \mathcal{D}^m$ of $m$ graphs, for any $T$-layered MPNN $\Theta$, we have,*
$$
R_{\mathcal{D},0}(\Theta) \leq R_{\mathcal{T},\gamma}(\Theta)
$$
$$
+ \mathcal{O}\left( \frac{1}{\gamma m} + h\|W_T\|_2 Z \sqrt{\frac{\log\left( \|W_T\|_2 \sqrt{m} \max\left( Z, \xi \sqrt{h} \max\left( B\|W_1\|_2, \bar{R}\|W_2\|_2 \right) \right) \right)}{\gamma^2 m}} + \sqrt{\frac{\frac{1}{\delta}}{m}} \right),
$$

*where $\bar{R}$ is a constant specified in [Garg et al., 2020] that satisfies $\bar{R} \leq L_\rho L_\Phi dB\|W_1\|_2 \xi$, and $Z = B\|W_1\|_2\|W_T\|_2$.*

We only consider the case $\max\left( Z, \xi\sqrt{h} \max\left( BB_1, \bar{R}B_2 \right) \right) = \xi\sqrt{h}\bar{R}B_2$, which is the generic case (see [Liao et al., 2021, Subsection A.5.2] for the other cases). Thus the generalization bound from [Garg et al., 2020] depends on the parameters with orders $\mathcal{O}\left( d^{T-1}\sqrt{\log(d^{2T-3})} \right), \mathcal{O}\left( h\sqrt{\log\sqrt{h}} \right)$ and $\mathcal{O}\left( \lambda\mathcal{C}\xi\sqrt{\log(\|W_2\|_2\lambda\xi^2)} \right)$.

Similarly to Subsection C.4.1, we consider a uniform Lipschitz bound $L$ for the message and update functions. We thus consider the upper bounds on $\xi, \lambda$ and $\mathcal{C}$, summarized in (61), which leads to

$$
\mathcal{O}\left( \lambda\mathcal{C}\xi\sqrt{\log(\|W_2\|_2\lambda\xi^2)} \right) = \mathcal{O}\left( L^{2T}\sqrt{\log(L^{4T-4})} \right).
$$

Hence, the asympotics of the Rademacher based generalization bound in [Garg et al., 2020] with respect to the maximal hidden dimension $h$, the Lipschitz bound $L$, the depth $T$ and the maximum node degree $d$ can be summarized as

$$
\mathcal{O}\left( d^{T-1}\sqrt{\log(d^{2T-3})} \right), \quad \mathcal{O}\left( h\sqrt{\log\sqrt{h}} \right) \quad \text{and} \quad \mathcal{O}\left( L^{2T}\sqrt{\log(L^{4T-4})} \right).
$$

**VC-Dimension Based Bound [Scarselli et al., 2018]**   The work by [Scarselli et al., 2018] considers graph neural networks in supervised classification or regression tasks, where the input is a graph $G$ with graph feature map $\mathbf{x}$ and one node of interest $v$ in which we want to produce a prediction. They apply a recurrent graph neural network on the graph $G$ with graph feature $\mathbf{x}$, and then evaluate the output graph feature map $\mathbf{f}$ only in $v$. They then calculate the loss between $\mathbf{f}(v)$ and its given desired target $\mathbf{y}$. More formally, the training dataset $\mathcal{T}$ is defined as $\mathcal{T} = \{(G^i, \mathbf{x}^i, v^i, \mathbf{y}^i) \mid 1 \leq i \leq m\}$, where $m$ is the number of graphs and each tuple $(G^i, \mathbf{x}^i, v^i, \mathbf{y}^i)$ denotes a graph $G^i$ with graph features $\mathbf{x}^i$, the supervised node $v^i$, and the desired target $\mathbf{y}^i$ for that node.

Given a graph $G = (V, E)$ with graph features $\mathbf{x}$ the graph neural network architecture is defined implicitly, as a method that solves a system of equations, and the solution is the output of the network. The equation is given by

$$\mathbf{f}_i = \sum_{j \in \mathcal{N}(i)} \Phi(\mathbf{x}_i, \mathbf{f}_j, \mathbf{x}_j), \; \forall i \in V \tag{62}$$

where $\Phi$ is a multi-layer-perceptron with input $[\mathbf{x}_i, \mathbf{f}_j, \mathbf{x}_j]$, and the solution $\mathbf{f}$ to (62) is defined as the output of this part of the network. The output of the network $\mathbf{o}_i \in \mathbb{R}$ for the node $i$ is then defined by

$$\mathbf{o}_i = g(\mathbf{x}_i, \mathbf{f}_i), \tag{63}$$

where $g$ is a multi-layer-perceptron. Given the training data set $\mathcal{T}$, the empirical loss $R_{\mathrm{emp}}$ is then defined by the sum of the squared errors, i.e.,

$$R_{\mathrm{emp}} = \sum_{i=1}^{m} (\mathbf{y}^i - \mathbf{o}_{v^i})^2.$$

One way to solve the fixed point problem (62) is by a fixed point iteration, which means that we can interpret the architecture as a recurrent message passing network (theoretically with infinite depth), where all message functions in all layers are equal to $\Phi$.

Scarselli et al. [2018] derive VC-dimension bounds for the mapping that takes as an input a tuple $(G, \mathbf{x}, v)$ of a graph $G$ with features $\mathbf{x}$ and node of interest $v$ and outputs $\mathbf{o}_v$ as defined in (62) and (63). The VC-dimension bound depends on the total number of parameters $p$ of the network and a predefined maximum graph size $N$. Furthermore, the bound for the VC-dimension depends on the choice of the activation function in the MLPs $\Phi$ and $g$. If the activation is given by tanh and logistic sigmoid activations the VC-dimension scales as $\mathcal{O}(p^4 N^2)$. Since $p$ can be related to the maximum hidden dimension $h$ by $p \in \mathcal{O}(h^2)$, the VC-dimension scales as $\mathcal{O}(h^8 N^2)$. Consequently, the asymptotics of the generalization bounds in [Scarselli et al., 2018] with respect to $h$ and $N$ can be summarized as

$$\mathcal{O}(h^4) \text{ and } \mathcal{O}(N).$$

For piecewise polynomial activations the VC-dimension scales as $\mathcal{O}(h^4 \log(N)N)$, hence the generalization bound scales in this case as

$$\mathcal{O}(h^2) \text{ and } \mathcal{O}(\sqrt{\log(N)N})$$

with respect to $h$ and $N$.

# D   Details on Numerical Experiments and Additional Experiments

In this section we report additional experiments and write all details corresponding to Section 4. We First give an example that illustrate our convergence theorem (Theorem 3.1), and then introduce a comparison between our generalization bound and the Rademacher complexity [Garg et al., 2020] and PAC-Bayesian [Liao et al., 2021] bounds, evaluated on synthetic datasets.

## D.1   Convergence Experiments

In this section, we show simple numerical experiments on the convergence of sampled MPNNs from a random geometric graph model, on toy data. We consider random geometric graphs [Penrose, 2003], which can be described by using RGMs with the kernel $W(x, y) = \mathbb{1}_{B_r(x)}(y)$ on $[0, 1]^2$, equipped with the uniform distribution and the standard Euclidean norm. Here $\mathbb{1}_{B_r(x)}$ is the indicator function of the ball around $x$ with radius $r$. Even though $\mathbb{1}_{B_r(x)}(y)$ is not Lipschitz continuous, and

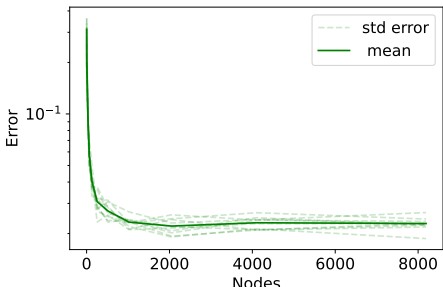 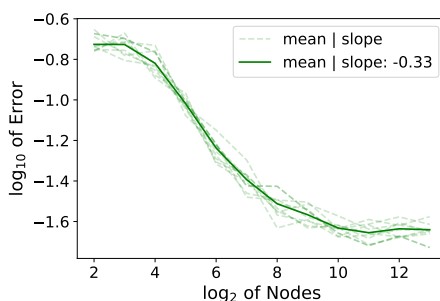

Figure 2: The average worst-case error between MPNNs realized on graphs and on the limit RGM, with varying number of nodes, drawn from the RGM $W(x, y) = \mathbb{1}_{B_r(x)}(y)$ (where $\mathbb{1}_{B_r(x)}$ is the indicator function of the ball around $x$ with radius $r = 0.2$ in the space $([0,1]^2, \|\cdot\|_{\mathbb{R}^2}, \mathcal{L})$), and a random low frequency signal. Left: graph sizes on the $x$-Axis and error on logarithmic $y$-Axis. Right: $\log_2$ of the graph sizes on the $x$-Axis and $\log_{10}$ of the error on the $y$-Axis. The slope of the curve represents the exponential dependency of the error on $N$.

hence does not satisfy the conditions of Theorems 3.1, $\mathbb{1}_{B_r(x)}(y)$ can be approximated by a Lipschitz continuous function. As the metric-space signal we consider a random low frequency signal (see Figure 2).

For our network, we choose untrained MPNNs with random weights, where each layer is defined using EdgeConv [Bronstein et al., 2017] with mean aggregation, and is implemented using Pytorch Geometric [Fey and Lenssen, 2019]. More precisely, we consider MPNNs with 2 layers. The message function in the first layer is defined as $\Phi^{(1)}(\mathbf{f}_i, \mathbf{f_j}) = h^{(1)}(\mathbf{f}_i, \mathbf{f}_j - \mathbf{f}_i)$, where $h^{(1)}$ is a 1-layered MLP with ReLU activation, input dimension 2 and output dimension 3. The message function in the second layer is defined as $\Phi^{(2)}(\mathbf{f}_i^{(1)}, \mathbf{f_j}^{(1)}) = h^{(2)}(\mathbf{f}_i^{(1)}, \mathbf{f}_j^{(1)} - \mathbf{f}_i^{(1)})$, where $h^{(2)}$ is a 1-layered MLP with ReLU activation, input dimension 6 and output dimension 1. The update functions are given by $\Psi(\mathbf{f}_i^{(1)}, \mathbf{m}_i^{(2)}) = \mathbf{m}_i^{(2)}$. This is a followed by an average pooling layer.

We ran the experiments that depend on random variables 10 times and report the average results with error bars that indicate the standard error. One run consists of the following steps. We consider 10 different graph sequences, where each graph sequence contains randomly sampled graphs of $2^i$ nodes, with $i = 1, \ldots, 13$. We then consider 50 (different) randomly initialized MPNNs, and compute for each graph sequence the worst-case error between the output of the cMPNN to its sampled graphs, i.e., for every graph size $N$, we pick the MPNN with the highest error. We then average the resulting 10 errors over the 10 different graph sequences, to approximate the expected error over the choice of the graph. In Figure 2, we plot the average error over the 10 runs on the logarithmic y-axis and the number of nodes on the x-Axis. We also provide a log-log-graph of this relation. Recall that in a log-log-graph a function of the form $f(x) = x^c$ appears as a line with slope $c$. We observe that in this toy example the worst-case error, which corresponds roughly to the uniform convergence result in Theorem 3.1, decays faster than our theoretical worst-case error bound $-1/6$. This suggests that, at least for band limited signals on random geometric graphs, our convergence bounds are not tight.

Computing the exact cMPNN would involve computing integrals. To approximate this integral, we sampled a large graph from the RGM. For the largest graph, we choose $2^{14}$ nodes. Our smaller graphs consist of $2^i$ nodes, with $i = 1, \ldots, 13$, and are sampled directly from the RGM. As the metric-space signal we consider a discrete random band-limited signal of resolution 256x256, defined as $f = \mathcal{F}^{-1}(v)$, where $v$ consists of randomly chosen Fourier coefficients in the low positive frequency band 20x20 such that the coefficients in the lowest positive frequency band 8x8 are amplified by a factor of 10, and $\mathcal{F}^{-1}$ is the inverse Finite Fourier Transform.

### D.2 Generalization Experiments

In this subsection, we provide details for the numerical experiments from Section 4 and report additional generalization experiments.

Table 2: Readout of the constants depending on the datasets. Each column represents the value of the respective dataset parameter used for the calculations of our generalization bounds.

|  | $\|W\|_\infty$ | $L_W$ | $\|f\|_\infty$ | $f_L$ | $N$ | $m$ | $d_{\min}$ |
|---|---|---|---|---|---|---|---|
| **ER-SBM** | 0.41 | 0.5 | 0.5 | 0.5 | 50 | 100K | 0.25 |
| **ER-EXP** | 0.5 | 1 | 0.5 | 0.5 | 50 | 100K | 0.373 |
| **EXP-SBM** | 0.5 | 1 | 0.5 | 0.5 | 50 | 100K | 0.25 |

### D.2.1 Dataset

We create three different synthetic datasets of random graphs from different random graph models. The domains of the graphons (the metric space), is taken as the Euclidean space $[0, 1]$. First, we consider Erdös-Rényi graphs with edge probably $0.4$ with constant signal, represented by $(W_1, f_1)$ with $W_1(x, y) = 0.4$ and $f_1(x) = 0.5$. We also consider a smooth version of a stochastic block model, represented by $(W_2, f_2)$ with $W_2(x, y) = \sin(2\pi x)\sin(2\pi y)/2\pi + 0.25$ and $f_2(x) = \sin(x)/2$. Last, we consider an exponential radial graphon, represented by $(W_3, f_3)$ with $W_3(x, y) = \exp(-|x - y|^2)/2$ and $f_3(x) = 0.5x$. For each graphon, we create 50K graphs of size 50. We call the Erdös-Renyi dataset **ER**, the stochstic block model dataset **SBM**, and the exponential radial dataset **EXP**. We then consider all possible pairs, i.e., **ER**-**SBM**, **ER**-**EXP** and **SBM**-**EXP**, and train a binary classifier for each pair. We split each dataset to 90% training examples and 10% test.

### D.2.2 MPNN Details

For our network, we choose MPNNs intialized with random weights, where each layer is defined using GraphSage [Hamilton et al., 2017], and is implemented with Pytorch Geometric [Fey and Lenssen, 2019]. We consider MPNNs with 1,2 and 3 layers. The message functions are defined by

$$\Phi^{(l)}(\mathbf{f}_i^{(l-1)}, \mathbf{f}_j^{(l-1)}) = \mathbf{f}_j^{(l-1)}.$$

The update functions are given by

$$\Psi^{(l)}(\mathbf{f}_i^{(l-1)}, \mathbf{m}_i^{(l-1)}) = \rho\big(W_1^{(l)}\mathbf{f}_i^{(l-1)} + W_2^{(l)}\mathbf{m}_i^{(l-1)}\big),$$

where $W_1^{(1)} \in \mathbb{R}^{128 \times 1}$, $W_2^{(1)} \in \mathbb{R}^{128 \times 1}$ and $W_1^{(2)}, W_1^{(3)} \in \mathbb{R}^{128 \times 128}$, $W_2^{(3)}, W_2^{(3)} \in \mathbb{R}^{128 \times 128}$. We then consider a global mean pooling layer, and apply a last linear layer $Q$ (including bias) with input dimension 128 and output dimension 2. This last linear layer is seen as part of the loss function in the analysis, and contributes to the generalization bound via the Lipschitz constant and infinity norm of the loss, as seen in Theorem 3.3.

### D.2.3 Experimental Setup

The loss is given by soft-max composed with cross-entropy (composed on the last MLP). We consider Adam with learning rate $lr = 0.01$. For experiments with weight decay, we use an $l^2$-regularization on the weights with factors $0.27, 0.15$ and $0.05$ for the **ER**-**SBM** dataset. For the **ER**-**EXP** dataset we consider weight decay factors $0.37, 0.15$ and $0.05$. For the **SBM**-**EXP** dataset we consider $0.28, 0.05$ and $0.05$. We train for 1 epoch. The batch size is 64. We consider 1, 2 and 3 layers.

### D.2.4 Details on Computations of Our Bound

We compute our generalization bound according to the full formula given in Theorem C.7. The terms depending on the dataset are: the size of the training dataset $m$, the average graph size $N$, the minimum degree $d$, the largest infinity norm of the graphons $\|W\|_\infty$, largest Lipschitz norm of the graphons $L_W$, the largest infinity norm of the metric-space signal $\|f\|_\infty$, the largest Lipschitz norm of the metric-spaces signals $L_f$ and the number of classes is $\Gamma = 2$. For every dataset, we summarize these terms depending on the dataset in Table 2.

Our bound depend also on the Lipschitz constants of the trained GraphSage MPNN, i.e., on the Lipschitz norms $L_{\Psi^{(l)}}$ and $L_{\Phi^{(l)}}$ of the update function $\Psi^{(l)}$ and message function $\Phi^{(l)}$, given in

Subsection D.2.2. We have $L_{\Phi^{(l)}} = \left\| [W_1^{(l)}, W_2^{(l)}] \right\|_\infty$ and $L_{\Phi^{(l)}} = 1$. We readout the norms $\left\| [W_1^{(l)}, W_2^{(l)}] \right\|_\infty$ for every layer, and plug it into our bound. The bound also depends on the infinity norm and Lipschitz constant of the loss. We compute these constants in the next subsection.

### D.2.5  Computation of the Infinity Norm and Lipschitz Constant of the Loss

Next we bound the Lipschitz constant and infinity norm of the loss. Namely, we derive properties of softmax composed on cross-entropy. Softmax composed with the cross-entropy loss in the case of binary classes take the form

$$\mathcal{L}_{\mathrm{CE}}(\mathbf{x}; \mathbf{y}) = -y_1 \log\left( \frac{e^{x_1}}{e^{x_1} + e^{x_2}} \right) - y_2 \log\left( \frac{e^{x_2}}{e^{x_1} + e^{x_2}} \right),$$

where $\mathbf{x} = (x_1, x_2) \in \mathbb{R}^2$ and $(y_1, y_2) \in \{e_1, e_2\}$ depends on the target label, where $e_1 = (1,0)$ and $e_2 = (0,1)$. When the target label is fixed, we write in short $\mathcal{L}_{\mathrm{CE}}(\mathbf{x}) := \mathcal{L}_{\mathrm{CE}}(\mathbf{x}; \mathbf{y})$.

**Lemma D.1.** *The loss $\mathcal{L}_{\mathrm{CE}}$ is Lipschitz continuous with Lipschitz constant 1. Additionally, $\mathcal{L}_{\mathrm{CE}}$ is locally bounded in the following sense:*

$$\|\mathcal{L}_{\mathrm{CE}}\|_{L^\infty([-K,K]^2)} \leq \log(1 + e^{2K}),$$

*where $\|\mathcal{L}_{\mathrm{CE}}\|_{L^\infty([-K,K]^2)} = \max_{\mathbf{x} \in [-K,K]^2} \|\mathcal{L}_{\mathrm{CE}}(\mathbf{x})\|$.*

*Proof.* We compute

$$\frac{\partial}{\partial x_1} \mathcal{L}_{\mathrm{CE}}(x_1, x_2) = -y_1\left(1 - \frac{e^{x_1}}{e^{x_1} + e^{x_2}}\right) + y_2 \frac{e^{x_1}}{e^{x_1} + e^{x_2}}$$

$$= (y_1 + y_2)\frac{e^{x_1}}{e^{x_1} + e^{x_2}} - y_1$$

$$= \frac{e^{x_1}}{e^{x_1} + e^{x_2}} - y_1$$

Since $\frac{e^{x_1}}{e^{x_1} + e^{x_2}} \in [0, 1]$ and $y_1 \in \{0, 1\}$, this implies

$$\left| \frac{\partial}{\partial x_1} \mathcal{L}_{\mathrm{CE}}(x_1, x_2) \right| \leq 1.$$

By symmetry we conclude that $\mathcal{L}_{\mathrm{CE}}$ is Lipschitz continuous with constant 1.

Last, let $(x_1, x_2) \in [-K, K]^2$ and without loss of generality $y_1 = 1$ and $y_2 = 0$. We have

$$|\mathcal{L}_{\mathrm{CE}}(x_1, x_2)| = -\log\left( \frac{e^{x_1}}{e^{x_1} + e^{x_2}} \right)$$

$$= \log\left( 1 + \frac{e^{x_2}}{e^{x_1}} \right)$$

$$\leq \log\left( 1 + e^{2K} \right).$$

$\square$

The above lemma tells us that in order to bound the infinity norm of the loss we must bound the domain of the loss - the output of the MPNN.

**Lemma D.2.** *Let $\Theta = \left( (\Phi^{(l)})_{l=1}^T, (\Psi^{(l)})_{l=1}^T \right)$ be a MPNN s.t. Assumption 7. is satisfied. Consider a graph with $N$ nodes and a graph feature map $\mathbf{f} \in \mathbb{R}^{N \times F}$. Then,*

$$\|\Theta_G^P(\mathbf{f})\|_\infty \leq A' + A'' \|\mathbf{f}\|_{\infty;\infty},$$

*where*

$$A' = \sum_{l=1}^T \left( L_{\Psi^{(l)}} \|\Phi^{(l)}(0,0)\|_\infty + \|\Psi^{(l)}(0,0)\|_\infty \right) \prod_{l'=l+1}^T L_{\Psi^{(l')}}\left(L_{\Phi^{(l')}} + 1\right)$$

*and*

$$A'' = \prod_{l=1}^T L_{\Psi^{(l)}}\left(1 + L_{\Phi^{(l)}}\right).$$

*Proof.* Let $G$ be a graph with weight matrix $\mathbf{W} = (W_{i,j})_{i,j=1\dots,N}$. Let $l = 0, \ldots, T-1$. Then, for $k = 0, \ldots, l$, we have

$$
\begin{aligned}
\|\mathbf{f}_i^{(k+1)}\|_\infty &= \left\|\Psi^{(k+1)}\left(\mathbf{f}_i^{(k)}, \mathbf{m}_i^{(k+1)}\right)\right\|_\infty \\
&\leq \left\|\Psi^{(k+1)}\left(\mathbf{f}_i^{(k)}, \mathbf{m}_i^{(k+1)}\right) - \Psi^{(k+1)}(0,0)\right\|_\infty + \|\Psi^{(k+1)}(0,0)\|_\infty \qquad (64) \\
&\leq L_{\Psi^{(k+1)}}\left(\|\mathbf{f}_i^{(k)}\|_\infty + \|\mathbf{m}_i^{(k+1)}\|_\infty\right) + \|\Psi^{(k+1)}(0,0)\|_\infty,
\end{aligned}
$$

where $\mathbf{m}_i^{(k+1)} = \frac{1}{\sum_{j=1}^N W_{i,j}} \sum_{j=1}^N W_{i,j} \Phi^{(k+1)}\left(\mathbf{f}_i^{(k)}, \mathbf{f}_j^{(k)}\right)$. For this message term, we have

$$
\begin{aligned}
\|\mathbf{m}_i^{(k+1)}\|_\infty &= \left\|\frac{1}{\sum_{j=1}^N W_{i,j}} \sum_{j=1}^N W_{i,j} \Phi^{(k+1)}\left(\mathbf{f}_i^{(k)}, \mathbf{f}_j^{(k)}\right)\right\|_\infty \\
&\leq \left\|\max_{j=1,\ldots,N} \Phi^{(k+1)}\left(\mathbf{f}_i^{(k)}, \mathbf{f}_j^{(k)}\right)\right\|_\infty \qquad (65) \\
&\leq \max_{j=1,\ldots,N} L_{\Phi^{(k+1)}} \|\mathbf{f}_j^{(k)}\|_\infty + \|\Phi^{(k+1)}(0,0)\|_\infty.
\end{aligned}
$$

Denote $\|\mathbf{f}\|_{\infty;\infty} = \max_{i=1,\ldots,N} \max_{j=1,\ldots,F} |\mathbf{f}_{i,j}|$ for $\mathbf{f} \in \mathbb{R}^{N \times F}$. We have as a result of (64) and (65)

$$
\begin{aligned}
&\|\mathbf{f}^{(k+1)}\|_{\infty;\infty} \\
&\leq L_{\Psi^{(k+1)}}\left(\|\mathbf{f}^{(k)}\|_{\infty;\infty} + \left(L_{\Phi^{(k+1)}}\|\mathbf{f}^{(k)}\|_{\infty;\infty} + \|\Phi^{(k+1)}(0,0)\|_\infty\right)\right) + \|\Psi^{(k+1)}(0,0)\|_\infty
\end{aligned}
$$

which we can write as

$$
\begin{aligned}
&\|\mathbf{f}^{(k+1)}\|_{\infty;\infty} \\
&\leq L_{\Psi^{(k+1)}}\left(1 + L_{\Phi^{(k+1)}}\right)\|\mathbf{f}^{(k)}\|_{\infty;\infty} + L_{\Psi^{(k+1)}}\|\Phi^{(k+1)}(0,0)\|_\infty + \|\Psi^{(k+1)}(0,0)\|_\infty.
\end{aligned}
$$

We apply Lemma B.11 to solve this recurrence relation, to get

$$
\begin{aligned}
\|\mathbf{f}^{(k)}\|_{\infty;\infty} &\leq \sum_{l=1}^k \left(L_{\Psi^{(l)}}\|\Phi^{(l)}(0,0)\|_\infty + \|\Psi^{(l)}(0,0)\|_\infty\right) \prod_{l'=l+1}^k L_{\Psi^{(l')}}(1 + L_{\Phi^{(l')}}) \\
&\quad + \|\mathbf{f}^{(0)}\|_{\infty;\infty} \prod_{l=1}^k L_{\Psi^{(l)}}(1 + L_{\Phi^{(l)}})
\end{aligned}
$$

Now, since for general bounded functions $F : \chi \to \mathbb{R}^n$ and $x_1, \ldots x_N \in \chi$

$$
\left\|\frac{1}{N}\sum_{i=1}^N F(x_i)\right\|_\infty \leq \|F\|_{\infty,\infty},
$$

the proof is done. $\qquad\square$

Note that using our analysis, for the MPNN architecture presented in Section D.2.2, the loss is not just $\mathcal{L}_{\mathrm{CE}}$, but the composition of $\mathcal{L}_{\mathrm{CE}}$ on the last linear layer of the network. We denote this total loss by $\mathcal{L}_{\mathrm{total}} = \mathcal{L}_{\mathrm{CE}} \circ Q$. Hence, in our analysis the Lipschitz constant of the total loss is bounded by

$$
L_{\mathcal{L}_{\mathrm{total}}} = \|Q\|_\infty,
$$

where $\|Q\|_\infty$ is the induced infinity norm of the matrix $Q$. The infinity norm bound of the total loss is bounded by

$$
\|\mathcal{L}_{\mathrm{total}}\|_\infty \leq \log(1 + e^{2(\|Q\|_\infty K + b)}),
$$

where $K$ is the infinity norm of the MPNN.

### D.2.6 Details on the Computation of Bounds from Other Papers

The papers [Liao et al., 2021] and [Garg et al., 2020] do not provide generalization bounds for general MPNNs, but only for a specific architecture – GNNs with mean field updates, as defined in Definition C.8, namely

$$\mathbf{f}_i^{(l)} = \Psi\left(W_1\mathbf{x}_i + W_2\rho\left(\sum_{u\in\mathcal{N}(v)}\Phi(\mathbf{f}_j^{(l-1)})\right)\right),$$

where $\rho$, $\Psi$ and $\Phi$ are nonlinear transformations, and $W_1$ and $W_2$ are linear transformations. This is followed by a global pooling layer, which takes as an input $\mathbf{f}^{(T-1)} \in \mathbb{R}^{N\times K}$, and returns the vector

$$\frac{1}{N}\mathbf{1}_N\mathbf{f}^{(T-1)}W_T \in \mathbb{R}^{1\times K},$$

where $W_T$ is a linear transformation. Here $\mathbf{1}_N$ denotes the vector $(1,\ldots,1) \in \mathbb{R}^{1\times N}$, where $N$ is the number of nodes in the graph. As described in Subsection C.4.1, GNNs with mean field updates are a special case of MPNNs.

The generalization bounds in [Liao et al., 2021] and [Garg et al., 2020] are formulated in terms of the following constants: $\zeta = \min\left(\|W_1\|_2, \|W_2\|_2, \|W_T\|_2\right)$, $|w|_2^2 = \|W_1\|_F^2 + \|W_2\|_F^2 + \|W_T\|_F^2$, $\lambda = \|W_1\|_2\|W_T\|_2$, $\xi = L_\Psi\frac{(d\mathcal{C})^{l-1}-1}{d\mathcal{C}-1}$, and the percolation complexity $\mathcal{C} = L_\Psi L_\rho L_\Phi\|W_2\|_2$, where $L_\Psi, L_\rho$ and $L_\Phi$ are the Lipschitz constants of $\Psi, \rho$ and $\Phi$. For the calculation of the generalization bounds, we use the fully non-asymptotic generalizations bounds, provided in [Liao et al., 2021, Subsection A.7]. There, the PAC-Bayesian based bound is given by

$$\sqrt{\frac{42^2B^2\left(\max\left(\eta^{-(T+1)},(\lambda\zeta)^{\frac{T+1}{T}}\right)\right)^2 T^2 h\log(4Th)|w|_2^2}{\gamma^2 m}}. \tag{66}$$

The Rademacher based bound is given by

$$48h\|W_T\|_2 Z\sqrt{\frac{3\log\left(24\|W_T\|_2\sqrt{m}\max\left(Z, M\sqrt{h}\max(B\|W_1\|_2, \bar{R}\|W_2\|_2)\right)\right)}{\gamma^2 m}}. \tag{67}$$

Note that GraphSage cannot be described in terms of mean field update networks, and vice versa. In order to still report some comparison between the generalization bounds, we offer some conversion between the constants of the two methods, and then apply the PAC-Bayes and Rademacher bounds on the converted bounds. It should be noted that the comparison is a bit like "comparing apples to oranges," but still gives insight into the respective bounds, their asymptotics, and their usefulness in practical situations.

Since the transformation by $\Psi(W_1(\cdot) + W_2\circ\rho(\cdot))$ can be seen as an update function, similarly to the one in GraphSage, we set in the PAC-Bayes bound $\|W_1\|_2 = \|W_2\|_2 = 1/T\sum_{l=1}^T L_{\Psi^{(l)}}$, where $L_{\Psi^{(l)}}$ is the Lipschitz constant of the update function of GraphSage in the $l$-th layer. The message function in GraphSage is the identity, which corresponds to $\Phi$. We thus convert this to $L_\Phi = 1$ in the PAC-Bayes generalization bound. Finally, we give a lower bound for the maximum degree over all graphs in the datasets by setting $d = Nd_{\min}$ (note that the PAC-Bayes and Rademacher complexity based bounds increase with increasing maximum degree).

### D.2.7 Generalization Comparison Results

The results are reported in Figure 3. The different experimental setting are given on the x-Axis. We report experiments for MPNNs with depth $T = 1, 2, 3$ with weight decay (WD) and without weight decay (w/o WD). The bound values are reported in a logarithmic y-Axis to improve comparability. In addition to the figures, we also provide numerical values of the bound calculations in Table 3.

Our generalization bound is tighter than the PAC-Bayes bound and the Rademacher bound under all settings, i.e., for all datasets, for all depths, with weight decay and also without weight decay.

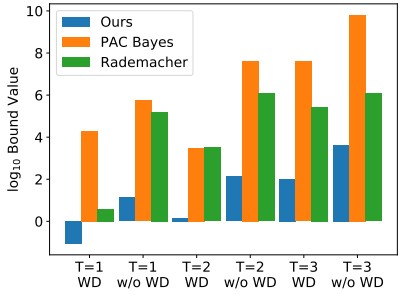

i) Generalization bounds on **ER**-**SBM**

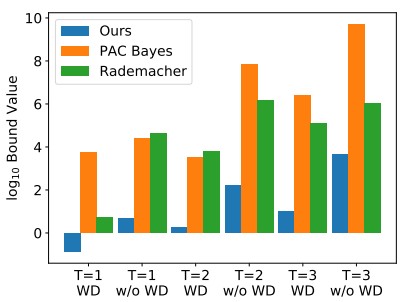

ii) Generalization bounds on **ER**-**EXP**

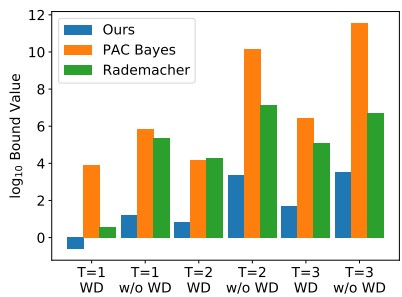

iii) Generalization bounds on **EXP**-**SBM**

Figure 3: The generalization bounds on all datasets, i.e., i) **ER-SBM**, ii) **ER-EXP** and iii) **EXP-SBM**, for different number of layers $T = 1, 2, 3$ with weight decay (WD) and without weight decay (w/o WD). Bounds are given in a $\log_{10}$-scale.

## D.3 Additional Comparison of the Generalization Bounds

In this subsection, we present additional plots of the generalization bounds which showcase the dependency on the average graph sizes in the dataset. The parameters in these plots are set not for a specific dataset and trained network. The plots can be interpreted as the bounds corresponding to training with certain constraints or regularization terms leading to the respective constants (Lipschitz bounds and infinity norms).

We consider a theoretical setting in which we assume that the following parameters are given: The dataset has 50K graphs, randomly sampled from RGMs with graphons that have maximum infinity norm $\|W\|_\infty = 0.4$ and Lipschitz norm $L_W = 0.5$. We assume that the metric-space signal are bounded by $0.5$ and have Lipschitz constants of maximum $0.5$. Furthermore, we assume there is a linear layer after pooling such that the norms of weight matrix and of the bias are upper bounded by $0.5$ and $0.1$, respectively. The infinity and Lipschitz norms of the loss function are assumed to be bounded by $1$.

We then consider different datasets with graphs of average size $N = 2^4, 2^5, \ldots, 2^{25}$. Since the PAC-Bayes and Rademacher generalization bounds scale with the maximum node degree $d$ of the graphs, we estimate the degree by setting $d = N \cdot d_{\min}$, where $d_{\min}$ is the graphon degree. We report our generalization bound with respect to the graph size in Figure 4. The comparison with other generalization bounds is given in Figure 5. As expected by our theoretical results, our generalization bound decays with respect to the average graph size. In contrast, we see that both the PAC Bayes based bound and the Rademacher based bound increase with respect to the increasing graph size.

In Figure 6 we showcase the dependency of our generalization bound on the Lipschitz constant of the graphons. For this, we fix the graph sizes in the dataset to 1000, and compute the resulting bounds for increasing Lipschitz norms. The rest of the parameters are as specified above. We plot the generalization bound for MPNNs with depth $1, 2$ and $3$ in Figure 6.

Table 3: Bound comparisons on all synthetic datasets.

| T = 1 WD | ER - SBM | ER - EXP | SBM - EXP |
|---|---|---|---|
| Rademacher | $3.9597 \times 10^{0}$ | $5.5278 \times 10^{0}$ | $3.6869 \times 10^{0}$ |
| PAC-Bayesian | $1.9597 \times 10^{4}$ | $5.8187 \times 10^{3}$ | $7.8245 \times 10^{3}$ |
| Ours | $\mathbf{8.8373 \times 10^{-2}}$ | $\mathbf{1.325 \times 10^{-1}}$ | $\mathbf{2.4495 \times 10^{-1}}$ |
| **T = 1 w/o WD** | | | |
| Rademacher | $1.7329 \times 10^{5}$ | $4.3686 \times 10^{4}$ | $2.2311 \times 10^{5}$ |
| PAC-Bayesian | $6.0161 \times 10^{5}$ | $2.6146 \times 10^{4}$ | $7.2969 \times 10^{5}$ |
| Ours | $\mathbf{1.3534 \times 10^{1}}$ | $\mathbf{4.4827 \times 10^{0}}$ | $\mathbf{1.7109 \times 10^{1}}$ |
| **T = 2 WD** | | | |
| Rademacher | $3.2439 \times 10^{3}$ | $6.3963 \times 10^{3}$ | $1.9428 \times 10^{4}$ |
| PAC-Bayesian | $3.0695 \times 10^{3}$ | $3.2992 \times 10^{3}$ | $1.4299 \times 10^{4}$ |
| Ours | $\mathbf{1.2817 \times 10^{0}}$ | $\mathbf{1.7047 \times 10^{0}}$ | $\mathbf{5.4857 \times 10^{0}}$ |
| **T = 2 w/o WD** | | | |
| Rademacher | $1.1943 \times 10^{6}$ | $1.4238 \times 10^{6}$ | $1.3619 \times 10^{7}$ |
| PAC-Bayesian | $4.0392 \times 10^{7}$ | $6.9262 \times 10^{7}$ | $1.3817 \times 10^{10}$ |
| Ours | $\mathbf{1.1875 \times 10^{2}}$ | $\mathbf{1.478 \times 10^{2}}$ | $\mathbf{1.9353 \times 10^{3}}$ |
| **T = 3 WD** | | | |
| Rademacher | $2.7221 \times 10^{5}$ | $1.2286 \times 10^{5}$ | $1.2529 \times 10^{5}$ |
| PAC-Bayesian | $3.9963 \times 10^{7}$ | $2.6016 \times 10^{6}$ | $2.6141 \times 10^{6}$ |
| Ours | $\mathbf{8.1312 \times 10^{1}}$ | $\mathbf{8.1904 \times 10^{0}}$ | $\mathbf{2.7574 \times 10^{1}}$ |
| **T = 3 w/o WD** | | | |
| Rademacher | $1.1762 \times 10^{6}$ | $1.0872 \times 10^{6}$ | $4.8271 \times 10^{6}$ |
| PAC-Bayesian | $6.225 \times 10^{9}$ | $5.1689 \times 10^{9}$ | $3.7028 \times 10^{11}$ |
| Ours | $\mathbf{3.1606 \times 10^{3}}$ | $\mathbf{3.5428 \times 10^{3}}$ | $\mathbf{1.8596 \times 10^{3}}$ |

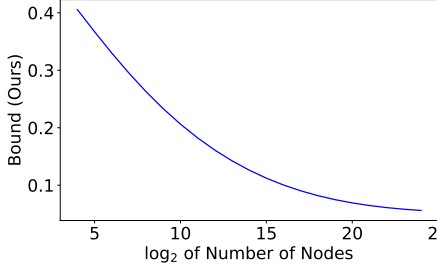 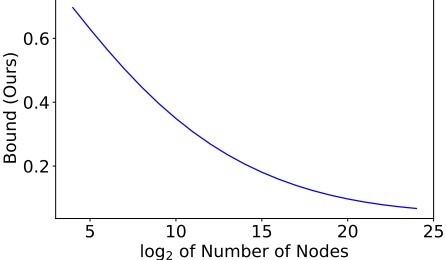

Figure 4: Our generalization bounds with respect to increasing average graph sizes. On the x-axis we give the average number of nodes in the dataset in $\log_2$-scale. On the y-axis, we give our generalization bound.

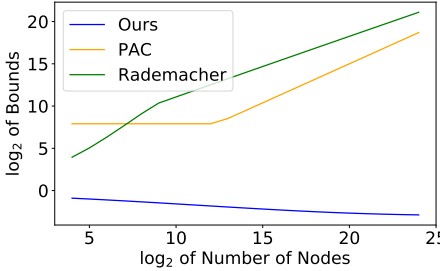 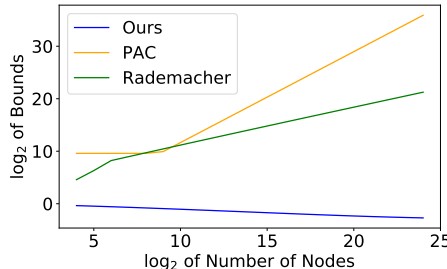

Figure 5: The generalization bounds with respect to increasing average graph sizes. On the x-axis, we give the average number of nodes in the dataset in $\log_2$-scale. On the y-Axis, we give our generalization bound, the PAC-Bayes based bound and the Rademacher complexity based bound for MPNNs with depth 2 (left) and depth 3 (right) also in $\log_2$-scale.

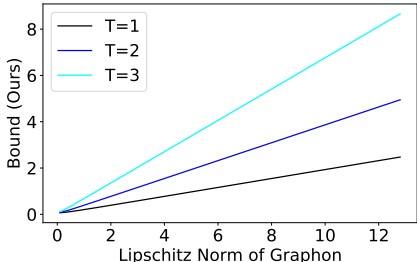

Figure 6: Our generalization bounds with respect to increasing average Lipschitz norm of the graphon. On the x-axis we give the maximal Lipschitz norm of the graphons from which we sampl the dataset. On the y-axis, we give our generalization bound. The rest of the parameters are equal to the parameters in the setting of Figure 5 (see Subsection D.3) with fixed graph size $N = 1000$.

# E   Background in Random Processes

In this section, we provide background information in probability theory, and focus on random processes and concentration of measure inequalities.

**Definition E.1** (Definition 7.1.1. in [Vershynin, 2018]). *A random process is a collection of random variables $(Y_t)_{t \in T}$ on the same probability space, which are indexed by the elements $t$ of some set $T$.*

The following lemma provides an upper bound on the probability that the sum of bounded independent random variables deviates from its expected value by more than a certain amount.

**Theorem E.2** (Hoeffding's Inequality). *Let $Y_1, \dots, Y_N$ be independent random variables such that $a \leq Y_i \leq b$ almost surely. Then, for every $k > 0$,*

$$\mathbb{P}\Big( \Big| \frac{1}{N} \sum_{i=1}^{N} (Y_i - \mathbb{E}[Y_i]) \Big| \geq k \Big) \leq 2 \exp \Big( - \frac{2k^2 N}{(b-a)^2} \Big).$$

**Definition E.3** (Definition 2.5.6 in [Vershynin, 2018]). *A random variable $Y$ is called a sub-Gaussian random variable if there exists a constant $K \in \mathbb{R}$ such that $\mathbb{E}\big[ \exp\big( Y^2/K^2 \big) \big] \leq 2$. The sub-Gaussian norm of a sub-Gaussian random variable $X$ is defined as*

$$\|Y\|_{\psi_2} = \inf \Big\{ t > 0 : \mathbb{E}\big[ \exp\big( Y^2/t^2 \big) \big] \leq 2 \Big\}.$$

**Lemma E.4** (Example 2.5.8 in [Vershynin, 2018]). *Any bounded random variable $Y$ is sub-Gaussian with*

$$\|Y\|_{\psi_2} \leq \frac{1}{\sqrt{\ln(2)}} \|Y\|_{\infty}.$$

**Definition E.5** (Sub-Gaussian increments, Definition 8.1.1 in [Vershynin, 2018]). *Consider a random process $(Y_x)_{x \in \chi}$ on a metric space $(\chi, d)$. We say that the process has* sub-Gaussian increments *if there exists a constant $K \geq 0$ such that*

$$\|Y_x - Y_{x'}\|_{\psi_2} \leq K d(x, x')$$

*for all $x, x' \in \chi$. We call $(\|Y_x - Y_{x'}\|_{\psi_2})_{x, x' \in \chi}$ the sub-Gaussian increments of $(Y_x)_{x \in \chi}$.*

**Lemma E.6** (Centering of sub-Gaussian random variables, Lemma 2.6.8 in [Vershynin, 2018]). *If $Y$ is a sub-Gaussian random variable, then so is $Y - \mathbb{E}[Y]$, and*

$$\|Y - \mathbb{E}[Y]\|_{\psi_2} \leq \Big(\frac{2}{\ln(2)} + 1\Big)\|Y\|_{\psi_2}.$$

**Lemma E.7** (Proposition 2.6.1 in [Vershynin, 2018]). *Let $Y_1, \ldots, Y_N$ be independent mean-zero sub-Gaussian random variables. Then, $\sum_{i=1}^{N} Y_i$ is also a sub-Gaussian random variable, and*

$$\|\sum_{i=1}^{N} Y_i\|_{\psi_2}^2 \leq \frac{2}{\sqrt{2}} e \sum_{i=1}^{N} \|Y_i\|_{\psi_2}^2.$$

**Theorem E.8** (Dudley's Inequality, Theorem 8.1.6 in [Vershynin, 2018]). *Let $(Y_x)_x$ be a random process on a metric space $(\chi, d)$ with sub-Gaussian increments, i.e., there exists a $K \geq 0$ such that $\|Y_x - Y_{x'}\|_{\psi_2} \leq K d(x, x')$ for all $x, x' \in \chi$. Then, for every $u \geq 0$, the event*

$$\sup_{x, x' \in \chi} |Y_x - Y_{x'}| \leq CK \Big(\int_0^{\infty} \sqrt{\log \mathcal{C}(\chi, \varepsilon, d)} d\varepsilon + u \mathrm{diam}(\chi)\Big)$$

*holds with probability at least $1 - 2\exp(-u^2)$, where $\mathcal{C}(\chi, \varepsilon, d)$ is defined in Definition A.1 and $C$ is a universal constant, specified in [Vershynin, 2018, Chapter 8].*