# OpenReview forum: "Generalization Analysis of Message Passing Neural Networks on Large Random Graphs"
_NeurIPS.cc/2022/Conference — NeurIPS 2022 Accept_

### Official Review · Reviewer_zvEs · 2022-07-09

**Rating:** 5
**Confidence:** 3
**Soundness:** 3 good
**Presentation:** 2 fair
**Contribution:** 3 good

**Summary:**

In this work, a new generalisation bound for MPNNs is proposed. Specifically, studying MPNNs over graphs and their continuous-space counterparts, it is showed that the uniform generalisation error can be upper-bounded by a formula that improves, not only by increasing the number of training graphs, but also by increasing the number of nodes in these graphs. This forms the foundation of a useful 'peace-of-mind' result: a MPNN-based graph classifier should generalise well even over small numbers of training graphs, provided those graphs are sufficiently large. The bound is verified empirically by training MPNNs on a random geometric graph model.

**Questions:**

Q1. Could you concisely clarify the practical implications of your work? Could you make this discussion clear in your paper from an early stage?

Q2. Could you provide additional experimental evidence for your bounds? Either on a wider variety of generative distributions, or (ideally) on several 'points' corresponding to sizes / distributions of existing graph classification datasets in the literature.

**Limitations:**

No concerns.

**Strengths And Weaknesses:**

**Originality**:
The derived bounds appear novel, and compared to prior art (Table 1), they feature for the first time a term that decays as the graph size increases. This is enabled through an original Monte Carlo lens of analysis, which allows to treat individual nodes as samples, even though the task is specified on the graph level. Specially, a novel mathematical apparatus is developed to handle the correlations between the nodes' neighbourhoods (since the nodes cannot be assumed to be provided i.i.d.).

**Quality**:
I find the mathematical results in this work to be solid. However, I believe the experimental evaluation could benefit from some aspects of real-world analysis -- see the significance section for more details.

**Clarity**:
The paper is, for the most part, well-written and somewhat easy to follow. However, I find that the paper could really benefit with making its aims (and practical implications) clear significantly earlier. As it stands, it takes until halfway of page 2 before it is even slightly clear what the implications of this work would be for the GNN practitioner, and even then the link feels a bit loose and 'left to the reader'. It would be highly beneficial to provide a clearer discussion on what the work's main contributions and its implications are, way earlier in the paper.

**Significance**:
While the obtained bounds are clearly of significance compared to earlier work in the area, I find the experimental evaluation to be fairly limited in comparison---focusing only on one specific generative model of graphs. It will strengthen the results of the paper significantly if we could connect them to 'real-world' distributions of graphs.

For example, one might perform a similar experiment to Section 4, over the various graph classification datasets in the TU Dataset benchmark (https://chrsmrrs.github.io/datasets/). For any one of those datasets, we can observe the (average) number of nodes, and compare the observed generalisation error w.r.t. an (empirically built) RGM over the graphs in the dataset. This would correspond to a particular 'point on the curve' of Figure 2, specialised on a particular dataset's generative distribution.

**Overall**:
All things taken into account, I think this has potential to be a strong paper, and a few more improvements would be needed before it would pass the bar of acceptance. Theoretically, it appears very solid (though I will remark I did not check the proofs in the Appendix at great length), but practically it is somewhat lacking in evaluation and motivation.

---

> ### Author Response · Authors · 2022-08-02
> **Response to Reviewer zvEs (1/2)**
>
> We are very grateful to the reviewer for the time taken to carefully assess our work and for the valuable feedback. We address each point individually. “Q” quotes the Reviewer and “A” marks the response by the authors:
>
> > **Q:**  *Quality: “... I believe the experimental evaluation could benefit from some aspects of real-world analysis ...”*
>
> **A:** Thank you for this important comment. We agree that the numerical experiments were lacking in the first version of this paper. We revised the paper and added more extensive experiments, comparing our generalization bound in various synthetic datasets to other bounds from the literature. The experiments clearly show that our bound is more realistic, and is orders of magnitude lower than other bounds.
>
> We follow the experiments done in the PAC-Bayes generalization paper  [LUZ21]. In the experiment, we learn a classifier on graphs sampled from synthetic random graph models. We compare our generalization bound to the PAC-Bayes [LUZ21] and Rademacher [GJJ20] bound. We consider MPNNs with 1,2, and 3 layers. In Item 1 in our Common Response, we showcase the generalization bounds in one of the datasets. The full results can be seen in Table 3, Page 59, and in Figure 3, Page 58 of the revised Appendix D.2. The results show how weight decay regularization leads to better generalization performance. This supports our theory as explained on Page 8, Lines 304-306:
>
> “Another insight is that the generalization bound becomes smaller the smaller the Lipschitz constants of the message and update functions are (see Remark 3.4). This indicates that regularization methods like weight decay promote generalization.”
>
> It is important to note that theoretical generalization bounds typically teach us about the asymptotic behavior of generalization, and which hyperparameters affect generalization, but they do not give numerical bounds which are realistic and less than 1. Nevertheless, in three of the scenarios that we studied our generalization bound gives realistic bounds, which are near 0.1 (0.089, 0.133 and 0.246). In comparison, all other previous works produce generalization bounds, which are orders of magnitude larger than 1, making them impractical. We believe that this is the first real GNN learning experiment for which a theory actually proves generalization.
>
>
> >**Q:** *Clarity: “... I find that the paper could really benefit with making its aims (and practical implications) clear significantly earlier...”*
>
> **A:** Thank you for this suggestion. We revised the paper accordingly and added on Page 2 top, Lines 38-42, right after the general intro to GNNs, the following paragraph:
>
> “Previous works developed generalization bounds that do not depend on any model of the data, namely, graphs in these works can be generated and labeled in any arbitrary way [STH18, GJJ20, LUZ21]. In this work, we consider a generative model for the graphs which is theoretically powerful and general on the one hand, and allows much tighter generalization bounds on the other hand.”
>
> We then further motivated our model in the new Subsection 1.1, where we justified the validity of our random graph model assumption. We explain how graphons, as generative models for graphs, are very general. This is justified by the fact that similarity in the graphon sense is a relaxation of the combinatorial similarity of graph isomorphism. Hence, graphons are natural generators of dense graphs, and graphs that sample the same graphon (with the same number of nodes) are "almost isomorphic." Moreover, since MPNNs cannot distinguish between isomorphic graphs, if a MPNN is to have any chance to classify graphs, graphs which sample the same graphon should come from the same class. We write: “The main assumption in our analysis is that graphs that are sampled from the same graphon belong to the same class. While this may seem like a limitation, it is actually a very mild and reasonable assumption…” Please see Subsection 1.1 from Page 2, Line 77 in the revised paper for more details.
>
> We also added a sentence on Page 8 about a practical implication of our bound for practitioners.
>
> “Another insight is that the generalization bound becomes smaller the smaller the Lipschitz constant of the message and update function (see Remark 3.4). This indicates that regularization methods like weight decay promote generalization.”
>
> >**Q:** *Overall: … but practically it is somewhat lacking in evaluation and motivation.*
>
> **A:** We thank the reviewer for these important comments, which we fully implemented in the revised version of the paper. Since we were able to comply with the points raised by the reviewer, we hope this will convince the reviewer to improve their overall assessment of the paper.

---

> > ### Author Response · Authors · 2022-08-02
> > **Response to Reviewer zvEs (2/2)**
> >
> > [Lov12] L. Lovász. Large networks and graph limits. Colloquium Publications, Budapest, 2012.
> >
> > [Lov67] L. Lovász. Operations with structures. Acta Mathematica Academiae Scientiarum Hungarica, 18(3):321–328, 1967.
> >
> > [Pen03] M. Penrose. Random geometric graphs. Oxford Scholarship Online, 2003
> >
> > [STH18] F. Scarselli, A. C. Tsoi, and M. Hagenbuchner. The Vapnik–Chervonenkis dimension of graph and recursive neural networks. Neural Networks, 108:248–259, 2018.
> >
> > [GJJ20] V. Garg, S. Jegelka, and T. Jaakkola. Generalization and representational limits of graph neural networks. In Hal Daumé III and Aarti Singh, editors, Proceedings of the 37th International Conference on Machine Learning, volume 119 of Proceedings of Machine Learning Research, pages 3419–3430. PMLR, 13–18 Jul 2020.
> >
> > [LUZ21]  R. Liao, R. Urtasun, and R. Zemel. A PAC-Bayesian approach to generalization bounds for graph neural networks. In International Conference on Learning Representations, 2021.

---

> > ### Comment · Reviewer_zvEs · 2022-08-08
> > **Rebuttal acknowledged**
> >
> > Thank you for your reply and the careful effort you've put in. I find that the paper's message is somewhat strengthened now, and the provided additional experiments significantly strengthen the paper's findings.
> >
> > While I don't find that my proposal was thoroughly followed (i.e. the setup still focuses on classifying across random graph models, rather than being grounded in a real-world dataset), I also don't think it's the responsibility of the authors to set a new precedent, especially if previous bound-oriented papers followed a similar direction.
> >
> > I am upgrading my score to a 5. Good luck!

---

### Official Review · Reviewer_6gwZ · 2022-07-10

**Rating:** 7
**Confidence:** 2
**Soundness:** 3 good
**Presentation:** 3 good
**Contribution:** 3 good

**Summary:**

This paper talks about the generalization error of message passing neural networks.

By assuming graphs are sampled from random graph models (RGM), the authors define continuous MPNN (cMPNN), and prove in Theorem 3.1 that the error between cMPNN and the corresponding graph MPNN decreases as the number of nodes in the graph increases, which is the main contrinution of this paper. With such results, the paper further proves a generalization bound in Theorem 3.3.

Numerical results confirm the correctness of the theoretical analysis.

**Questions:**

* Citation format is not consistent with NeurIPS 2022, need to follow the template provided.

* This paper builds its theoretical results with several assumptions. It would be great to see some discussions about which assumptions are reasonable, and which assumptions might be too strong to deviate from reality.

**Limitations:**

As mentioned by the authors, this is a theoretical work with no clear negative social impact.

The two limitations in this work are concluded in section 5 of the paper.

**Strengths And Weaknesses:**

Strengths:

* The theoretical analysis in this paper is quite solid, which first builds the relationship between graph MPNN and continuous MPNN, then states the convergence and generalization results. Even though this paper is very theoretical and not easy to read, the definitions and preliminary knowledges in this paper are written clearly, which provides backgrounds to readers not very familiar with the theory involved in this paper.

* The convergence bound, which decreases as number of nodes increases, seems novel compared with previous bounds.


Weakness:

* One of the main contributions of the paper is that the generalization bounds becomes smaller as the number of nodes in the graphs becomes larger. This is indeed an improvement over previous bounds. However, as a practitioner, it is not clear to me what insights such theoretical work could bring to reality. For example, how could we utilize the theories in this paper to guide the design of MPNNs with lower generalization error?

---

> ### Author Response · Authors · 2022-08-02
> **Response to Reviewer 6gwZ**
>
> We are very grateful to the reviewer for the time taken to carefully assess our work and for the valuable feedback. We address each point individually. “Q” quotes the Reviewer and “A” marks the response by the authors:
>
> > **Q:** *“... However, as a practitioner, it is not clear to me what insights such theoretical work could bring to reality. For example, how could we utilize the theories in this paper to guide the design of MPNNs with lower generalization error?”*
>
>  **A:** Thank you for this point. We revised the paper accordingly and explained on Page 8, Lines 304-306, how regularizing the weights of the message and update functions promotes generalization:
>
>  ‘’Another insight is that the generalization bound becomes smaller the smaller the Lipschitz constants of the message and update functions (see Remark 3.4). This indicates that regularization methods like weight decay promote generalization.”
>
> This claim is backed up by new experiments that we added to the paper, comparing our generalization bound in various synthetic datasets to other bounds from the literature. The experiments show how adding weight decay to the loss improves generalization (see Section 4 and Appendix D).
>
>  In Section 4 and Appendix D we report new experiments that compare our bound to the PAC-Bayes and Rademacher complexity bounds. The results clearly show that our bounds are tighter - they are orders of magnitude smaller. In fact, theoretical generalization bounds typically teach us about the asymptotic behavior of generalization, and which hyperparameters affect generalization, but they rarely give numerical bounds which are realistic and less than 1. Nevertheless, in three of the scenarios that we studied our generalization bound gives realistic bounds, which are near 0.1 (0.089, 0.133 and 0.246). In comparison, all other previous works produce generalization bounds, which are orders of magnitude larger than 1, making them impractical. We believe that this is the first real GNN learning experiment for which a theory actually proves generalization. We also believe that future works, by us and others, will build upon our approach and gradually improve the generalization bounds. One of the ultimate goals is to derive tight generalization bounds that would be able to guarantee generalization in practical learning scenarios. The current work is one of the first steps in this journey.
>
>
> > **Q:** *“Citation format is not consistent with NeurIPS 2022, need to follow the template provided.”*
>
> **A:** Thank you for this comment, but we, unfortunately, could not find the issue. Up to our knowledge, we can use any bibliography style and citation format as long as we keep it consistent. Please let us know if we are wrong, and where our error is, so we can correct it.
>
> > **Q:** *This paper builds its theoretical results with several assumptions. It would be great to see some discussions about which assumptions are reasonable, and which assumptions might be too strong to deviate from reality.*
>
> **A:** Thank you for this important comment. We added a discussion that justifies the validity of our random graph model assumption in the revised paper in Subsection 1.1 from Page 2, Line 77 to Page 3, Line 98. We explain how graphons as generative models for graphs are very general. This is justified by the fact that similarity in the graphon sense is a relaxation of the combinatorial similarity of graph isomorphism. Hence, since MPNNs cannot distinguish between isomorphic graphs, if a MPNN is to have a chance to classify graphs, graphs which sample the same graphon should come from the same class.  We write: “The main assumption in our analysis is that graphs that are sampled from the same graphon belong to the same class. While this may seem like a limitation, it is actually a very mild and reasonable assumption…” Please see Subsection 1.1 from Page 2, Line 77 in the revised paper for more details.
>
> Last, in the conclusion of the paper, we discuss which of the assumptions we see as limiting, and suggest future directions for extensions (Page 9, Line 352).
>
> “First, the dependency of the generalization bound on the size of the graph $N$ is $\mathcal{O}(N^{-\frac{1}{2(D_{\chi}+1)}})$, which is typically slower than the observed decay in experiments  (See Appendix D.1). One potential future direction is to improve this dependency using a more sophisticated model of the trained network and of the message and update functions. Secondly, our model of the data is somewhat limited. One future direction is to allow deformations of the RGMs, to consider a continuum of RGMs instead of a finite set, and to consider sparse graphs.”
>
> ---
>
>  [Lov12] L. Lovász. Large networks and graph limits. Colloquium Publications, Budapest, 2012.
>
>  [Lov67] L.  Lovász. Operations with structures. Acta Mathematica Academiae Scientiarum Hungarica, 18(3):321–328, 1967.
>
>  [Pen03] M. Penrose. Random Geometric Graphs. Oxford Scholarship Online, 2003

---

> > ### Comment · Reviewer_6gwZ · 2022-08-10
> > **Response to the authors**
> >
> > Thanks for your detailed replies! The explanation for the importance of different assumptions helps me understand the paper better. My concerns have been addressed. And I will raise my score accordingly.

---

> ### Author Response · Authors · 2022-08-09
> **Reminder About the Rebuttal Discussion**
>
> Dear Reviewer 6gwZ ,
>
> We would like to kindly encourage you to read and comment about our rebuttal. We believe that the paper is much improved as a results of the reviewer's suggestions - more experiments, practical insights, and a discussion that supports the validity of the theoretical assumptions.
>
> The other two reviewers acknowledged the improvement in the paper and upgraded their scores. We are hopeful that the revised paper will also meet your expectations.
>
> Kind regards,
>
> The authors

---

### Official Review · Reviewer_tNHF · 2022-07-11

**Rating:** 7
**Confidence:** 2
**Soundness:** 3 good
**Presentation:** 3 good
**Contribution:** 3 good

**Summary:**

A novel generalization bound for a class of message passing neural networks (MPNNs) on graphs is derived. The generalization gap decreases with the average number of nodes of graphs sampled from the data distribution. This is in contrast to other generalization gaps in the literature, which typically increase in the number of nodes or degree. This result relies in part on a convergence result that the authors prove, which bounds the expected difference of an MPNN output on sampled graphs and the output of a newly-defined continuous MPNN on the underlying random graph model.


**Questions:**

1. In Appendix D, it is mentioned that you approximate the continuous MPNN with an MPNN on a large graph (lets call this output $Y \in \mathbf{R}^F$), which is then subsampled to smaller graphs, for which the ''finite'' MPNN outputs (call these $Z_1, \ldots, Z_{50} \in \mathbf{R}^F$) are compared. This may make it easier for the ''finite'' MPNNs to be close to $Y$. Perhaps it would be better to sample several large graphs, giving outputs $Y_1, \ldots, Y_{a} \in \mathbf{R}^F$, and then measure how close ''finite'' MPNNs are to $\frac{1}{a} \sum_{i=1}^a Y_i$ on independently sampled small graphs.

2. Perhaps mention in Section 1 that your analysis holds for mean aggregation (apologies if I have missed the mention).


**Limitations:**

Yes, limitations are mentioned in the conclusion.


**Strengths And Weaknesses:**

Strengths:
1. The derived generalization error contains a term that decreases in the size of the  graph, in contrast to previous results in which error increases in the size of the graph. This is novel, and may have important implications.
2. The paper defines a continuous analogue of message passing neural networks with mean aggregation, an important class of GNNs. Previous work has done this for spectral GNNs that use polynomial filters of the normalized graph Laplacian. Unlike previous work, this work develops convergence bounds that are uniform in the choice of model, which is important for deriving useful generalization bounds.
3. The assumptions and data distribution mostly seem reasonable to me.

Weaknesses:
1. Minor:  numerical experiments are a bit lacking, and it may be interesting to consider things like Lipschitz continuous graphons of varying constants, or other GNNs. See also the questions section.

Note: I did not have time to check the proofs of this paper, which are very long and full of calculations.

---

> ### Author Response · Authors · 2022-08-02
> **Response to Reviewer tNHF**
>
> We are very grateful to the reviewer for the time taken to carefully assess our work and for the valuable feedback. We address each point individually. “Q” quotes the Reviewer and “A” marks the response by the authors:
>
> > **Q:** *Minor Weakness: “numerical experiments are a bit lacking, and it may be interesting to consider things like Lipschitz continuous graphons of varying constants, or other GNNs. See also the questions section.”*
>
> **A:** We agree that the numerical experiments were lacking in the first version of this paper. We revised the paper and added more extensive experiments, comparing our generalization bound in various synthetic datasets to other bounds from the literature. The experiments clearly show that our bound is more realistic. The experiments are reported in Section 4 in the main part of the paper, and more extensive results and details are given in Appendix D. We also provide the plot the reviewer requested about the dependency of the generalization bound on the Lipschitz constant of the graphon (See Page 60, Figure 6), and additional plots and tables that show how the different parameters, like Lipschitz constants, affect the generalization bound in Appendix D.3.
>
> It is important to note that theoretical generalization bounds typically teach us about the asymptotic behavior of generalization, and which hyperparameters affect generalization, but they rarely give numerical bounds which are realistic and less than 1. Nevertheless, in three of the scenarios that we studied our generalization bound gives realistic bounds, which are near 0.1 (0.089, 0.133 and 0.246). In comparison, all other previous works produce generalization bounds that are orders of magnitude larger than 1, making them impractical. We believe that this may be the first real GNN learning experiment for which a theory actually proves generalization.
>
> In the experiment, we learn a classifier on graphs sampled from synthetic random graph models. We consider three datasets (see Section 4 in the main paper and Appendix D.2 for more details), and consider MPNNs with 1,2, and 3 layers. We each time train with weight decay (WD) and without weight decay (w/o WD). The results show how weight decay regularization leads to better generalization performance. This supports our theory as explained on Page 8, Lines 304-306:
>
> “Another insight is that the generalization bound becomes smaller the smaller the Lipschitz constants of the message and update functions are (see Remark 3.4). This indicates that regularization methods like weight decay promote generalization.”
>
> An example of the results on one of the datasets that we considered is reported in the Common Response. The rest of the results are reported in Section 4 of the revised paper, and in Appendix D (especially in Appendices D.2.7 and D.3).
>
> We hope that these new results, which corroborate our theory, will convince the reviewer to improve their overall assessment of the paper.
>
> > **Q:**
> *“In Appendix D, it is mentioned that you approximate the continuous MPNN with an MPNN on a large graph (lets call this output Y∈RF), which is then subsampled to smaller graphs, for which the ''finite'' MPNN outputs (call these $Z_1,…,Z_{50}\in \mathbb{R}^F$) are compared. This may make it easier for the ''finite'' MPNNs to be close to Y.  Perhaps it would be better to sample several large graphs, giving outputs $Y_1,…,Y_a \in \mathbb{R}^F$, and then measure how close ''finite'' MPNNs are to $1/a\sum_{i=1}^aY_i$ on independently sampled small graphs.”*
>
> **A:** We agree that this experimental setup may make it easier for the finite MPNNs to be close to Y. We thus changed the setup: We still consider one large graph. But instead of sampling from the large graph, we now sample the sequence of small graphs i.i.d. from the initial random graph model. This makes the setup harder than the reviewer suggests, since it assumes that one large graph already approximates the RGM instead of the average of many large graphs. By the central limit theorem, the output of a MPNN on one graph has a larger variance than the average output on many graphs. Still, the results of this new experiment show convergence of the MPNNs applied to the small graphs to the MPNN applied to the large graph. The convergence speed decreased, compared to the old setup, from -0.58 to -0.38, which is now closer to our theoretical convergence speed of -0.25. We thank the reviewer for this helpful correction! The revised figures can be seen in Subsection D.1 of the appendix (Figure 2).
>
> Please note that we moved the convergence experiments to the appendix, in order to have space for the new generalization experiments, which are more important. We believe that now the experiments support our theory much better.
>
> > **Q:** *“Perhaps mention in Section 1 that your analysis holds for mean aggregation (apologies if I have missed the mention).”*
>
> **A:** Thanks, we added it to Section 1 (See Page 2, Line 37).

---

> > ### Comment · Reviewer_tNHF · 2022-08-03
> > **Reply to the authors**
> >
> > We thank the authors for listening to and addressing reviewer comments. The numerical experiments seem much improved. In particular, this new empirical setup for measuring convergence (sampling the "finite" graphs independently of the "large" graph) is much better, and shows a convergence speed closer to the theory. Also, the numerical comparison of the various bounds in the literature is much appreciated. Further discussion of implications of the theory (as mentioned in discussions with other reviewers) is also very good to have.
> >
> > I will raise my score to a 7: if the theoretical results are correct, then this paper seems like a significant contribution; but I have not read the proofs in much detail.

---

### Author Response · Authors · 2022-08-02
**Common Response**

We thank the reviewers for their thorough and insightful remarks. We fully implemented all remarks in the revised version of the paper, which we believe improved the paper significantly.

The main changes:

1) **More extensive experiments on generalization**

Based on the suggestions from the reviewers, we add the following experiments.

We consider 3 synthetic datasets that correspond to 3 random graph models as shown below. All datasets have: #graphs = 100K, #nodes per graph = 50, #classes = 2, train/test split = 90%/10%.

We created three different synthetic datasets of random graphs from different graphons. We consider Erdös-Rényi graphs with edge probability 0.4, a smooth version of a stochastic block model, and an exponential radial graphon. The corresponding signals are given in Appendix D.2.1. For each graphon, we create 50K graphs of size 50. We then consider all pairs of graph models and train a binary classifier for each pair.

Here, we give as an example the results from one of these datasets. See Page 59, Table 3 for the full table, and Page 58 Figure 3 for bar charts. The rows show the results for different depths T=1,2,3 of the MPNN and for training with weight decay (WD) and without weight decay (w/o WD).

|            | T=1 WD                          | T=1 w/o WD                    | T=2 WD                        | T=2 w/o WD                    | T=3 WD                        | T=3 w/o WD                    |
|------------|---------------------------------|-------------------------------|-------------------------------|-------------------------------|-------------------------------|-------------------------------|
| Rademacher | $3.9597\times 10^0$              | $1.7328\times 10^5$          | $3.2439\times 10^3$         | $1.1943\times 10^6$          | $2.7221\times 10^5$          | $1.1762\times 10^6$          |
| PAC-Bayes       | $1.9597 \times 10 ^4$          | $6.0161\times 10^5$           | $3.0659\times 10^3$          | $4.0392\times 10^7$           | $3.9963\times 10^7$          | $6.225\times 10^9$           |
| Ours       | $\mathbf{8.9113\times 10^{-2}}$ | $\mathbf{ 1.4408\times 10^1}$ | $\mathbf{1.4788\times 10^0}$ | $\mathbf{1.3526 \times 10^2}$ | $\mathbf{1.0255 \times 10^2}$ | $\mathbf{4.3375 \times 10^3}$ |

From the table (and Table 3 in the revised Appendix D.2), we can see that our bound is significantly tighter than the Rademacher and PAC-Bayes bounds under all settings. We include more discussions in the revised version of the paper, Section 4 and Appendix D.
The results also corroborate our observation that weight decay is a good technique for promoting generalization in MPNN. See point 4 below.

2) **Making the aim and motivation for our construction clearer early on**

We revised the paper to better point out the aim and motivation for our construction. We thus added on Page 2 top, lines 38-42, right after the general intro to GNNs, the following paragraph:

“Previous works developed generalization bounds that do not depend on any model of the data, namely, graphs in these works can be generated and labeled in any arbitrary way [STH18, GJJ20, LUZ21]. In this work, we consider a generative model for the graphs which is theoretically powerful and general on the one hand, and allows much tighter generalization bounds on the other hand”

3) **Justifying our model**

We added a discussion that justifies the validity of our random graph model assumption in the revised paper in Subsection 1.1 from Page 2, Line 77 to Page 3, Line 98. We explain how graphons, as generative models for graphs, are very general. This is justified by the fact that similarity in the graphon sense is a relaxation of the combinatorial similarity of graph isomorphism. Therefore, since MPNNs cannot distinguish between isomorphic graphs, if a MPNN is to have a chance to classify graphs, graphs which sample the same graphon should come from the same class. We write:

 “The main assumption in our analysis is that graphs that are sampled from the same graphon belong to the same class. While this may seem like a limitation, it is actually a very mild and reasonable assumption…”

Please see Subsection 1.1 from Page 2, Line 77 in the revised paper for more details.

4) **Discussing real-world implications**

We wrote on Page 8 Line 304 a practical implication of our theory:

“Another insight is that the generalization bound becomes smaller the smaller the Lipschitz constants of the message and update functions  (see Remark 3.4). This indicates that regularization methods like weight decay promote generalization.’’

Other than that, we believe that followup papers, by us and others, will gradually derive tighter and tighter generalization bounds. The ultimate goal is to end up with a bound that predicts generalization in real-life scenarios. The current work is one of the first steps in this journey.


*The new sections in the revised paper are Section 1.1, Section 4, and Appendix D.*

---

### Meta-Review · Area_Chair_f8ki · 2022-08-25

**Recommendation:** Accept
**Confidence:** Certain

**Metareview:**

This paper gives a new approximation and generalization error bound for a class of MPNN (Message Passing Neural Networks) in a setting where the underlying graph is randomly generated. First, a discretization error from the continuous limit is given, and second the generalization error on a finite training set is given. Some numerical experiments are also given as an empirical evaluation of the theory.

Overall, this is a solid theoretical work with sufficient novelty. The rate of convergence is new and the community would benefit from the analysis. The presentation is also good. The readers can grasp the overall contribution rather easily and the theoretical results are also clearly described.
The major weakness of this paper is the numerical experiments. However, combined with the theoretical contribution, this paper has enough value.

The authors properly responded to the reviewers' questions. An additional experimental result is also given. I recommend the authors to include the additional experiment properly to enhance the empirical evaluation.

In summary, this paper gives a good contribution and can be accepted.

# minor point: (1) The citation style is not of NeurIPS standard. Please look at author instruction. (2) The abstract is presented in the Italic style. However, the standard format is the Roman style. I recommend to fix it.



**Award:**

No

---

### Decision · Program_Chairs · 2022-09-14

Accept